# Relative defects in relative theories:
# Trapped higher-form symmetries and irregular punctures in class S

Lakshya Bhardwaj[1], Simone Giacomelli[1,2,3],
Max Hübner[1,4] and Sakura Schäfer-Nameki[1]

**1** Mathematical Institute, University of Oxford, Andrew-Wiles Building,
Woodstock Road, Oxford, OX2 6GG, UK
**2** Dipartimento di Fisica, Università di Milano-Bicocca,
Piazza della Scienza 3, I-20126 Milano, Italy
**3** INFN, sezione di Milano-Bicocca, Piazza della Scienza 3,
I-20126 Milano, Italy
**4** Department of Physics and Astronomy, University of Pennsylvania,
Philadelphia, PA 19104, USA

## Abstract

A relative theory is a boundary condition of a higher-dimensional topological quantum field theory (TQFT), and carries a non-trivial defect group formed by mutually non-local defects living in the relative theory. Prime examples are 6d $\mathcal{N} = (2,0)$ theories that are boundary conditions of 7d TQFTs, with the defect group arising from surface defects. In this paper, we study codimension-two defects in 6d $\mathcal{N} = (2,0)$ theories, and find that the line defects living inside these codimension-two defects are mutually non-local and hence also form a defect group. Thus, codimension-two defects in a 6d $\mathcal{N} = (2,0)$ theory are relative defects living inside a relative theory. These relative defects provide boundary conditions for topological defects of the 7d bulk TQFT. A codimension-two defect carrying a non-trivial defect group acts as an irregular puncture when used in the construction of 4d $\mathcal{N} = 2$ Class S theories. The defect group associated to such an irregular puncture provides extra "trapped" contributions to the 1-form symmetries of the resulting Class S theories. We determine the defect groups associated to large classes of both conformal and non-conformal irregular punctures. Along the way, we discover many new classes of irregular punctures. A key role in the analysis of defect groups is played by two different geometric descriptions of the punctures in Type IIB string theory: one provided by isolated hypersurface singularities in Calabi-Yau threefolds, and the other provided by ALE fibrations with monodromies.

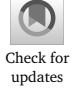

# 1 Introduction

Recently, the study of higher-form symmetries [1–3] has gained momentum [4–58], as these symmetries provide a lot of insight into deep physical phenomena in quantum field theories. On the one hand, they provide insights into the phase structure and confinement properties in the IR, while on the other hand, they provide insights into the properties of extended defects in the UV. More abstractly, they encode information about the class of observables that can be computed in a given quantum field theory, as a higher-symmetry is associated to a class of backgrounds that can be turned on while computing correlation functions.

Higher-form symmetries are often entwined with a larger structure known as *defect group*[1]. These are groups formed by equivalence classes of mutually non-local defects in a theory. The non-locality of the defects implies that the theory is ill-defined on its own. For it to be well-defined, the theory needs to arise on the boundary of a topological quantum field theory (TQFT) in one higher dimension. We call such theories as *relative theories*[2]. On the other

---

[1]This terminology was introduced in [4].

[2]An important note on terminology: Our definition of 'relative theory' differs from the definition employed in the work [59] in a key manner. In that work, any theory arising on the boundary of a higher-dimensional TQFT is called a relative theory, while a theory existing independently of a higher-dimensional TQFT is called an absolute theory. According to this definition, the theories having a 't Hooft anomaly, which can be thought of as theories living on the boundaries of SPT phases in one higher dimension, are relative theories. For us, one important feature

hand, theories without mutually non-local defects are called *absolute theories*.

A natural generalization of the notion of a relative theory is the notion of a *relative defect* inside a bulk theory. We define a relative defect to be a defect that carries mutually non-local sub-defects. As a consequence, for a relative defect to be well-defined, it needs to be attached to a topological system in one higher dimension. For a relative defect inside an absolute bulk theory, the corresponding topological system is a TQFT. On the other hand, for a relative defect inside a relative bulk theory, the corresponding system is a topological defect of the TQFT associated to the relative bulk theory. The defect group of a relative defect captures *higher-form symmetries localized on the worldvolume of the defect*, along with its interplay with the higher-form symmetries of the bulk theory.

In this paper, we show that codimension-two defects inside a bulk 6d $\mathcal{N} = (2, 0)$ theory provide examples of relative defects inside a relative bulk theory. We find that line sub-defects inside the codimension-two defects are mutually non-local and hence form a non-trivial defect group in general.

One necessary (but not sufficient) condition for a codimension-two defect to be relative is that it should provide an *irregular puncture* when used for the Class S construction of 4d $\mathcal{N} = 2$ theories via compactification of 6d $\mathcal{N} = (2, 0)$ theories on punctured Riemann surfaces [60–72]. In this context, an irregular puncture is defined as a singularity of a Higgs field (participating in a Hitchin system on the Riemann surface) where poles of order higher than a simple pole appear. On the other hand, a *regular puncture* is a singularity where only simple poles of the Higgs field appear. This follows from the results of [31], which can be rephrased to state that a codimension-two defect associated to a regular puncture carries a trivial defect group.

One interesting application of the defect groups associated to codimension-two defects is in the computation of the *1-form symmetry* of a 4d $\mathcal{N} = 2$ Class S theory constructed using irregular punctures. There are many interesting 4d $\mathcal{N} = 2$ theories that lie in this class, including 4d $\mathcal{N} = 2$ Argyres-Douglas SCFTs and asymptotically free 4d $\mathcal{N} = 2$ theories. Prior work [31], building upon the works [1,73,74], has provided a general recipe for computing the 1-form symmetries of 4d $\mathcal{N} = 2$ Class S theories involving only regular punctures. In this work, we extend their analysis to include arbitrary irregular punctures. The defect groups associated to the irregular punctures play a key role in this analysis.

When only regular punctures are involved, the 1-form symmetry is determined roughly by the 1-cycles on the Riemann surface. When irregular punctures are involved, there are extra contributions to the 1-form symmetry that are localized at the locations of the irregular punctures. We call such extra contributions as *trapped 1-form symmetries*.

Striking examples of trapped 1-form symmetries are provided by Argyres-Douglas (AD) theories of type AD[$G, G'$] that are constructed by compactifying 6d $\mathcal{N} = (2, 0)$ theories on a sphere with a single puncture that is irregular. Such AD theories, and also a vast number of other 4d $\mathcal{N} = 2$ SCFTs, can also be constructed by compactifying Type IIB on canonical isolated hypersurface Calabi-Yau three-fold singularities (IHS) $X$ [17,29,30,36,55,75–86], for a recent review see [87]. Using string theoretic methods, the 1-form symmetries of these 4d $\mathcal{N} = 2$ SCFTs can be computed from the topology of the boundary 5-manifold $\partial X$ [17,29,30,36,55]. It is found that the 1-form symmetries of such AD theories are generally non-trivial. Since the sphere used in the Class S construction does not carry any non-trivial 1-cycles, the 1-form symmetries of these AD theories must be entirely trapped. Indeed, the general analysis developed in this paper correctly reproduces the 1-form symmetries of these AD theories from the defect groups associated to the irregular puncture.

The irregular punctures introduced in [81,88,89] will be called *IHS punctures* because the

---

defining a relative theory is the existence of mutually non-local defects. Thus, we would call a theory having a 't Hooft anomaly as an absolute theory rather than a relative theory.

$4d$ $\mathcal{N}=2$ SCFTs obtained by compactifying $6d$ $\mathcal{N}=(2,0)$ theories on spheres with these punctures can also be constructed by compactifying Type IIB on IHS singularities. For untwisted punctures, this relationship was discussed in [81]. In this paper, we show that this relationship can also be extended to the twisted punctures introduced in [81, 89].

In addition to the punctures introduced in [81, 88, 89], we study many other classes of irregular punctures. Many of these punctures have not appeared in prior literature. We divide punctures into two classes: conformal and non-conformal punctures. The distinction is made by considering the $4d$ $\mathcal{N}=2$ theory obtained by compactifying the $6d$ $\mathcal{N}=(2,0)$ theory corresponding to the puncture on a sphere with the puncture inserted at one point (and no other punctures). If this $4d$ $\mathcal{N}=2$ theory is (super)conformal, then we call the puncture a *conformal puncture*. On the other hand, if this $4d$ $\mathcal{N}=2$ theory is not conformal, then we call the puncture a *non-conformal puncture*. For example, the IHS punctures are conformal punctures. However, they are not all the conformal punctures, and we study many other classes of conformal punctures that correspond to non-isolated singularities. In addition to these, we also study many classes of non-conformal punctures. A subclass of the conformal and non-conformal punctures considered in this paper can be constructed using Hanany-Witten brane constructions [90–93] in Type IIA superstring theory.

The bulk of this paper is devoted to the computation of defect groups associated to codimension-two defects in $6d$ $\mathcal{N}=(2,0)$ theories of types A, D and E. To each (conformal or non-conformal) puncture, we associate a $4d$ $\mathcal{N}=2$ generalized quiver theory having the property that the 1-form symmetry of the generalized quiver determines completely the data associated to the defect group of the puncture. Such a theory is a gauge theory containing various kinds of gauge algebras, but the matter content can be provided not only by hypermultiplets, but also by strongly coupled $4d$ $\mathcal{N}=2$ SCFTs that we call "matter SCFTs". The 1-form symmetry of the generalized quiver theory, and hence all the data associated to the defect group of the puncture, is computed using the properties of the matter SCFTs.

For the punctures admitting a Type IIA construction, the generalized quiver is simply read off from the associated brane configuration. On the other hand, for punctures not admitting a Type IIA construction, the associated generalized quiver is conjectural, and we provide many pieces of evidence to support this conjecture, that we discuss below.

First of all, a piece of the defect group (that contains both trapped and non-trapped parts) can be computed by using the data of the Higgs field in the vicinity of the puncture. Using the data of the Higgs field, one can realize the puncture by compactifying Type IIB on an ALE fibration over a punctured plane. The monodromy of the Higgs field around the puncture captures the monodromy of the ALE fibration. Now one can apply the tools developed in [17, 29, 30, 36, 55] to deduce this piece of the defect group of the puncture from the data of the monodromy of the ALE fibration. However, this doesn't provide full information related to the defect group.

Second, for the IHS punctures, the trapped part of the defect group can be computed using Type IIB compactified on the corresponding IHS singularity, using the technology developed in [17, 29, 30, 36, 55]. The obtained trapped defect group can then be checked against the trapped defect group predicted by the conjecture. However, this does not provide information on the non-trapped part of the defect group.

Third, we use other types of IHS singularities discussed in [29] that construct $4d$ $\mathcal{N}=2$ trinion theories. These trinion theories are composed out of the data of three irregular punctures. The defect group of a trinion theory can be obtained using the data of the defect groups of the three punctures, and it involves not only the trapped parts of these defect groups, but also some of the non-trapped parts. On the other hand, the defect group of the trinion theory can be independently obtained as in [29]. Matching the two results provides a check for the non-trapped part of the defect groups of punctures predicted by our proposal.

Finally, for untwisted A type punctures, the defect group of the puncture can be read from the 1-form symmetry of a $3d$ Lagrangian theory obtained by circle reduction of a $4d$ $\mathcal{N} = 2$ Class S theory constructed using the puncture (and a $\mathcal{P}_0$ puncture). This provides another check for our proposal.

We also study the defect groups of untwisted IHS punctures for the E-type $(2,0)$ theory. For such punctures, we can compute the trapped part of the defect group using the IHS singularity. Moreover, using the ALE fibration description, we can also compute a piece containing a combination of trapped and non-trapped parts. Now, it turns out for such IHS punctures, that these two pieces of information is enough to determine the whole information about the defect group associated to these punctures.

The rest of this paper is structured as follows: Section 2 discusses generalities about relative theories and relative defects. Section 3 provides evidence for the existence of 1-form symmetries trapped inside irregular punctures. Section 4 describes how the 1-form symmetry of an arbitrary class S theory can be described in terms of defect groups associated to the participating punctures. Section 5 discusses various techniques we use for computing the defect groups associated to punctures. In particular, subsection 5.3 discusses the computation of (part of) defect group associated to a puncture by realizing it as an ALE fibration with a monodromy in Type IIB string theory. Section 6 discusses the map between IHS punctures and IHS singularities. Using these IHS singularities, we compute the trapped parts of the defect groups associated to IHS punctures. Sections 7, 8 and 9 discuss our main proposals for computing defect groups associated to large classes of untwisted A- and D-types, and twisted D-type punctures. The explicit forms of the defect groups are provided, and many checks are performed by computing parts of these defect groups via other methods. Section 10 computes the defect groups of untwisted E-type IHS punctures by combining the results of the IHS computation with the results of the ALE fibration computation. Finally, we have a couple of appendices. Appendix A provides a glossary of the most frequently used notations in the paper. Appendix B presents the `Mathematica` code used to derive the monodromy action on the spectral cover sheets about arbitrary punctures which is the key piece of data informing the boundary topology in ALE fibration picture.

## 2 Relative Defects and Relative Theories

We begin by discussing the notions of relative theories and relative defects. This is important because our starting point, namely $6d$ $\mathcal{N} = (2,0)$ theories, are relative theories. We are interested in understanding the 1-form symmetries of $4d$ $\mathcal{N} = 2$ Class S theories obtained by compactifying $6d$ $\mathcal{N} = (2,0)$ theories with arbitrary regular and irregular punctures. The contribution to the 1-form symmetry of the irregular puncture is encoded in a defect group associated to the puncture, which in turn is associated to the fact that these punctures are relative codimension-two defects inside $6d$ $\mathcal{N} = (2,0)$ theories.

### 2.1 Relative Theories

**Non-locality of defects.** Many interesting theories are *relative* [3,4,25,59,94–96]. That is, they carry defects that are mutually non-local. In such theories, the non-locality occurs between $p$-dimensional defects and $(d - p - 2)$-dimensional defects, where $d$ is the spacetime dimension of the relative theory. The non-locality exhibits itself as a phase ambiguity in defining correlation functions of these defects. Consider such a relative theory $\mathfrak{T}_d$ on a Euclidean spacetime manifold $M_d$. Consider the correlation function on $M_d$ of a $p$-dimensional defect $\mathfrak{D}_p$ inserted along a sub-manifold $\Sigma_p$, and a $(d-p-2)$-dimensional defect $\mathfrak{D}_{d-p-2}$ inserted along a sub-manifold $\Sigma_{d-p-2}$. As $\Sigma_p$ is moved in a loop around $\Sigma_{d-p-2}$, the correlation function

comes back to itself along with an additional phase

$$\exp\left(2\pi i \left\langle \mathfrak{D}_p, \mathfrak{D}_{d-p-2} \right\rangle\right), \tag{1}$$

where $\left\langle \mathfrak{D}_p, \mathfrak{D}_{d-p-2} \right\rangle \in \mathbb{R}/\mathbb{Z}$ captures the mutual non-locality between the defects $\mathfrak{D}_p$ and $\mathfrak{D}_{d-p-2}$. Thus the correlation function under consideration suffers from the above phase ambiguity.

**Examples of relative theories.**   Well-known examples of relative theories are $2d$ chiral RCFTs [97–102], for which the non-locality is exhibited by local operators (i.e. 0-dimensional defects) associated to modules of the chiral algebras. Another class of well-known examples are $6d$ $\mathcal{N} = (2,0)$ SCFTs [94] specified by Lie algebras $A_m$, $D_n$, $E_6$ and $E_7$ (but not $E_8$), for which the non-locality is exhibited by 2-dimensional surface defects in these theories.

**TQFT associated to a relative theory.**   Such a relative theory $\mathfrak{T}_d$ is better understood as a non-topological boundary condition of a non-invertible $(d+1)$-dimensional TQFT $\mathfrak{S}_{d+1}$[3]. The TQFT carries *invertible* topological defects[4] described by a group

$$\mathcal{L} = \prod_{p=1}^{d-1} \mathcal{L}_p = \mathcal{L}_1 \times \mathcal{L}_2 \times \cdots \times \mathcal{L}_{d-2} \times \mathcal{L}_{d-1}, \tag{2}$$

where $\mathcal{L}_p$ is the abelian group formed by $p$-dimensional topological defects under fusion. The $p$-dimensional topological defects braid non-trivially with $(d-p)$-dimensional topological defects. The braiding defines a bi-homomorphism

$$\langle \cdot, \cdot \rangle : \mathcal{L}_p \times \mathcal{L}_{d-p} \to \mathbb{R}/\mathbb{Z}. \tag{3}$$

Moreover the bi-homomorphism is such that it makes $\mathcal{L}_{d-p}$ isomorphic to the Pontraygin dual $\widehat{\mathcal{L}}_p$ of $\mathcal{L}_p$. That is,

$$\mathcal{L}_{d-p} \cong \widehat{\mathcal{L}}_p. \tag{4}$$

These bihomomorphisms define what we call a *pairing* on the group $\mathcal{L}$. In addition to the $\mathcal{L}_p$ participating in $\mathcal{L}$, one can additionally consider a group $\mathcal{L}_d$ of $d$-dimensional invertible topological defects in the TQFT $\mathfrak{S}_{d+1}$, that may have a non-trivial action on the topological defects in $\mathcal{L}_{p<d}$.

The group $\mathcal{L}_p$ forms the $(d-p)$-form symmetry group of the TQFT $\mathfrak{S}_{d+1}$. Thus $\mathcal{L}$ captures the higher-form symmetries of $\mathfrak{S}_{d+1}$, and $\mathcal{L}_d$ captures 0-form symmetries (that can act on the higher-form symmetries valued in $\mathcal{L}$). The pairing on $\mathcal{L}$ describes 't Hooft anomalies between these higher-form symmetries.

**Non-locality from braiding.**   The $p$-dimensional defects $\mathfrak{D}_p$ of the relative theory $\mathfrak{T}_d$ arise at the end-points of the $(p+1)$-dimensional topological defects of the TQFT $\mathfrak{S}_{d+1}$. This includes both invertible and non-invertible $(p+1)$-dimensional defects. For example, for the surface defects of $6d$ $\mathcal{N} = (2,0)$ theories, the corresponding topological 3-dimensional defects are all invertible. On the other hand, for the local operators of chiral RCFTs, the corresponding topological line defects include both invertible and non-invertible ones. The non-locality of these defects of the relative theory can now be understood in terms of braiding of the corresponding topological defects of the TQFT.

---

[3]This is sometimes also referred to as the Symmetry TFT, or SymTFT in [57].

[4]There can also be non-invertible topological defects that we ignore in what follows. We will consider some properties of such non-invertible topological defects when we discuss relative defects in relative theories later.

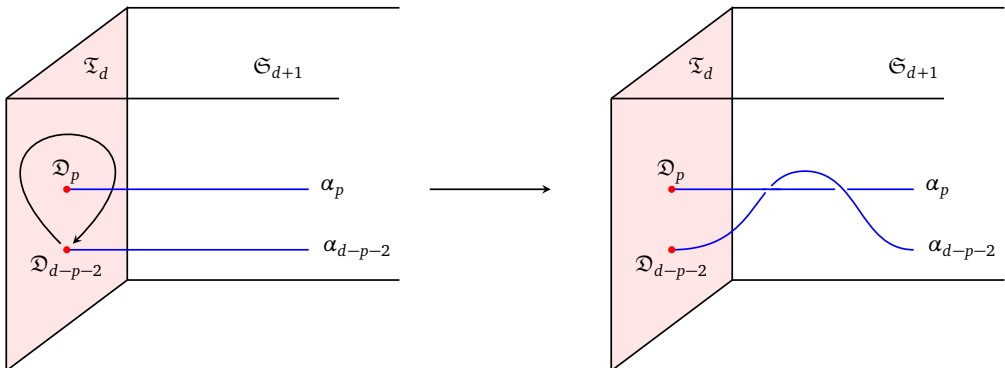

Figure 1: Moving a defect $\mathfrak{D}_{d-p-2}$ of the relative theory $\mathfrak{T}_d$ around another defect $\mathfrak{D}_p$ of the relative theory $\mathfrak{T}_d$ creates a braiding between the topological defects $\alpha_p \in \mathcal{L}_{p+1}$ and $\alpha_{d-p-2} \in \mathcal{L}_{d-p-1}$ of the TQFT $\mathfrak{S}_{d+1}$.

$$\mathfrak{D}_p \quad \underline{\phantom{xxxxxxxxxxxx} \bullet \phantom{xxxxxxxxxxxx}} \quad \mathfrak{D}_p' \quad \implies \quad \mathfrak{D}_p \sim \mathfrak{D}_p'$$
$$\mathfrak{J}_{p-1}$$

Figure 2: An equivalence relation in which two defects $\mathfrak{D}_p$ and $\mathfrak{D}_p'$ are equivalent if there exists a non-trivial junction $\mathfrak{J}_{p-1}$ between them.

Restricting ourselves to the invertibles, let us consider two defects $\mathfrak{D}_p$ and $\mathfrak{D}_{d-p-2}$ of the relative theory arising at the end-points of topological defects $\alpha_p \in \mathcal{L}_{p+1}$ and $\alpha_{d-p-2} \in \mathcal{L}_{d-p-1}$ respectively. Then, we have the equality

$$\left\langle \mathfrak{D}_p, \mathfrak{D}_{d-p-2} \right\rangle = \left\langle \alpha_p, \alpha_{d-p-2} \right\rangle, \tag{5}$$

where the left hand side captures the non-locality between $\mathfrak{D}_p$ and $\mathfrak{D}_{d-p-2}$, while the right hand side captures the braiding between the topological defects $\alpha_p \in \mathcal{L}_{p+1}$ and $\alpha_{d-p-2} \in \mathcal{L}_{d-p-1}$. See figure 1.

**Equivalence classes of defects under screenings.** For $p > 0$, each $(p+1)$-dimensional topological defect (that cannot be expressed as a sum of other $(p+1)$-dimensional topological defects) of the TQFT $\mathfrak{S}_{d+1}$ characterizes an equivalence class of $p$-dimensional defects of the relative theory $\mathfrak{T}_d$. Two $p$-dimensional defects $\mathfrak{D}_p$ and $\mathfrak{D}_p'$ are in the same equivalence class if there exists a $(p-1)$-dimensional junction $\mathfrak{J}_{p-1}$ defect at the intersection of $\mathfrak{D}_p$ and $\mathfrak{D}_p'$. See figure 2. One also says that, such a $\mathfrak{J}_{p-1}$ *screens* $\mathfrak{D}_p$ to $\mathfrak{D}_p'$.

**6d $\mathcal{N} = (2,0)$ theories as relative theories.** The group $\mathcal{L}$, along with the pairing on it, is often referred to as the *defect group* of the relative theory $\mathfrak{T}_d$. For example, the defect group of a 6d $\mathcal{N} = (2,0)$ SCFT specified by an A, D, E algebra $\mathfrak{g}$ is completely localized in $\mathcal{L}_3$ and takes the form

$$\mathcal{L} = \mathcal{L}_3 = \widehat{Z}_G, \tag{6}$$

where $\widehat{Z}_G$ is the Pontryagin dual of the center $Z_G$ of the simply connected group $G$ associated to the Lie algebra $\mathfrak{g}$. The pairing on $\mathcal{L}$ is just a bi-homomorphism $\mathcal{L}_3 \times \mathcal{L}_3 = \widehat{Z}_G \times \widehat{Z}_G \to \mathbb{R}/\mathbb{Z}$. The above bi-homomorphism $\widehat{Z}_G \times \widehat{Z}_G \to \mathbb{R}/\mathbb{Z}$ provides an isomorphism $\widehat{Z}_G \to Z_G$ that will feature prominently in the later parts of the paper. A 6d $\mathcal{N} = (2,0)$ SCFT of type $\mathfrak{g}$ also contains a non-trivial

$$\mathcal{L}_d = \mathcal{L}_6 = \mathcal{O}_{\mathfrak{g}}, \tag{7}$$

Table 1: Defect group $\mathcal{L}_3$, its pairing $\langle \cdot, \cdot \rangle$ and the 0-form group $\mathcal{O}_{\mathfrak{g}}$ for 6d $\mathcal{N} = (2,0)$ theory of type $\mathfrak{g}$. Trivial groups are denoted by zero. We denote a generator of $\mathcal{L}_3$ for $\mathfrak{g} = A_{n-1}, E_6, E_7$ as $f$; a generator of $\mathcal{L}_3$ for $\mathfrak{g} = D_{2n+1}$ as $s$; and generators of $\mathcal{L}_3 \simeq \mathbb{Z}_2 \times \mathbb{Z}_2$ for $\mathfrak{g} = D_{2n}$ as $s, c$. $S_3$ denotes the group formed by permutations of three objects.

| $\mathfrak{g}$ | $\mathcal{L}_3$ | $\mathcal{O}_{\mathfrak{g}}$ | $\langle \cdot, \cdot \rangle$ |
|---|---|---|---|
| $A_{n-1}$ | $\mathbb{Z}_n$ | $\mathbb{Z}_2$ | $\langle f, f \rangle = \frac{1}{n}$ |
| $D_4$ | $\mathbb{Z}_2 \times \mathbb{Z}_2$ | $S_3$ | $\langle s,s \rangle = 0, \quad \langle c,c \rangle = 0, \quad \langle s,c \rangle = \frac{1}{2}$ |
| $D_{4n+1}$ | $\mathbb{Z}_4$ | $\mathbb{Z}_2$ | $\langle s,s \rangle = \frac{3}{4}$ |
| $D_{4n+2}$ | $\mathbb{Z}_2 \times \mathbb{Z}_2$ | $\mathbb{Z}_2$ | $\langle s,s \rangle = \frac{1}{2}, \quad \langle c,c \rangle = \frac{1}{2}, \quad \langle s,c \rangle = 0$ |
| $D_{4n+3}$ | $\mathbb{Z}_4$ | $\mathbb{Z}_2$ | $\langle s,s \rangle = \frac{1}{4}$ |
| $D_{4n+4}$ | $\mathbb{Z}_2 \times \mathbb{Z}_2$ | $\mathbb{Z}_2$ | $\langle s,s \rangle = 0, \quad \langle c,c \rangle = 0, \quad \langle s,c \rangle = \frac{1}{2}$ |
| $E_6$ | $\mathbb{Z}_3$ | $\mathbb{Z}_2$ | $\langle f, f \rangle = \frac{2}{3}$ |
| $E_7$ | $\mathbb{Z}_2$ | $0$ | $\langle f, f \rangle = \frac{1}{2}$ |
| $E_8$ | $0$ | $0$ | — |

where $\mathcal{O}_{\mathfrak{g}}$ is the group formed by outer-automorphisms of $\mathfrak{g}$. There is furthermore an action of $\mathcal{L}_d$ on $\mathcal{L}$, which is just the natural action of $\mathcal{O}_{\mathfrak{g}}$ on $\widehat{Z}_G$. The data of the defect group, pairing and outer-automorphism group for different values of $\mathfrak{g}$ is tabulated in table 1.

## 2.2 Absolute Theories from Relative Theories

**Polarization.** Starting from a relative theory, one can construct many *absolute* theories, where an absolute theory is defined by the fact that its genuine[5] defects do not have any non-locality. Thus the correlation functions of genuine defects in an absolute theory do not suffer from phase ambiguities. A general procedure for constructing absolute theories out of a relative theory $\mathfrak{T}_d$ employs the use of a topological boundary condition[6] $\mathfrak{B}_d^\Lambda$ of the TQFT $\mathfrak{S}_{d+1}$. An absolute $d$-dimensional theory $\mathfrak{T}_d^\Lambda$ is then obtained by compactifying the TQFT $\mathfrak{S}_{d+1}$ on an interval, with the relative theory $\mathfrak{T}_d$ placed at one end of the interval, and the topological boundary condition $\mathfrak{B}_d^\Lambda$ placed at the other end of the interval. See figure 3.

The essential data of the boundary condition $\mathfrak{B}_d^\Lambda$, as far as the invertible topological defects are concerned, is the *polarization*[7] $\Lambda \subseteq \mathcal{L}$, which is a maximal group of the form

$$\Lambda = \prod_{p=1}^{d-1} \Lambda_p \,, \tag{8}$$

---

[5]We call a defect *genuine* if it can be defined independently of any higher-dimensional defects. On the other hand, a *non-genuine* defect is one that arises at the junctions, boundaries or corners of other higher-dimensional defects.

[6]Since we regard theories with anomalies as absolute theories, what we call a topological boundary condition is not actually a boundary condition when there are anomalies. In general, $\mathfrak{B}_d^\Lambda$ is a topological interface between the TQFT $\mathfrak{S}_{d+1}$ and the $(d+1)$-dimensional SPT phase capturing the anomaly. See figure 3. For convenience, we will refer to $\mathfrak{B}_d^\Lambda$ as a 'topological boundary condition of the TQFT $\mathfrak{S}_{d+1}$' in what follows.

[7]Here we only consider what are called *pure* polarizations in [25]. There can also be *mixed* polarizations that are discussed in that paper.

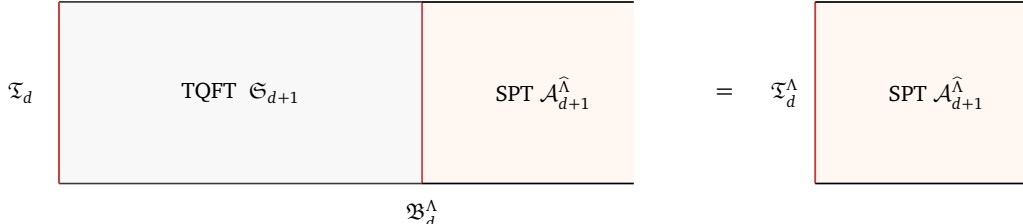

Figure 3: A polarization $\Lambda$ is associated to a topological interface (that we refer to as a "boundary condition") between the TQFT $\mathfrak{S}_{d+1}$ and an SPT phase $\mathcal{A}_{d+1}^{\widehat{\Lambda}}$. A compactification of the TQFT $\mathfrak{S}_{d+1}$ on a segment with the relative theory $\mathfrak{T}_d$ at one end and $\mathfrak{B}_d^{\Lambda}$ at the other end, leads to the absolute theory $\mathfrak{T}_d^{\Lambda}$, which comes attached to the SPT phase $\mathcal{A}_{d+1}^{\widehat{\Lambda}}$. The SPT phase captures the 't Hooft anomaly of the higher-form symmetry $\widehat{\Lambda}$ of the absolute theory $\mathfrak{T}_d^{\Lambda}$.

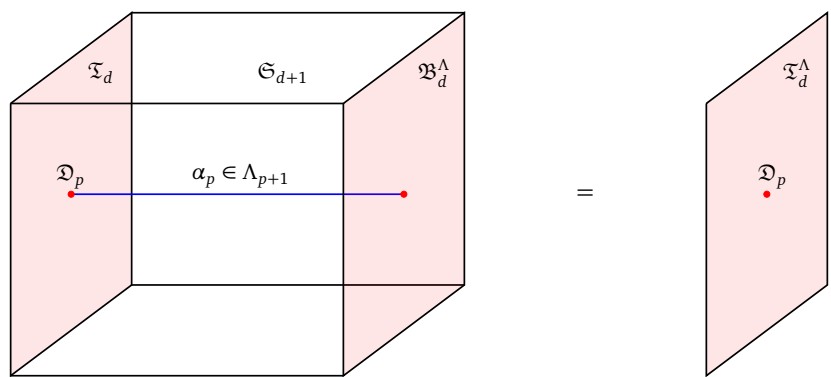

Figure 4: A defect $\mathfrak{D}_p$ of the relative theory $\mathfrak{T}_d$ attached to a topological defect $\alpha_p \in \Lambda_{p+1}$ of the TQFT $\mathfrak{S}_{d+1}$ becomes a genuine defect of the absolute theory $\mathfrak{T}_d^{\Lambda}$, as $\alpha_p$ can end on the topological boundary $\mathfrak{B}_d^{\Lambda}$.

where each $\Lambda_p$ is a subgroup of $\mathcal{L}_p$, such that

$$\langle \alpha_p, \alpha_{d-p-2} \rangle = 0, \tag{9}$$

for arbitrary $\alpha_p \in \Lambda_{p+1}$ and $\alpha_{d-p-2} \in \Lambda_{d-p-1}$. In other words, a polarization $\Lambda$ is a maximal subset of the defect group $\mathcal{L}$ such that the pairing on $\mathcal{L}$ trivializes when restricted to $\Lambda$.

**Genuine defects.** The group $\Lambda_p$ characterizes the subgroup of $p$-dimensional topological defects valued in $\mathcal{L}_p$ of the TQFT $\mathfrak{S}_{d+1}$ that can end on the boundary $\mathfrak{B}_d^{\Lambda}$. Thus, the $p$-dimensional defects of the relative theory $\mathfrak{T}_d$ that lie in the equivalence classes lying in $\Lambda_{p+1} \subseteq \mathcal{L}_{p+1}$ become genuine $p$-dimensional defects of the absolute theory $\mathfrak{T}_d^{\Lambda}$. See figure 4. The condition (9) now ensures that these genuine defects are mutually local, which is required for $\mathfrak{T}_d^{\Lambda}$ to be an absolute theory.

**Higher-form symmetries.** Since $p$-dimensional topological defects valued in $\Lambda_p$ of the TQFT $\mathfrak{S}_{d+1}$ can end on the boundary $\mathfrak{B}_d^{\Lambda}$, they descend to the trivial $p$-dimensional topological defect in the absolute theory $\mathfrak{T}_d^{\Lambda}$. More generally, the $p$-dimensional topological defects valued in $\mathcal{L}_p$ of the TQFT $\mathfrak{S}_{d+1}$ descend to $p$-dimensional topological defects of the absolute theory $\mathfrak{T}_d^{\Lambda}$ valued in $\mathcal{L}_p/\Lambda_p$. In total, the invertible topological defects of the absolute theory $\mathfrak{T}_d^{\Lambda}$ form a

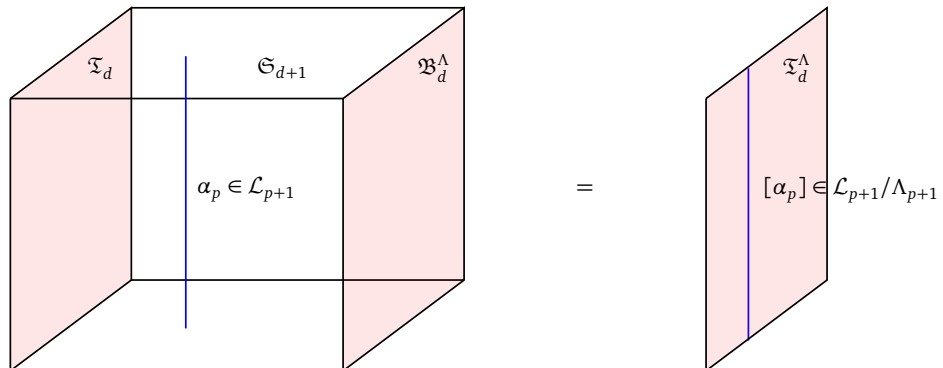

Figure 5: A topological defect $\alpha_p \in \mathcal{L}_{p+1}$ of the TQFT $\mathfrak{S}_{d+1}$ descends to a $(p + 1)$-dimensional topological defect $[\alpha_p] \in \mathcal{L}_{p+1}/\Lambda_{p+1}$ of the absolute theory $\mathfrak{T}_d^\Lambda$.

group

$$\mathcal{L}/\Lambda = \prod_{p=1}^{d-1} \mathcal{L}_p/\Lambda_p. \tag{10}$$

See figure 5. These topological defects give rise to higher-form symmetries of the the absolute theory $\mathfrak{T}_d^\Lambda$. The $p$-form symmetry group $\Gamma^{(p)}\left[\mathfrak{T}_d^\Lambda\right]$ of the absolute theory $\mathfrak{T}_d^\Lambda$ is

$$\Gamma^{(p)}\left[\mathfrak{T}_d^\Lambda\right] = \mathcal{L}_{d-p-1}/\Lambda_{d-p-1} \cong \widehat{\Lambda}_{p+1}, \tag{11}$$

where $\widehat{\Lambda}_{p+1}$ denotes the Pontryagin dual of $\Lambda_{p+1}$. The isomorphism $\mathcal{L}_{d-p-1}/\Lambda_{d-p-1} \cong \widehat{\Lambda}_{p+1}$ follows from the isomorphism (4) and the fact that the pairing between $\Lambda_{p+1}$ and $\Lambda_{d-p-1}$ is trivial.

The genuine $p$-dimensional defects of $\mathfrak{T}_d^\Lambda$ lying in equivalence classes in $\Lambda_{p+1}$ are charged objects under the $p$-form symmetry $\Gamma^{(p)}\left[\mathfrak{T}_d^\Lambda\right]$. Consider a $p$-dimensional defect $\mathfrak{D}_p$ of $\mathfrak{T}_d^\Lambda$ lying in an equivalence class $\alpha_p \in \Lambda_{p+1}$. The action of a $p$-form symmetry element $\widehat{\alpha}_p \in \widehat{\Lambda}_{p+1}$ on the defect $\mathfrak{D}_p$ is given by the phase factor

$$\widehat{\alpha}_p(\alpha_p) \in U(1), \tag{12}$$

which is the image of the element $\alpha_p \in \Lambda_{p+1}$ under the homomorphism $\widehat{\alpha}_p : \widehat{\Lambda}_{p+1} \to U(1)$. Alternatively, let $\alpha_{d-p-2} \in \mathcal{L}_{d-p-1}$ be a lift of the element of $\mathcal{L}_{d-p-1}/\Lambda_{d-p-1}$ obtained by applying the isomorphism $\widehat{\Lambda}_{p+1} \to \mathcal{L}_{d-p-1}/\Lambda_{d-p-1}$ on the element $\widehat{\alpha}_p \in \widehat{\Lambda}_{p+1}$. Then, the phase factor (12) can be also be expressed as

$$\widehat{\alpha}_p(\alpha_p) = \langle \alpha_{d-p-2}, \alpha_p \rangle, \tag{13}$$

in terms of the pairing on $\mathcal{L}$. This has a nice pictorial representation shown in figure 6.

**Non-genuine defects.** The $p$-dimensional defects of the relative theory $\mathfrak{T}_d$ lying in equivalence classes in the set $\mathcal{L}_{p+1} - \Lambda_{p+1}$ become non-genuine defects of the absolute theory $\mathfrak{T}_d^\Lambda$. Consider a defect $\mathfrak{D}_p$ lying in an equivalence class $\alpha_p \in \mathcal{L}_{p+1} - \Lambda_{p+1}$. It arises on the boundary of a topological defect of the absolute theory $\mathfrak{T}_d^\Lambda$ described by the element $[\alpha_p] \in \mathcal{L}_{p+1}/\Lambda_{p+1}$ obtained by projecting $\alpha_p \in \mathcal{L}_{p+1}$ to $\mathcal{L}_{p+1}/\Lambda_{p+1}$. In other words, $\mathfrak{D}_p$ is a defect arising in the twisted sector of the absolute theory $\mathfrak{T}_d^\Lambda$ associated to the higher-form symmetry element $[\alpha_p]$.

**Absolute $\mathcal{N} = (2, 0)$ theories.** One can now construct absolute 6d $\mathcal{N} = (2, 0)$ theories by classifying polarizations of $\mathcal{L} = \widehat{Z}_G$. The result of the classification can be found in the table right before table 1 of [25]. For example, for $\mathfrak{g} = \mathfrak{so}(2n)$, there always exists a polarization $\Lambda_{SO}$ which is a $\mathbb{Z}_2$ subgroup of $\mathcal{L} = \widehat{Z}_{Spin(2n)}$. For $\mathfrak{g} = \mathfrak{su}(n)$, a polarization exists only if $n = m^2$.

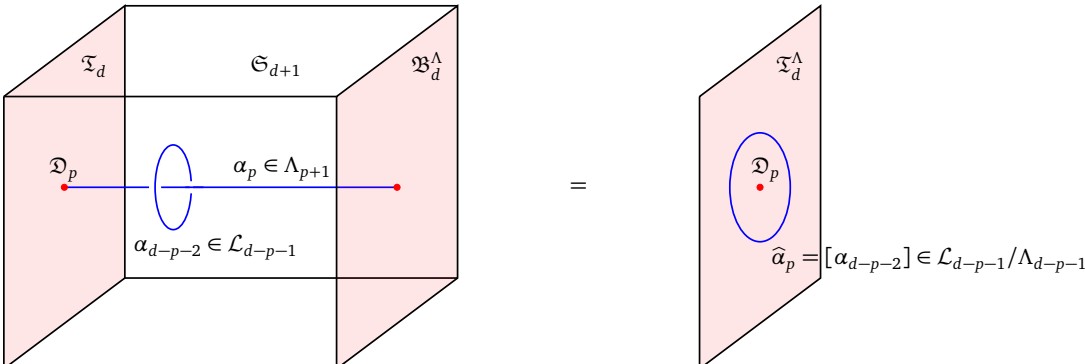

Figure 6: The action of a $p$-form symmetry $\widehat{\alpha}_p$ on a genuine defect $\mathfrak{D}_p$ of the absolute theory $\mathfrak{T}_d^\Lambda$ is obtained by braiding a topological defect $\alpha_{d-p-2} \in \mathcal{L}_{d-p-1}$ with a topological defect $\alpha_p \in \mathcal{L}_{p+1}$ of the TQFT $\mathfrak{S}_{d+1}$, where $\mathfrak{D}_p$ arises at the end of $\alpha_p$ and $\widehat{\alpha}_p$ descends from $\alpha_{d-p-2}$.

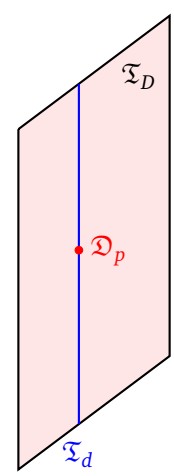

Figure 7: A sub-defect $\mathfrak{D}_p$ living inside a relative defect $\mathfrak{T}_d$ of a theory $\mathfrak{T}_D$.

## 2.3 Relative Defects in Absolute Theories

In an analogous way, we define relative defects as those defects that carry mutually non-local sub-defects. The sub-defects are constrained to live in the worldvolume of the relative defect. See figure 7.

Let us begin by considering relative defects in absolute theories. The structure of such a relative defect is very similar to that of a relative theory. In fact, we can apply the whole machinery discussed in previous subsections, but now regarding $\mathfrak{T}_d$ not as a relative $d$-dimensional theory, but instead a relative $d$-dimensional defect inside an absolute $D$-dimensional theory $\mathfrak{T}_D$, where $D > d$.

$\mathfrak{S}_{d+1}$ is now a TQFT which is attached to the absolute theory $\mathfrak{T}_D$ along the relative defect $\mathfrak{T}_d$. See figure 8. $\mathcal{L}$ again describes higher-form symmetries of this TQFT. The defects $\mathfrak{D}_p$ are sub-defects living inside the relative defect $\mathfrak{T}_d$, and $\mathcal{L}$ captures equivalence classes and non-locality of (some of) these sub-defects. We refer to $\mathcal{L}$ as the *defect group of the relative defect* $\mathfrak{T}_d$.

Choosing a topological boundary condition[8] $\mathfrak{B}_d^\Lambda$, associated to a polarization $\Lambda$ of $\mathcal{L}$, of the

---

[8]Whenever the topological boundary condition is actually a topological interface converting the TQFT $\mathfrak{S}_{d+1}$ to an SPT phase, the SPT phase captures the anomaly of the higher-form symmetries localized along the absolute defect discussed in what follows.

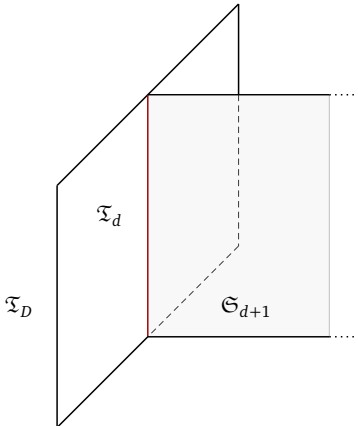

Figure 8: Relative defect $\mathfrak{T}_d$ in absolute theory $\mathfrak{T}_D$ with TQFT $\mathfrak{S}_{d+1}$ attached.

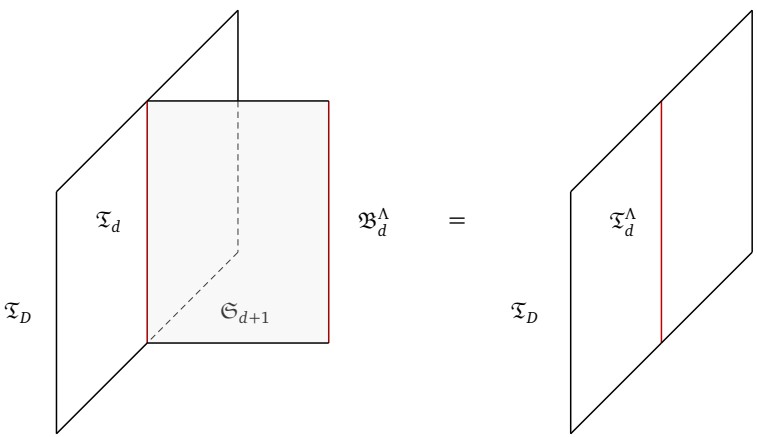

Figure 9: Absolute defect $\mathfrak{T}_d^\Lambda$ in absolute theory $\mathfrak{T}_D$. The absolute defect follows from the relative defect $\mathfrak{T}_d$ by choosing a topological boundary condition $\mathfrak{B}_d^\Lambda$ for the defect TQFT $\mathfrak{S}_{d+1}$.

TQFT $\mathfrak{S}_{d+1}$ leads to an absolute defect $\mathfrak{T}_d^\Lambda$ of the absolute theory $\mathfrak{T}_D$. See figure 9. The group $\mathcal{L}/\Lambda = \widehat{\Lambda}$ describes invertible topological sub-defects of the absolute defect $\mathfrak{T}_d^\Lambda$. These topological sub-defects give rise to *higher-form symmetries of the absolute defect* $\mathfrak{T}_d^\Lambda$. The background fields for such higher-form symmetries live along the worldvolume of $\mathfrak{T}_d^\Lambda$.

The sub-defects lying in equivalence classes in $\Lambda$ become genuine sub-defects of the absolute defect $\mathfrak{T}_d^\Lambda$. These genuine sub-defects are charged objects under the higher-form symmetries $\mathcal{L}/\Lambda$ of the absolute defect $\mathfrak{T}_d^\Lambda$. On the other hand, the sub-defects lying in equivalence classes in $\mathcal{L} - \Lambda$ become non-genuine sub-defects of the absolute defect $\mathfrak{T}_d^\Lambda$. The sub-defects belonging to $\mathcal{L} - \Lambda$ arise at the ends of topological sub-defects valued in $\mathcal{L}/\Lambda = \widehat{\Lambda}$ associated to higher-form symmetries of the absolute defect $\mathfrak{T}_d^\Lambda$.

## 2.4 Relative Defects in Relative Theories

Now let us consider relative defects in relative theories, which is the main topic of this paper. The structure of such a relative defect (in a relative theory) is much more interesting than the structure of a relative defect in an absolute theory, as now the defect group of the relative defect interacts non-trivially with the defect group of relative theory itself.

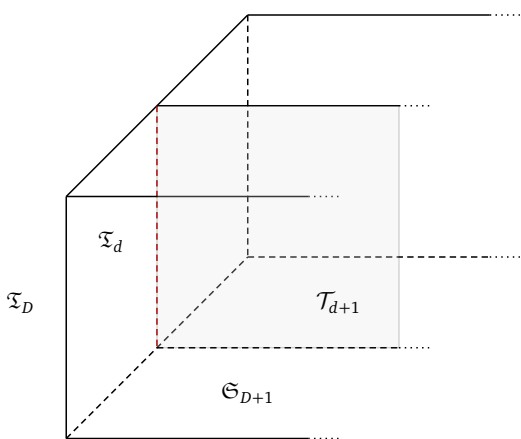

Figure 10: Relative defect $\mathfrak{T}_d$ inside a relative theory $\mathfrak{T}_D$. The relative theory $\mathfrak{T}_D$ is attached to a TQFT $\mathfrak{S}_{D+1}$, while the relative defect $\mathfrak{T}_d$ is attached to a topological defect $\mathfrak{S}_{d+1}$ of the TQFT $\mathfrak{S}_{D+1}$.

**Topological defect associated to relative defect.** We consider a $d$-dimensional relative defect $\mathfrak{T}_d$ in a $D$-dimensional relative theory $\mathfrak{T}_D$ for $d < D$. The relative theory is attached to a $(D+1)$-dimensional TQFT $\mathfrak{S}_{D+1}$, and the relative defect is attached to a non-invertible $(d+1)$-dimensional topological defect $\mathfrak{S}_{d+1}$ of the TQFT $\mathfrak{S}_{D+1}$. See figure 10.

Let $\mathcal{L}_D$ be the defect group of the TQFT $\mathfrak{S}_{D+1}$. The $(p+1)$-dimensional topological defects of $\mathfrak{S}_{D+1}$ valued in $\mathcal{L}_{D,p+1}$ can end on the topological defect $\mathfrak{S}_{d+1}$ as long as $p \le d$. Thus, a general invertible $p$-dimensional topological sub-defect of the topological defect $\mathfrak{S}_{d+1}$ comes attached to an invertible $(p+1)$-dimensional topological defect of the TQFT $\mathfrak{S}_{D+1}$. See figure 11. Let $\mathcal{L}_{d,p}$ be the group of invertible $p$-dimensional sub-defects of $\mathfrak{S}_{d+1}$. Then, we have a map

$$\pi_p : \mathcal{L}_{d,p} \to \mathcal{L}_{D,p+1} \tag{14}$$

that maps a $p$-dimensional sub-defect of $\mathfrak{S}_{d+1}$ to the bulk $(p+1)$-dimensional topological defect to which the $p$-dimensional sub-defect is attached to. The kernel of this projection map

$$\mathcal{L}_{d,p}^0 = \ker(\pi_p) \tag{15}$$

describes invertible $p$-dimensional sub-defects of $\mathfrak{S}_{d+1}$ that can exist independently without the presence of a $(p+1)$-dimensional bulk topological defect.

Another interplay between the groups $\mathcal{L}_{d,p}$ and $\mathcal{L}_D$ is as follows. Let $S^{D-d-1}$ be a $(D-d-1)$-dimensional sphere that links the worldvolume of $\mathfrak{S}_{d+1}$ in a small neighborhood of the worldvolume. Wrapping a $(D-p-1)$-dimensional topological defect valued in $\mathcal{L}_{D,D-p-1}$ along $S^{D-d-1}$ and squeezing the sphere $S^{D-d-1}$ onto the worldvolume of $\mathfrak{S}_{d+1}$, leaves behind a $(d-p)$-dimensional topological sub-defect of $\mathfrak{S}_{d+1}$ valued in $\mathcal{L}_{d,d-p}^0$. See figure 12. That is, we have a map

$$s_{d-p} : \mathcal{L}_{D,D-p-1} \to \mathcal{L}_{d,d-p}^0. \tag{16}$$

Let us define

$$\mathcal{L}_{d,p}^T := \mathcal{L}_{d,p}^0 / \mathrm{Im}(s_p), \tag{17}$$

which we refer to as the *trapped* part of $\mathcal{L}_{d,p}$. This is because $\mathcal{L}_{d,p}^T$ roughly describes the topological $p$-dimensional invertible sub-defects of $\mathfrak{S}_{d+1}$ that neither arise at the ends of topological defects of the bulk theory $\mathfrak{S}_{D+1}$, nor can be obtained from them via the squeezing procedure discussed above.

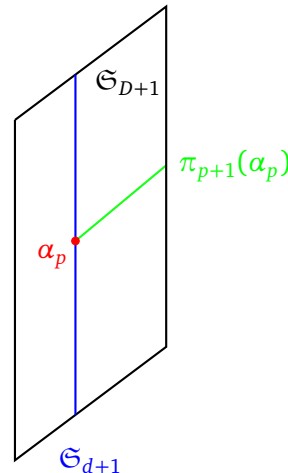

Figure 11: A topological sub-defect $\alpha_p \in \mathcal{L}_{d,p+1}$ of the topological defect $\mathfrak{S}_{d+1}$ arises at the end of a topological defect $\pi_{p+1}(\alpha_p) \in \mathcal{L}_{D,p+2}$ of the TQFT $\mathfrak{S}_{D+1}$.

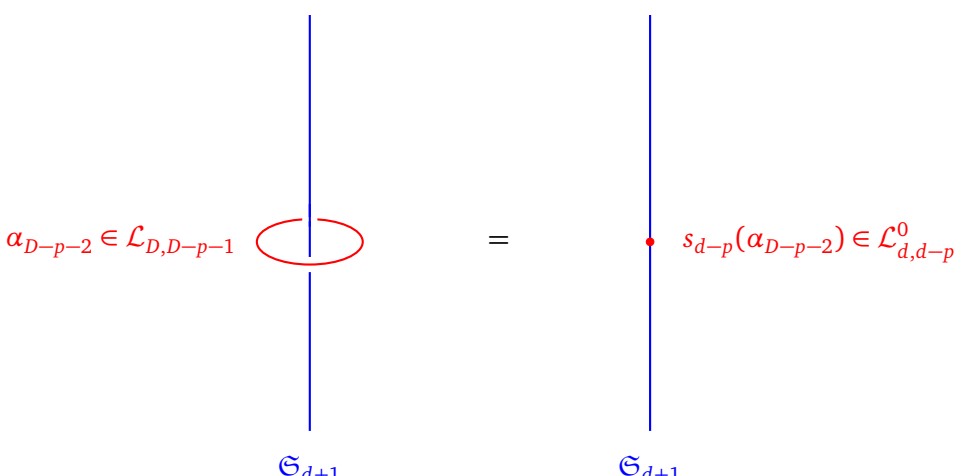

Figure 12: Wrapping a topological defect $\alpha_{D-p-2} \in \mathcal{L}_{D,D-p-1}$ of the TQFT $\mathfrak{S}_{D+1}$ along a sphere $S^{D-d-1}$ surrounding the topological defect $\mathfrak{S}_{d+1}$, and squeezing it, gives rise to a sub-defect $s_{d-p}(\alpha_{D-p-2}) \in \mathcal{L}_{d,d-p}$ of the topological defect $\mathfrak{S}_{d+1}$. Since the sub-defect is not attached to any other topological defect, it is actually valued in $\mathcal{L}^0_{d,d-p}$.

We have non-trivial braidings between elements of $\mathcal{L}_{d,p}$ and elements of $\mathcal{L}_{d,d-p}$ that imply

$$\mathcal{L}_{d,d-p} \cong \widehat{\mathcal{L}}_{d,p}, \tag{18}$$

where $\widehat{\mathcal{L}}_{d,p}$ is the Pontryagin dual of $\mathcal{L}_{d,p}$ for $1 \le p \le d-1$. These braidings define a pairing $\langle \cdot, \cdot \rangle_{\mathcal{L}_d}$ on

$$\mathcal{L}_d := \prod_{p=1}^{d-1} \mathcal{L}_{d,p} \tag{19}$$

that needs to be consistent with the pairing $\langle \cdot, \cdot \rangle_{\mathcal{L}_D}$ on $\mathcal{L}_D$, such that

$$\langle \alpha_p, \alpha_{d-p} \rangle_{\mathcal{L}_d} = \langle \pi_p(\alpha_p), \widetilde{\alpha}_{d-p} \rangle_{\mathcal{L}_D}, \tag{20}$$

for all $\alpha_p \in \mathcal{L}_{d,p}$ and $\alpha_{d-p} \in \mathcal{L}_{d,d-p}$, where $\widetilde{\alpha}_{d-p}$ is any element of $\mathcal{L}_{D,D-p-1}$ such that $s_{d-p}(\widetilde{\alpha}_{d-p}) = \alpha_{d-p}$.

**Non-locality of sub-defects.** We call $\mathcal{L}_d$ the *defect group associated to the relative defect* $\mathfrak{T}_d$. We have seen above that this defect group has a non-trivial interplay with the defect group $\mathcal{L}_D$ of the relative theory $\mathfrak{T}_D$. $\mathcal{L}_{d,p+1}$ captures (some of) the equivalence classes of $p$-dimensional sub-defects of the relative defect $\mathfrak{T}_d$. The $p$-dimensional sub-defects of the relative defect $\mathfrak{T}_d$ in general arise at the ends of the $(p+1)$-dimensional defects of the relative theory $\mathfrak{T}_D$. Consider such a $p$-dimensional sub-defect $\mathfrak{D}_{d,p}$ of the relative defect $\mathfrak{T}_d$ that arises at the end of a $(p+1)$-dimensional defect $\mathfrak{D}_{D,p+1}$ of the relative theory $\mathfrak{T}_D$. Moreover, let the defect $\mathfrak{D}_{D,p+1}$ lie in an equivalence class $\alpha_{D,p+1} \in \mathcal{L}_{D,p+2}$, and the sub-defect $\mathfrak{D}_{d,p}$ lie in an equivalence class $\alpha_{d,p} \in \mathcal{L}_{d,p+1}$. Then, we have the relation

$$\alpha_{D,p+1} = \pi_{p+1}(\alpha_{d,p}). \tag{21}$$

The pairing on $\mathcal{L}_d$ captures non-locality of these sub-defects as they are moved around within the worldvolume of the relative defect $\mathfrak{T}_d$.

**Polarizations.** Now suppose we have picked an absolute theory $\mathfrak{T}_D^{\Lambda_D}$ from the relative theory $\mathfrak{T}_D$ by choosing a topological boundary condition $\mathfrak{B}_D^{\Lambda_D}$ of the TQFT $\mathfrak{S}_{D+1}$, with the associated polarization being

$$\Lambda_D = \prod_{p=1}^{D-1} \Lambda_{D,p} \subset \mathcal{L}_D. \tag{22}$$

We want to understand the possible polarizations associated to the absolute defects that can be obtained out of the relative defect $\mathfrak{T}_d$ with the above fixed choice of absolute theory $\mathfrak{T}_D^{\Lambda_D}$. Abstractly, we need a $d$-dimensional topological sub-defect[9] $\mathfrak{B}_d^{\Lambda_d}$ of the topological boundary condition $\mathfrak{B}_D^{\Lambda_D}$ on which one can end the topological defect $\mathfrak{S}_{d+1}$ of the TQFT $\mathfrak{S}_{D+1}$. See figure 13. The essential data of $\mathfrak{B}_d^{\Lambda_d}$, as far as invertible sub-defects of $\mathfrak{S}_{d+1}$ are concerned, is a polarization

$$\Lambda_d = \prod_{p=1}^{d-1} \Lambda_{d,p} \subset \mathcal{L}_d, \tag{23}$$

---

[9]Just like $\mathfrak{B}_D^{\Lambda_D}$ is a topological interface with a $(D+1)$-dimensional SPT phase in general, the topological sub-defect $\mathfrak{B}_d^{\Lambda_d}$ is also in general a topological interface between the topological defect $\mathfrak{S}_{d+1}$ and a topological defect inside the SPT phase associated to $\mathfrak{B}_D^{\Lambda_D}$. Even when the $(D+1)$-dimensional SPT phase associated to $\mathfrak{B}_D^{\Lambda_D}$ is trivial, $\mathfrak{B}_d^{\Lambda_d}$ is still in general a topological interface, which now converts $\mathfrak{S}_{d+1}$ into a $(d+1)$-dimensional SPT phase. Again, for brevity, we will ignore the interface nature of $\mathfrak{B}_d^{\Lambda_d}$.

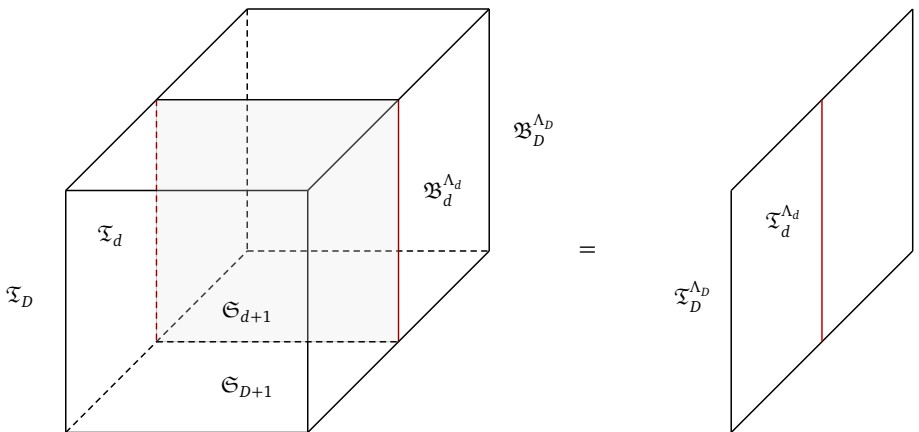

Figure 13: Choosing a topological sub-defect $\mathfrak{B}_d^{\Lambda_d}$ of the boundary condition $\mathfrak{B}_D^{\Lambda_D}$ of the TQFT $\mathfrak{S}_{D+1}$ constructs an absolute defect $\mathfrak{T}_d^{\Lambda_d}$ of the absolute theory $\mathfrak{T}_D^{\Lambda_D}$.

where $\Lambda_{d,p} \subseteq \mathcal{L}_{d,p}$ describes the subgroup of $p$-dimensional invertible sub-defects of $\mathfrak{S}_{d+1}$ that can end on $\mathfrak{B}_d^{\Lambda_d}$. A consistency condition is that

$$\pi_p(\Lambda_{d,p}) \subseteq \Lambda_{D,p+1}. \tag{24}$$

That is, if a $p$-dimensional topological sub-defect $\alpha_p \in \mathcal{L}_{d,p}$ of $\mathfrak{S}_{d+1}$ can end, then the bulk topological defect $\pi_p(\alpha_p) \in \mathcal{L}_{D,p+1}$ it is attached to ends as well. Another consistency condition is that

$$s_{d-p}(\Lambda_{D,D-p-1}) \subseteq \Lambda_{d,d-p}. \tag{25}$$

That is, if a topological defect $\alpha_{D-p-1} \in \mathcal{L}_{D,D-p-1}$ can end, then the topological sub-defect $s_{d-p}(\alpha_{D-p-1}) \in \mathcal{L}_{d,d-p}$ (obtained by squeezing $\alpha_{D-p-1}$) must end as well. Choosing such a $\mathfrak{B}_d^{\Lambda_d}$ gives us an absolute defect $\mathfrak{T}_d^{\Lambda_d}$ living inside the absolute theory $\mathfrak{T}_D^{\Lambda_D}$.

**Higher-form symmetries.**  As discussed earlier, $\mathcal{L}_D/\Lambda_D = \widehat{\Lambda}_D$ captures higher-form symmetries of the absolute theory $\mathfrak{T}_D^{\Lambda_D}$. Similarly, $\mathcal{L}_d/\Lambda_d = \widehat{\Lambda}_d$ captures 'higher-form symmetries of the absolute defect' $\mathfrak{T}_d^{\Lambda_d}$. The $p$-form symmetry group $\Gamma^{(p)}\left[\mathfrak{T}_d^{\Lambda_d}\right]$ of the absolute defect $\mathfrak{T}_d^{\Lambda_d}$ is

$$\Gamma^{(p)}\left[\mathfrak{T}_d^{\Lambda_d}\right] = \mathcal{L}_{d,d-p-1}/\Lambda_{d,d-p-1} \cong \widehat{\Lambda}_{d,p+1}. \tag{26}$$

The background field $B_{d,p+1} \in C^{p+1}(\Sigma_d, \widehat{\Lambda}_{d,p+1})$ of the $p$-form symmetry $\Gamma^{(p)}\left[\mathfrak{T}_d^{\Lambda_d}\right]$ of the absolute defect $\mathfrak{T}_d^{\Lambda_d}$ is a $(p+1)$-cochain valued in $\widehat{\Lambda}_{d,p+1}$ on the $d$-dimensional worldvolume $\Sigma_d$ of the absolute defect $\mathfrak{T}_d^{\Lambda_d}$.

However, these background fields interact non-trivially with the background fields for the higher-form symmetries $\widehat{\Lambda}_D$ of the bulk absolute theory $\mathfrak{T}_D^{\Lambda_D}$. Let the background field for the bulk $p$-form symmetry be denoted by $B_{D,p+1} \in C^{p+1}(\Sigma_D, \widehat{\Lambda}_{D,p+1})$, which is a $(p+1)$-cochain valued in $\widehat{\Lambda}_{D,p+1}$ on the $D$-dimensional spacetime manifold $\Sigma_D$ where the absolute theory $\mathfrak{T}_D^{\Lambda_D}$ lives. For example, the fact that topological defects valued in $\Lambda_D$ can end on $\mathfrak{T}_d^{\Lambda_d}$ to give rise to topological defects valued in $\Lambda_d$ imposes the following relationship on the background fields

$$\delta B_{D,D-d+p} = \pi_{d-p-1}^{\vee}(B_{d,p+1}) \wedge \delta_{D-d}^{\Sigma_d}, \tag{27}$$

where $\delta_{D-d}^{\Sigma_d}$ is the cochain Poincaré dual to $\Sigma_d$ inside $\Sigma_D$, and

$$\pi_{d-p-1}^{\vee} : \widehat{\Lambda}_{d,p+1} \to \widehat{\Lambda}_{D,D-d+p} \tag{28}$$

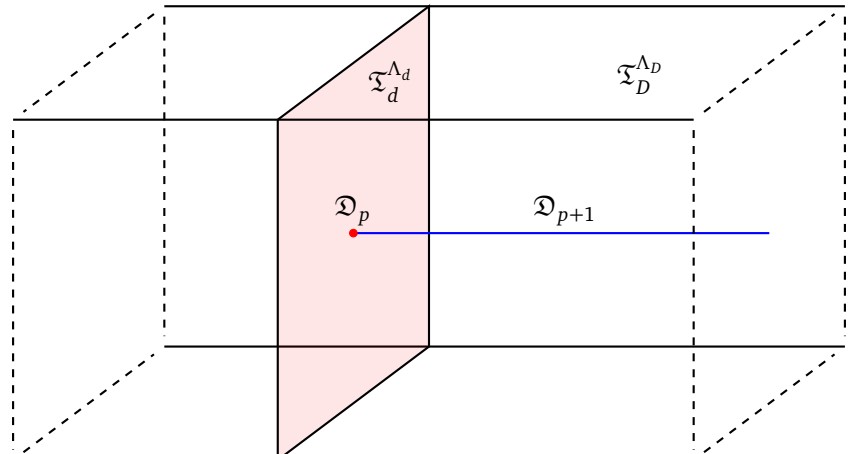

Figure 14: A "genuine" sub-defect $\mathfrak{D}_p$ of the absolute defect $\mathfrak{T}_d^{\Lambda_d}$ arising at the end of a genuine defect $\mathfrak{D}_{p+1}$ of the absolute theory $\mathfrak{T}_D^{\Lambda_D}$. Such a configuration occurs when the equivalence class $[\mathfrak{D}_p]$ of $\mathfrak{D}_p$ is in $\Lambda_{d,p+1}$ (which implies $[\mathfrak{D}_{p+1}] = \pi_{p+1}([\mathfrak{D}_p]) \in \Lambda_{D,p+2}$).

descends from the map $\pi_{d-p-1} : \mathcal{L}_{d,d-p-1} \rightarrow \mathcal{L}_{D,d-p}$ along with the use of isomorphisms $\mathcal{L}_{d,d-p-1} \cong \widehat{\Lambda}_{d,p+1}$ and $\mathcal{L}_{D,d-p} \cong \widehat{\Lambda}_{D,D-d+p}$. The reader can check that (28) is a well-defined map constructed this way, thanks to the condition (24).

**Genuine and non-genuine defects.** Let us now look at the fate of the non-topological defects after choosing the polarizations $\Lambda_D$ and $\Lambda_d$. These defects provide charged objects under the higher-form symmetries discussed above. Consider a sub-defect $\mathfrak{D}_p$ of the relative defect $\mathfrak{T}_d$ lying in an equivalence class in $\Lambda_{d,p+1}$. Let $\mathfrak{D}_p$ be attached to a defect $\mathfrak{D}_{p+1}$ of the relative theory $\mathfrak{T}_D$. Then, (24) implies that $\mathfrak{D}_{p+1}$ is a genuine defect of the absolute theory $\mathfrak{T}_D^{\Lambda_D}$. Moreover, $\mathfrak{D}_p$ is a sub-defect of the absolute defect $\mathfrak{T}_d^{\Lambda_d}$ arising at the end of $\mathfrak{D}_{p+1}$ on $\mathfrak{T}_d^{\Lambda_d}$, without any additional topological sub-defects of $\mathfrak{T}_d^{\Lambda_d}$ attached to the defect-subdefect configuration formed by $\mathfrak{D}_{p+1}$ and $\mathfrak{D}_p$. See figure 14. Let $\alpha_p \in \Lambda_{d,p+1}$ be the equivalence class associated to $\mathfrak{D}_p$. The transformation of $\mathfrak{D}_p$ under a $p$-form symmetry element $\widehat{\alpha}_p \in \Gamma^{(p)}\left[\mathfrak{T}_d^{\Lambda_d}\right] \cong \widehat{\Lambda}_{d,p+1}$ living on the absolute defect $\mathfrak{T}_d^{\Lambda_d}$ is given by evaluating

$$\widehat{\alpha}_p(\alpha_p) \in U(1). \tag{29}$$

Now consider the situation when the sub-defect $\mathfrak{D}_p$ lies in an equivalence class $\alpha_p$ in $\mathcal{L}_{d,p+1} - \Lambda_{d,p+1}$ while the equivalence class $\pi_{p+1}(\alpha_p)$ of the defect $\mathfrak{D}_{p+1}$ (that $\mathfrak{D}_p$ is attached to) lies in $\Lambda_{D,p+2}$. This means that while $\mathfrak{D}_{p+1}$ is a genuine defect of the absolute theory $\mathfrak{T}_D^{\Lambda_D}$, the sub-defect $\mathfrak{D}_p$ of the absolute defect $\mathfrak{T}_d^{\Lambda_d}$ arising at the end of $\mathfrak{D}_{p+1}$ is not completely genuine, in the sense that it is attached to an additional $(p+1)$-dimensional topological sub-defect of $\mathfrak{T}_d^{\Lambda_d}$. See figure 15. The $(p+1)$-dimensional topological sub-defect is specified by the element $[\alpha_p] \in \mathcal{L}_{d,p+1}/\Lambda_{d,p+1} \cong \widehat{\Lambda}_{d,d-p-1}$.

Finally, consider the situation when the equivalence class $\alpha_p$ lies in $\mathcal{L}_{d,p+1} - \Lambda_{d,p+1}$ and the equivalence class $\pi_{p+1}(\alpha_p)$ lies in $\mathcal{L}_{D,p+2} - \Lambda_{D,p+2}$. In this case, the defect $\mathfrak{D}_{p+1}$ is a non-genuine defect of the absolute defect $\mathfrak{T}_d^{\Lambda_d}$ attached to the $(p+2)$-dimensional topological defect of the absolute theory $\mathfrak{T}_D^{\Lambda_D}$ specified by the element $[\pi_{p+1}(\alpha_p)] \in \mathcal{L}_{D,p+2}/\Lambda_{D,p+2} \cong \widehat{\Lambda}_{D,D-p-2}$. Similarly, the sub-defect $\mathfrak{D}_p$ of the absolute defect $\mathfrak{T}_d^{\Lambda_d}$ arising at the end of $\mathfrak{D}_{p+1}$ is also not

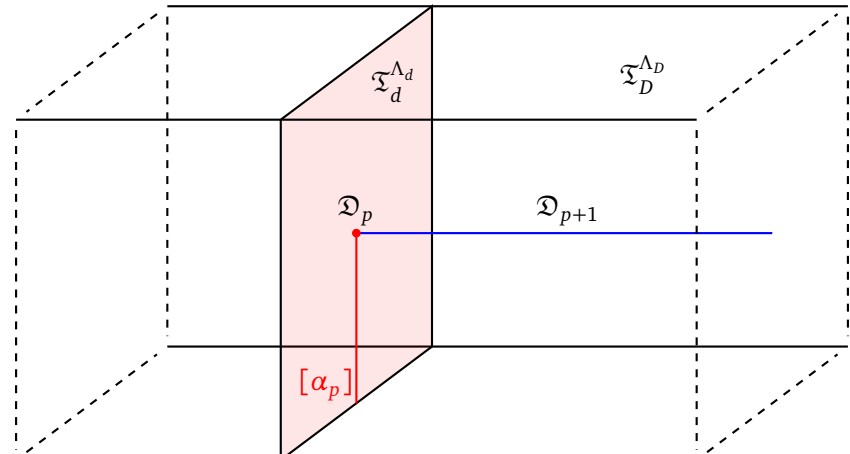

Figure 15: A "non-genuine" sub-defect $\mathfrak{D}_p$ of the absolute defect $\mathfrak{T}_d^{\Lambda_d}$ arising at the end of a genuine defect $\mathfrak{D}_{p+1}$ of the absolute theory $\mathfrak{T}_D^{\Lambda_D}$. Such a configuration occurs when the equivalence class $\alpha_p = [\mathfrak{D}_p] \in \mathcal{L}_{d,p+1} - \Lambda_{d,p+1}$ and $[\mathfrak{D}_{p+1}] = \pi_{p+1}(\alpha_p) \in \Lambda_{D,p+2}$. In such a situation, the non-topological sub-defect $\mathfrak{D}_p$ is further attached to a topological sub-defect $[\alpha_p] \in \mathcal{L}_{d,p+1}/\Lambda_{d,p+1}$ of the absolute defect $\mathfrak{T}_d^{\Lambda_d}$.

genuine, being attached to the $(p+1)$-dimensional topological sub-defect of the absolute defect $\mathfrak{T}_d^{\Lambda_d}$ specified by the element $[\alpha_p] \in \mathcal{L}_{d,p+1}/\Lambda_{d,p+1} \cong \widehat{\Lambda}_{d,d-p-1}$. Moreover, the $(p+1)$-dimensional topological sub-defect $[\alpha_p]$ lives at the intersection of the $(p+2)$-dimensional topological defect $[\pi_{p+1}(\alpha_p)]$ and the absolute defect $\mathfrak{T}_d^{\Lambda_d}$. See figure 16.

**Twisted sector relative defects.** Assume that the TQFT $\mathfrak{S}_{D+1}$ has a 0-form symmetry group $\mathcal{L}_{D,D}$. Let the ends of the topological defects valued in $\mathcal{L}_{D,D}$ at the location of the relative theory $\mathfrak{T}_D$ give rise to topological defects of the relative theory $\mathfrak{T}_D$ that are also valued in $\mathcal{L}_{D,D}$. Then, we can consider codimension-two relative defects $\mathfrak{T}_{d=D-2}$ of the relative theory $\mathfrak{T}_D$ that arise at the end of a topological codimension-one defect $\alpha \in \mathcal{L}_{D,D}$ of $\mathfrak{T}_D$. For such a relative defect, the non-invertible topological defect $\mathfrak{S}_{d+1=D-1}$ also arises at the end of the codimension-one topological defect $\alpha \in \mathcal{L}_{D,D}$ of the TQFT $\mathfrak{S}_{D+1}$.

The (not necessarily topological) defects that arise at the ends of $p$-form symmetry generating topological defects are called *twisted sector defects* for the $p$-form symmetry. In this language, the above discussed codimension-two relative defects and the topological defect $\mathfrak{S}_{d+1=D-1}$ are twisted sector defects for the 0-form symmetry $\mathcal{L}_{D,D}$.

The defect group of such a twisted sector relative defect has almost the same structure as for the untwisted relative defects discussed above, though there are slight modifications due to the action of $\mathcal{L}_{D,D}$ on $\mathcal{L}_{D,p}$ for $p < D$. The co-domain of the map $\pi_p$ is the group obtained by modding out $\mathcal{L}_{D,p+1}$ by the action of $\alpha \in \mathcal{L}_{D,D}$. On the other hand, the domain of the map $s_{d-p}$ is the subgroup of $\mathcal{L}_{D,D-p-1}$ left invariant by the action of $\alpha \in \mathcal{L}_{D,D}$.

## 2.5 Relative Defects in 6d $(2,0)$ Theories

In this paper we study codimension-two defects of 6d $\mathcal{N} = (2,0)$ theories, and find that in general such defects are relative defects. Since 6d $\mathcal{N} = (2,0)$ theories are relative theories, we obtain examples of the case of relative defects inside relative theories discussed in the

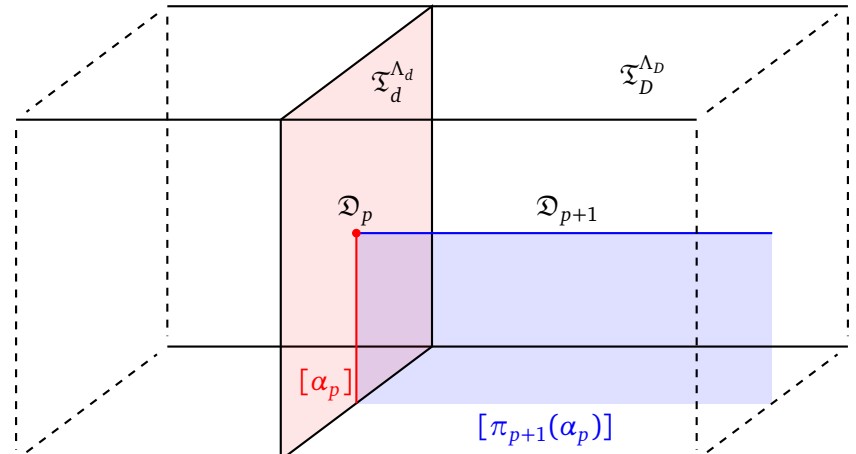

Figure 16: A "non-genuine" sub-defect $\mathfrak{D}_p$ of the absolute defect $\mathfrak{T}_d^{\Lambda_d}$ arising at the end of a non-genuine defect $\mathfrak{D}_{p+1}$ of the absolute theory $\mathfrak{T}_D^{\Lambda_D}$. Such a configuration occurs when the equivalence class $\alpha_p = [\mathfrak{D}_p] \in \mathcal{L}_{d,p+1} - \Lambda_{d,p+1}$ and $[\mathfrak{D}_{p+1}] = \pi_{p+1}(\alpha_p) \in \mathcal{L}_{D,p+2} - \Lambda_{D,p+2}$. In such a situation, the non-topological sub-defect $\mathfrak{D}_p$ is attached to a topological sub-defect $[\alpha_p] \in \mathcal{L}_{d,p+1}/\Lambda_{d,p+1}$ of the absolute defect $\mathfrak{T}_d^{\Lambda_d}$, and the non-topological defect $\mathfrak{D}_{p+1}$ is attached to a topological defect $[\pi_{p+1}(\alpha_p)] \in \mathcal{L}_{D,p+2}/\Lambda_{D,p+2}$ of the absolute theory $\mathfrak{T}_D^{\Lambda_D}$. $[\pi_{p+1}(\alpha_p)]$ ends on $\mathfrak{T}_d^{\Lambda_d}$ and $[\alpha_p]$ arises at this end.

previous subsection. This is a specialization of the general analysis so far to the case of

$$D = 6, \qquad d = 4. \tag{30}$$

Below, we will keep the subscripts $d$ and $D$ for ease of comparison with the earlier analysis. Recall that a 6d $(2,0)$ theory of type $\mathfrak{g}$ has defect group

$$\mathcal{L}_D = \mathcal{L}_{D,3} = \widehat{Z}_G. \tag{31}$$

A codimension-two defect (meaning $d = 4$) of the 6d $(2,0)$ theory has in general a defect group

$$\mathcal{L}_d = \mathcal{L}_{d,1} \times \mathcal{L}_{d,2} \times \mathcal{L}_{d,3}. \tag{32}$$

In this paper, we study only the $\mathcal{L}_{d,2}$ part of the defect group, while leaving the study of $\mathcal{L}_{d,1} \times \mathcal{L}_{d,3}$ part of the defect group to future works. In fact, $\mathcal{L}_{d,2}$ is the more interesting part of $\mathcal{L}_d$ as it can interact with the defect group $\mathcal{L}_D = \mathcal{L}_{D,3}$ of the 6d $(2,0)$ theory itself. $\mathcal{L}_{d,2}$ captures equivalence classes of non-topological line defects in the relative codimension-two defect.

We find that $\mathcal{L}_{d,2}$ can be expressed as

$$\mathcal{L}_{d,2} = \mathcal{W} \times \mathcal{H}, \tag{33}$$

with the pairing on $\mathcal{L}_{d,2}$ being such that it provides an isomorphism $\mathcal{W} \to \widehat{\mathcal{H}}$. That is, we can write

$$\mathcal{W} \cong \widehat{\mathcal{H}}, \tag{34}$$

where $\widehat{\mathcal{H}}$ is the Pontryagin dual of $\mathcal{H}$. Moreover, we have

$$\mathcal{L}_{d,2}^0 = \mathcal{W} \times \mathcal{H}^T, \tag{35}$$

where $\mathcal{H}^T$ is a subgroup of $\mathcal{H}$ and $\mathcal{L}^0_{d,2}$ is the kernel of the map

$$\pi_2 : \mathcal{L}_{d,2} \to \mathcal{L}_{D,3} \tag{36}$$

describing which line defects of the codimension-two defect arise at the ends of surface defects of the bulk $6d$ $\mathcal{N} = (2,0)$ theory.

Dually, the image $\text{Im}(s_2)$ of the map

$$s_2 : \mathcal{L}_{D,3} \to \mathcal{L}_{d,2} \tag{37}$$

relating surface defects of the $(2,0)$ theory and line defects of the codimension-two defect via the squeezing procedure, is such that

$$\text{Im}(s_2) \subseteq \mathcal{W} \subset \mathcal{L}_{d,2} . \tag{38}$$

Let us define

$$\mathcal{W}^T := \mathcal{W}/\text{Im}(s_2) . \tag{39}$$

This allows us to express the trapped part $\mathcal{L}^T_{d,2}$ of $\mathcal{L}_{d,2}$ as

$$\mathcal{L}^T_{d,2} = \mathcal{L}^0_{d,2}/\text{Im}(s_2) = \mathcal{W}^T \times \mathcal{H}^T . \tag{40}$$

The pairing on $\mathcal{L}_{d,2}$ descends to a pairing on $\mathcal{L}^T_{d,2}$ which is such that it provides an isomorphism

$$\mathcal{W}^T \cong \widehat{\mathcal{H}^T} , \tag{41}$$

where $\widehat{\mathcal{H}^T}$ is the Pontryagin dual of $\mathcal{H}^T$. We also discuss *twisted* codimension-two defects that arise at the ends of topological codimension-one defects implementing the outer-automorphism 0-form symmetries, and they have a similar structure as discussed above.

## 2.6 Compactification: Relative Theories from Relative Defects

Suppose we compactify a relative theory $\mathfrak{T}_D$ on a $(D-d)$-dimensional compactification manifold $\Sigma_{D-d}$. Moreover, let us place $d$-dimensional relative defects $\mathfrak{T}^i_d$ of various types $i$ at points $p_i$ on $\Sigma_{D-d}$. Such a compactification yields a relative $d$-dimensional theory $\widetilde{\mathfrak{T}}_d$.

The defect group $\widetilde{\mathcal{L}}_d$ of the resulting theory $\widetilde{\mathfrak{T}}_d$ obtains contributions not only from the defect group $\mathcal{L}_D$ of the parent theory $\mathfrak{T}_D$, but also from the defect group $\mathcal{L}^i_d$ of the relative defects $\mathfrak{T}^i_d$ employed in the compactification. In particular, the component $\widetilde{\mathcal{L}}_{d,p}$ will obtain contributions not only from $\mathcal{L}_{D,q}$ for $q \geq p$, but also from $\mathcal{L}^i_{d,p}$.

One of main topics of study in this paper is to study a class of these kinds of compactifications. For us $\mathfrak{T}_D$ is a $6d$ $\mathcal{N} = (2,0)$ theory, which is compactified on a Riemann surface $\Sigma_2$. The relative defects $\mathfrak{T}^i_d$ are codimension-two defects of the $6d$ $\mathcal{N} = (2,0)$ theory. Since these relative defects are inserted at points on the Riemann surface, they are also referred to as punctures on the Riemann surface. The resulting theory $\widetilde{\mathfrak{T}}_d$ is a $4d$ $\mathcal{N} = 2$ Class S theory.

We are interested in computing the component $\widetilde{\mathcal{L}}_{d,2}$ of the defect group $\widetilde{\mathcal{L}}_d$ of the Class S theory $\widetilde{\mathfrak{T}}_d$. This component characterizes equivalence classes of line defects in the Class S theory. It receives contributions from the surface defects of the $(2,0)$ theory valued in $\mathcal{L}_D = \mathcal{L}_{D,3}$ and compactified along 1-cycles of the Riemann surface $\Sigma_2$. These kinds of contributions have been studied extensively in the past literature [1, 31, 73, 74].

However, as discussed above, there are also contributions from the line defects of the relative defects valued in $\mathcal{L}^i_{d,2}$ that we determine in this paper.

# 3 Evidence for Trapped 1-Form Symmetries

There is much evidence for the existence of additional sources of 1-form symmetries in Class S theories, coming from the punctures used in the Class S construction. We will present two: Type IIB constructions of 4d $\mathcal{N} = 2$ SCFTs using isolated hypersurface singularities (IHS) can have non-trivial 1-form symmetries. In turn, some of these theories have a realization as class S theories based on a sphere with a *single irregular puncture*. We will develop the dictionary between irregular punctures and IHS in section 6. This correspondence then very strongly suggests that irregular punctures have trapped 1-form symmetries! This is because a sphere has no non-trivial 1-cycles that can give rise to the usual type of 1-form symmetries in Class S theories that were discussed in [31].

The second motivation comes from the collision of regular punctures, to result in irregular ones. If the setup with regular punctures had 1-form symmetry, then again it is indicative that the resulting theory with irregular punctures should also have this symmetry, again pointing towards the existence of trapped 1-form symmetries.

## 3.1 The Defect Group of Type IIB on IHS

Type IIB compactification on canonical non-compact Calabi-Yau three-fold singularities $X$ engineers 4d $\mathcal{N} = 2$ SCFTs [78, 82]. A singularity $X$ that admit a resolution $\pi : \tilde{X} \to X$, which satisfy

$$K_{\tilde{X}} = \pi^* K_X + \sum_i a_i S_i, \tag{42}$$

with $a_i \geq 0$, where $K$ is the canonical class, and $S_i$ the exceptional divisors of the resolution, are called canonical singularities. When $a_i > 0$ the singularity is terminal, and if $a_i = 0$ for all $i$ it admits a crepant (i.e. Calabi-Yau) resolution.

The type of Calabi-Yau singularities that we will consider in this paper are so-called isolated hypersurface singularities (IHS): these are hypersurface equations in $\mathbb{C}^4$, of the type

$$P(x_1, x_2, x_3, x_4) = 0, \tag{43}$$

where $P$ is a polynomnial in $x_i$, satisfying various requirements, e.g. the existence of an isolated canonical singularity. IHSs can be classified and we refer to this as the Kreuzer-Skarke-Yau-Yu (KS-YY) classification [79, 85]. To specify an IHS we will use the notation in [55], which indicates the type and vanishing orders of $P$.

In [17, 29, 30, 36, 55] the deformation theory of the isolated hypersurface singularity (IHS) realization of such 4d $\mathcal{N} = 2$ theories was used to compute the line defects, and thereby 1-form symmetry. This is computed from the homology of the link

$$\mathcal{L}_X = \mathrm{Tor}\,\mathrm{ker}(h_2) \subseteq H_2(\partial X, \mathbb{Z}), \tag{44}$$

where

$$h_2 : H_2(\partial X, \mathbb{Z}) \to H_2(X, \mathbb{Z}) \tag{45}$$

is the map lifting a 2-cycle on the boundary $\partial X$ to a 2-cycle in the bulk $X$. The pairing on $\mathcal{L}_X$ descends from the linking pairing on $H_2(\partial X, \mathbb{Z})$.

The line defects are D3-branes wrapped on relative 3-cycles, (modulo screening by local operators realized as D3-branes on compact 3-cycles). This in turn can be identified with the homology $H_2(\partial X, \mathbb{Z})$ of the link, under the assumption that there is no torsion in the second homology of the bulk $X$. We will provide explicit examples of these defect groups in the following. Here we should note that they can be computed from the hypersurface singularity

using the code in [55]. Examples are AD theories of type $AD[G, G']$, with certain choices for $(G, G') = (A, D), (A, E), (D, D)$.

The non-triviality of the defect groups of lines for such IHS has an important, quite central, implication for class S construction: We will see in section 6 that many of these hypersurfaces have realizations as class S with a single irregular puncture $\mathcal{P}$ on a sphere. The results from the IHS realization imply that there are trapped line operators denoted by $\mathcal{L}_{\mathcal{P}}^{T}$ in such class S theories. In their Type IIB realization on the singularity $X_{\mathcal{P}}$ correspond to the relative homology groups (44):

$$\mathcal{L}_{\mathcal{P}}^{T} = \mathcal{L}_{X_{\mathcal{P}}} . \tag{46}$$

Thus, the trapped defect group $\mathcal{L}_{\mathcal{P}}^{T}$ of a puncture $\mathcal{P}$ lying in the IHS class can be computed from the data of the Calabi-Yau singularity $X_{\mathcal{P}}$ associated to $\mathcal{P}$ using the techniques of [30,55], which can be applied to any IHS. The relative homology groups are computed in these cases by using the deformation data of the IHS. We will not review this, but provide the results from these computations in the subsequent sections, which provide a prediction for the trapped part of the defect group for Class S theories.

## 3.2 Collisions of Regular Punctures

4d $\mathcal{N} = 2$ theories of class S constructed purely from regular twisted punctures can carry a non-trivial defect group of lines [31]. In many cases irregular punctures arise from operations on these theories, examples for such operations include the collision of punctures, gauging of flavor symmetries and (de)coupling of matter. The defect group associated with irregular punctures can therefore be bootstrapped from the results in [31] whenever the consequences of these operations on defect groups is understood. Here we will adopt this approach to show that irregular punctures must in general carry a nontrivial trapped defect group. It is enough to consider a simple set of Lagrangian theories to illustrate this.

As is well known, most 4d $\mathcal{N} = 2$ conformal linear quivers admit a class S description involving a collection of regular punctures on a sphere and in particular we will be concerned here with orthosymplectic quivers (alternating $\mathfrak{so}$ and $\mathfrak{usp}$ gauge algebras with bifundamental half-hypermultiplets in between), which are realized by compactifying the 6d $\mathcal{N} = (2, 0)$ theory of type $D$ on a sphere with both twisted and untwisted regular punctures. Qualitatively, the structure of such theories is as follows: there is a central part of the quiver in which the ranks of the gauge algebras stay constant and the quiver locally looks like $\ldots \mathfrak{usp}(2N-2) - \mathfrak{so}(2N) - \mathfrak{usp}(2N-2) \ldots$ This pattern is reproduced by gluing together a collection of trinions with a full untwisted and two twisted punctures, one full and the other minimal. This trinion describes the half-hypermultiplet in the bifundamental of $\mathfrak{usp}(2N-2) \times \mathfrak{so}(2N)$. This central part of the quiver is decorated at both ends by tails, whose gauge nodes display decreasing rank as we move towards the ends of the quiver. Many such tails are possible and in the class S formalism the choice of tail translates into the choice of regular puncture. The dictionary between $D_N$ punctures (both twisted and untwisted) and quiver tails is described in detail in [61]. Overall, if we have a conformal linear quiver with $k$ gauge groups, the class S description involves $k + 3$ regular punctures on the sphere. $k + 1$ of them are minimal twisted and the other two determine the structure of the tails.

As a special case of this construction, we can consider the following linear quivers:

$$\begin{aligned} k \text{ even}: \quad & \mathsf{N} + n - 1 - \mathfrak{so}(2N) - \mathfrak{usp}(2N - 2n - 2) - \cdots - \mathfrak{so}(4n + 2) - \mathfrak{usp}(2n) - 1, \\ k \text{ odd}: \quad & \mathsf{N} + n - 1 - \mathfrak{so}(2N) - \mathfrak{usp}(2N - 2n - 2) - \cdots - \mathfrak{usp}(4n) - \mathfrak{so}(2n + 2), \end{aligned} \tag{47}$$

which carry a non-trivial defect group of lines. Every $\mathfrak{so}$ gauge algebra contributes a $\mathbb{Z}_2^2$ factor to the defect group. For every $n < N$ these linear quivers can be realized using the 6d theory

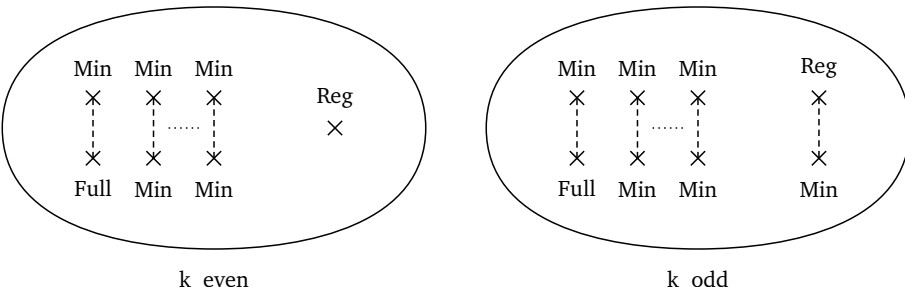

Figure 17: Class S configurations for the quivers in (47). In both cases the UV curve is a sphere, quivers with an even or odd number of gauge nodes are distinguished by the puncture structure.

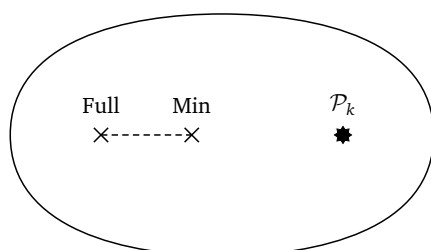

Figure 18: The Class S construction for conformal quivers (48).

of type $D_N$ with regular punctures only. We display the corresponding punctured spheres in figure 17. The tails on the left are trivial and the corresponding puncture is full twisted.

Notice that the flavor symmetry of our quivers is larger than that we naively expect from the class S description, since the full twisted puncture carries $\mathfrak{usp}(2N-2)$ flavor symmetry but from the flavors on the left end of the quivers we get $\mathfrak{usp}(2N+2n-2)$. This is the crucial feature we need for our analysis, as we will now see.

The next step is to give infinite mass to $n$ fundamentals of $\mathfrak{so}(2N)$ in (47), so that they decouple. This move clearly does not change the defect group of lines of the theory since we are not decoupling $N-1$ of the remaining flavors. After this modification the $\mathfrak{so}(2N)$ gauge algebra is no longer conformal and therefore the new quivers cannot be described using regular punctures only. We therefore ask if we can still provide a class S description. The answer is yes but we need to introduce irregular punctures. Indeed the conformal quivers

$$
\begin{aligned}
&k \text{ even}: \quad \mathsf{N} - \mathfrak{usp}(2N - 2n - 2) - \cdots - \mathfrak{so}(4n + 2) - \mathfrak{usp}(2n) - 1\,, \\
&k \text{ odd}: \quad \mathsf{N} - \mathfrak{usp}(2N - 2n - 2) - \cdots - \mathfrak{usp}(4n) - \mathfrak{so}(2n + 2)
\end{aligned}
\tag{48}
$$

are known to be described by a 6d $\mathcal{N} = (2, 0)$ $D_N$ theory on a sphere with a full untwisted puncture and an irregular puncture [103]. In that reference these theories were dubbed $D_k(SO(2N))$ and we will discuss them more in detail later. The only difference between the quivers (48) and those in (47) after the mass deformation is the gauging of the $\mathfrak{so}(2N)$ flavor symmetry and the addition of the $N-1$ flavors. This modification is easily implemented in class S since it simply corresponds to gluing the sphere with the irregular puncture together with the trinion we described before (full and minimal twisted punctures and full untwisted puncture). The resulting Riemann surface is a sphere with the irregular puncture and two twisted punctures, one full and one minimal. This is depicted in figure 18 where we denote with $\mathcal{P}_k$ the irregular puncture.

By comparing figures 17 and 18 we see that the effect of the mass deformation, which does not change the defect group, is to induce the collision of the $k+1$ regular punctures into the irregular puncture $\mathcal{P}_k$. This irregular puncture must carry trapped defect group of lines, as the

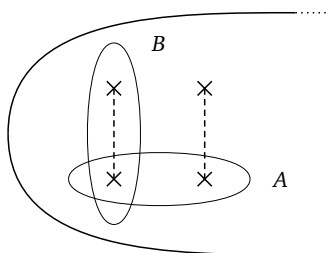

Figure 19: Picture of four twisted punctures in some patch of the UV curve together with one-cycles $A, B$ associated with lines.

1-cycles on the Riemann surface shown in figure 18 do not give rise to any non-trivial defect group.

In fact, the existence of trapped defect groups inside $\mathcal{P}_k$ can be understood as follows. Consider a configuration of four twisted regular punctures shown in figure 19. As discussed in [31], each of the cycles A and B contributes a $\mathbb{Z}_2$ group of line defects. As we collide the four punctures together, the A and B cycles, and hence the contributions associated to them, become trapped at the puncture.

The above analysis shows that irregular punctures can carry nontrivial trapped defect groups that can provide trapped contributions to one-form symmetries of absolute $4d$ $\mathcal{N} = 2$ Class S theories. The Lagrangian theories we have just discussed represent special cases of this phenomenon and in the rest of this paper we will explain how to determine the contribution from irregular punctures to the defect group of line defects.

# 4  1-Form Symmetries of Arbitrary Class S Theories

Here we discuss the computation of 1-form symmetry of an arbitrary Class S theory obtained by compactifying a $6d$ $\mathcal{N} = (2,0)$ theory of A, D, E type an arbitrary genus $g$ Riemann surface containing arbitrary number of twisted and untwisted, regular and irregular punctures, along with an arbitrary number of closed outer-automorphism twist lines.

The 1-form symmetry of the Class S theory can be expressed in terms of data associated to the punctures participating in the class S construction. This data associated to punctures is referred to as the defect group associated to the punctures.

Thus, using the analysis presented in this section, one can reduce the computation of 1-form symmetries of Class S theories to the computation of defect groups associated to punctures. We discuss the computations of defect groups of punctures in the following sections.

## 4.1  The Defect Group of a Puncture

**Decomposition into electric and magnetic lines.** A codimension-two defect (i.e. a puncture[10]) $\mathcal{P}$ is characterized by a defect group of line defects

$$\mathcal{L}_{\mathcal{P}} = \mathcal{W}_{\mathcal{P}} \times \mathcal{H}_{\mathcal{P}}. \tag{49}$$

The "electric" part $\mathcal{W}_{\mathcal{P}}$ is Pontryagin dual to the "magnetic" part $\mathcal{H}_{\mathcal{P}}$, i.e.

$$\mathcal{W}_{\mathcal{P}} = \widehat{\mathcal{H}_{\mathcal{P}}}. \tag{50}$$

---

[10]Here we consider both untwisted and twisted punctures. The codimension-two defect associated to an untwisted puncture is a genuine codimension-two defect of the $6d$ theory. On the other hand, the codimension-two defect associated to a twisted puncture is a non-genuine codimension-two defect of the $6d$ theory that lives at the end of a codimension-one topological defect associated to an outer-automorphism 0-form symmetry of the $6d$ theory.

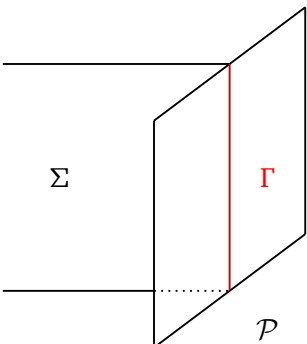

Figure 20: Bulk surface defect $\Sigma$ supporting $g \in Z_{\mathcal{P}} \subset \mathcal{S}_{6d}$, ending on the co-dimension-two defect $\mathcal{P}$ and bound by the line defect $\Gamma$ supporting $h \in \mathcal{H}_{\mathcal{P}}$. This implies $\tilde{\pi}_{\mathcal{P}}(h) = g$.

The pairing for two elements $(w_1, h_1), (w_2, h_2) \in \mathcal{W}_{\mathcal{P}} \times \mathcal{H}_{\mathcal{P}}$ is given by

$$\langle w_1, h_2 \rangle - \langle w_2, h_1 \rangle \in \mathbb{R}/\mathbb{Z}, \tag{51}$$

where $\langle w, h \rangle \in \mathbb{R}/\mathbb{Z}$ is given by applying the element $w \in \widehat{\mathcal{H}}_{\mathcal{P}}$ on the element $h \in \mathcal{H}_{\mathcal{P}}$.

**Magnetic lines and their relationship to surface defects.** The elements of $\mathcal{H}_{\mathcal{P}}$ are line defects living on $\mathcal{P}$ that arise at the ends of surface defects[11] of the bulk 6d $\mathcal{N} = (2, 0)$ theory ending on the co-dimension-two defect $\mathcal{P}$. See figure 20.

Thus, for an untwisted puncture, there is a natural projection map

$$\tilde{\pi}_{\mathcal{P}} : \mathcal{H}_{\mathcal{P}} \to \mathcal{S}_{6d} \tag{52}$$

that forgets the data of the line defect arising at the end, and spits out only the bulk surface defect. The bulk surface defects are valued in a group $\mathcal{S}_{6d}$ which we identify with the group $\widehat{Z}_G$ Pontryagin dual to the center $Z_G$ of the simply connected group $G$ associated to the A,D,E algebra $\mathfrak{g}$ specifying the 6d $\mathcal{N} = (2, 0)$ theory. The mutual non-locality of the surface defects is captured by a pairing on $\mathcal{S}_{6d} = \widehat{Z}_G$, which provides an isomorphism $p_{6d}$ between $\widehat{Z}_G$ and $Z_G$. See table 1 in section 2.1.

On the other hand, for a twisted puncture, the analogous projection map is

$$\tilde{\pi}_{\mathcal{P}} : \mathcal{H}_{\mathcal{P}} \to \mathcal{S}_{6d}^o, \tag{53}$$

where $o$ is the outer-automorphism symmetry element that the twisted puncture $\mathcal{P}$ is attached to, and $\mathcal{S}_{6d}^o$ is obtained from $\mathcal{S}_{6d}$ by modding out by the action of $o$ as follows

$$\mathcal{S}_{6d}^o = \frac{\mathcal{S}_{6d}}{(1-o) \cdot \mathcal{S}_{6d}}, \tag{54}$$

where

$$\mathcal{S}_{6d} \supseteq (1-o) \cdot \mathcal{S}_{6d} := \left\{ \alpha - o \cdot \alpha \in \mathcal{S}_{6d} \,\middle|\, \alpha \in \mathcal{S}_{6d} \right\} \tag{55}$$

and $o \cdot \alpha \in \mathcal{S}_{6d}$ is obtained by applying the action of the outer-automorphism $o$ on $\alpha \in \mathcal{S}_{6d}$. The co-domain of $\tilde{\pi}_{\mathcal{P}}$ is $\mathcal{S}_{6d}^o$ because the move shown in figure 21 relates a line defect arising at the end of $\alpha \in (1-o) \cdot \mathcal{S}_{6d}$ to a line defect arising at the end of a trivial line defect. Thus, modulo

---

[11]These are not all such line defects: In fact, a general element of $\alpha \in \mathcal{H}_{\mathcal{P}} \times \mathcal{W}_{\mathcal{P}}$ arises at the end of a surface defect. The key point is that the bulk surface defect depends only on the projection of $\alpha$ onto the $\mathcal{H}_{\mathcal{P}}$ factor, but does not depend on the projection of $\alpha$ onto the $\mathcal{W}_{\mathcal{P}}$ factor.

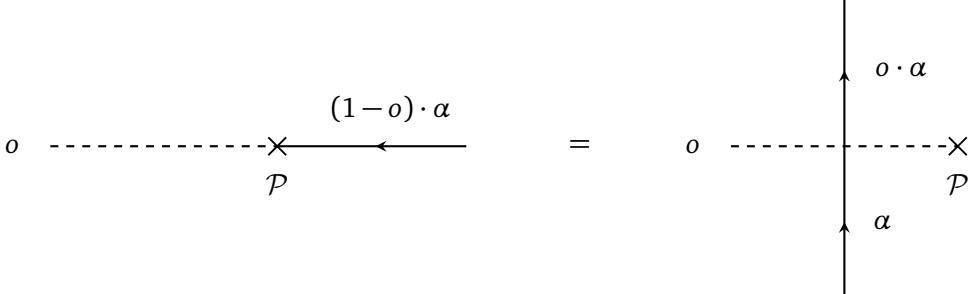

Figure 21: A surface defect $(1 - o) \cdot \alpha$ ending on an $o$-twisted puncture $\mathcal{P}$ can be topologically related to surface defect not ending on the puncture.

screenings, line defects living on $\mathcal{P}$ arising at the ends of surface defects in $(1 - o) \cdot \mathcal{S}_{6d}$ are equivalent to genuine line defects living on $\mathcal{P}$. For future purposes, let us define the projection map

$$\pi_o : \mathcal{S}_{6d} \rightarrow \mathcal{S}_{6d}^o \tag{56}$$

associated to (54).

Moving forwards, we often combine the twisted and untwisted cases by regarding the untwisted case as a special case of the twisted cases obtained by choosing $o$ to be the identity element in the outer-automorphism group.

**Trapped magnetic lines.** Now consider two elements $h_1, h_2 \in \mathcal{H}_\mathcal{P}$ whose image in $\mathcal{S}_{6d}^o$ is the same. Then, $h_1 - h_2 \in \mathcal{H}_\mathcal{P}$ describes a line defect living on $\mathcal{P}$ which is not attached to any bulk surface defect. We call such a line defect as a line defect *trapped* at the codimension-two defect (or puncture) $\mathcal{P}$. There is a subgroup $\mathcal{H}_\mathcal{P}^T \subseteq \mathcal{H}_\mathcal{P}$ which are trapped line defects. In total, we have a short exact sequence

$$0 \rightarrow \mathcal{H}_\mathcal{P}^T \rightarrow \mathcal{H}_\mathcal{P} \rightarrow Z_\mathcal{P} \rightarrow 0, \tag{57}$$

where the map

$$i_\mathcal{P} : \mathcal{H}_\mathcal{P}^T \rightarrow \mathcal{H}_\mathcal{P} \tag{58}$$

in (57) is the inclusion map making $\mathcal{H}_\mathcal{P}^T$ a subgroup of $\mathcal{H}_\mathcal{P}$, and

$$Z_\mathcal{P} = \widetilde{\pi}_\mathcal{P}(\mathcal{H}_\mathcal{P}) \subseteq \mathcal{S}_{6d}^o \tag{59}$$

is the subgroup of bulk surface defects in $\mathcal{S}_{6d}$ that can end on $\mathcal{P}$. The map

$$\pi_\mathcal{P} : \mathcal{H}_\mathcal{P} \rightarrow Z_\mathcal{P} \tag{60}$$

in (57) is the map $\widetilde{\pi}_\mathcal{P}$ with co-domain restricted to the image $Z_\mathcal{P} \subseteq \mathcal{S}_{6d}^o$ of $\widetilde{\pi}_\mathcal{P}$.

**Electric lines and their relationship to surface defects.** The elements of $\mathcal{W}_\mathcal{P}$ are genuine line defects living on $\mathcal{P}$ that are mutually non-local with the line defects in $\mathcal{H}_\mathcal{P}$. A subgroup $\mathcal{W}_\mathcal{P}^\mathcal{S}$ of $\mathcal{W}_\mathcal{P}$ is obtained from bulk surface defects by the following procedure. First define

$$\mathcal{S}_{6d} \supseteq \mathcal{S}_{6d,o} := \left\{ \alpha \in \mathcal{S}_{6d} \middle| o \cdot \alpha = \alpha \right\}. \tag{61}$$

For an untwisted puncture, $o$ is identity and hence $\mathcal{S}_{6d,o} = \mathcal{S}_{6d}$. The bulk surface defects in $\mathcal{S}_{6d,o}$ can be wrapped along a loop linking $\mathcal{P}$ as the outer-automorphism codimension-one topological defect associated to $o$ leaves them invariant. Now take a surface defect in $\mathcal{S}_{6d,o}$

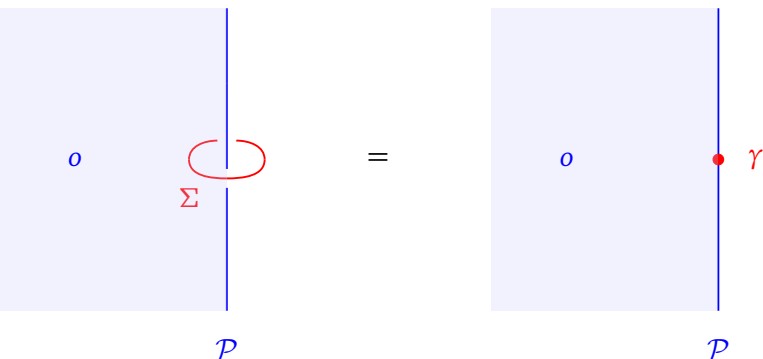

Figure 22: The bulk surface defect $\Sigma$ labelled by $v \in \mathcal{S}_{6d,o}$ links the $o$-twisted co-dimension-two defect $\mathcal{P}$ (left). The 'squeezing' operation gives a line defect $\gamma$ labelled by $w \in \mathcal{W}_{\mathcal{P}}^{\mathcal{S}}$ (right). This implies $\pi_{\mathcal{P}}^{\mathcal{S}}(v) = w$.

and wrap it along such a loop. Squeezing the loop to zero size leaves behind a line defect in $\mathcal{W}_{\mathcal{P}}^{\mathcal{S}}$ living on $\mathcal{P}$. See figure 22. In other words, we define $\mathcal{W}_{\mathcal{P}}^{\mathcal{S}}$ to be the subgroup of genuine line defects living on $\mathcal{P}$ that can be 'lifted' to bulk surface defects of the $6d$ theory. Thus, we have a projection map

$$\pi_{\mathcal{P}}^{\mathcal{S}} : \mathcal{S}_{6d,o} \to \mathcal{W}_{\mathcal{P}}^{\mathcal{S}}. \tag{62}$$

Before moving forward, notice that the pairing on $\mathcal{S}_{6d}$ descends to a pairing between $\mathcal{S}_{6d,o}$ and $\mathcal{S}_{6d}^{o}$. To show this, we need to show that $\langle \alpha, \beta \rangle = 0$ if $\alpha \in \mathcal{S}_{6d,o} \subseteq \mathcal{S}_{6d}$ and $\beta \in (1-o)\cdot\mathcal{S}_{6d} \subseteq \mathcal{S}_{6d}$. Indeed,

$$\langle \alpha, \beta \rangle = \langle \alpha, \gamma \rangle - \langle \alpha, o \cdot \gamma \rangle = \langle \alpha, \gamma \rangle - \langle o \cdot \alpha, o \cdot \gamma \rangle = \langle \alpha, \gamma \rangle - \langle \alpha, \gamma \rangle = 0, \tag{63}$$

establishing the well-defined-ness of the pairing between $\mathcal{S}_{6d,o}$ and $\mathcal{S}_{6d}^{o}$.

We now argue that

$$\mathcal{W}_{\mathcal{P}}^{\mathcal{S}} \simeq \widehat{Z}_{\mathcal{P}}. \tag{64}$$

That is, $\mathcal{W}_{\mathcal{P}}^{\mathcal{S}}$ can be identified as the Pontryagin dual of $Z_{\mathcal{P}}$. Because of the lifting procedure, the pairing of a line defect $\alpha \in \mathcal{W}_{\mathcal{P}}^{\mathcal{S}}$ with a line defect $\beta \in \mathcal{H}_{\mathcal{P}}$ depends only on the bulk surface defect $\pi_{\mathcal{P}}(\beta) \in Z_{\mathcal{P}}$. Thus

$$\alpha \in \widehat{Z}_{\mathcal{P}} \tag{65}$$

and we have a map

$$i_{\mathcal{P}}^{\mathcal{S}} : \mathcal{W}_{\mathcal{P}}^{\mathcal{S}} \to \widehat{Z}_{\mathcal{P}}. \tag{66}$$

This map is injective because if a non-zero element of $\mathcal{W}_{\mathcal{P}}^{\mathcal{S}}$ has trivial pairing with $Z_{\mathcal{P}}$, then it has a trivial pairing with the whole of $\mathcal{H}_{\mathcal{P}}$, which is in contradiction to the definition of $\mathcal{W}_{\mathcal{P}}$. To see that the map is surjective, notice that for each element $\beta \in \widehat{Z}_{\mathcal{P}}$ there is always an element $\alpha \in \mathcal{S}_{6d,o}$ such that $i_{\mathcal{P}}^{\mathcal{S}} \circ \pi_{\mathcal{P}}^{\mathcal{S}}(\alpha) = \beta$, because we can write

$$i_{\mathcal{P}}^{\mathcal{S}} \circ \pi_{\mathcal{P}}^{\mathcal{S}} = \widehat{j}_{\mathcal{P}} \circ p_{6d}^{o}, \tag{67}$$

where $\widehat{j}_{\mathcal{P}} \circ p_{6d}^{o}$ is surjective, as

$$\widehat{j}_{\mathcal{P}} : \widehat{\mathcal{S}}_{6d}^{o} \to \widehat{Z}_{\mathcal{P}} \tag{68}$$

is the Pontryagin dual of the natural inclusion map

$$j_{\mathcal{P}} : Z_{\mathcal{P}} \to \mathcal{S}_{6d}^{o} \tag{69}$$

and

$$p_{6d}^{o} : \mathcal{S}_{6d,o} \to \widehat{\mathcal{S}}_{6d}^{o} \tag{70}$$

is the isomorphism between $\mathcal{S}_{6d,o}$ and $\widehat{\mathcal{S}}_{6d}^{o}$ induced by the pairing between $\mathcal{S}_{6d}^{o}$ and $\mathcal{S}_{6d,o}$. In what follows, we sometimes use $\widehat{Z}_{\mathcal{P}}$ to denote the line defects in $\mathcal{W}_{\mathcal{P}}^{\mathcal{S}}$.

Let us also define

$$\mathcal{S}_{6d,o} \supseteq Y_{\mathcal{P}} := \ker(\widehat{j}_{\mathcal{P}} \circ p_{6d}^{o}), \tag{71}$$

which allows us to write

$$\widehat{Z}_{\mathcal{P}} = \mathcal{S}_{6d,o}/Y_{\mathcal{P}}. \tag{72}$$

$Y_{\mathcal{P}} \subseteq \mathcal{S}_{6d,o}$ is physically the subgroup of bulk surface defects which give rise to the identity line defect (up to screening) on $\mathcal{P}$ after performing the above "squeezing" procedure.

**Trapped electric lines.**  The groups $\mathcal{W}_{\mathcal{P}}$ and $\widehat{Z}_{\mathcal{P}}$ sit in a short exact sequence

$$0 \to \widehat{Z}_{\mathcal{P}} \to \mathcal{W}_{\mathcal{P}} \to \mathcal{W}_{\mathcal{P}}^{T} \to 0, \tag{73}$$

which is Pontryagin dual to the short exact sequence (57). We have

$$\mathcal{W}_{\mathcal{P}}^{T} = \widehat{\mathcal{H}}_{\mathcal{P}}^{T}, \tag{74}$$

which characterizes the *trapped* part of $\mathcal{W}_{\mathcal{P}}$, capturing the line defects that cannot be "lifted" to bulk surface defects. Correspondingly, we call the defect group

$$\mathcal{L}_{\mathcal{P}}^{T} := \mathcal{W}_{\mathcal{P}}^{T} \times \mathcal{H}_{\mathcal{P}}^{T} \tag{75}$$

as the *trapped defect group* associated to $\mathcal{P}$.  The pairing for two elements $(w_1, h_1), (w_2, h_2) \in \mathcal{W}_{\mathcal{P}}^{T} \times \mathcal{H}_{\mathcal{P}}^{T}$ is given by

$$\langle w_1, h_2 \rangle_T - \langle w_2, h_1 \rangle_T \in \mathbb{R}/\mathbb{Z}, \tag{76}$$

where $\langle w, h \rangle_T \in \mathbb{R}/\mathbb{Z}$ is given by applying the element $w \in \widehat{\mathcal{H}}_{\mathcal{P}}^{T}$ on the element $h \in \mathcal{H}_{\mathcal{P}}^{T}$.

**Genuine lines and summary.**  For future purposes, let us also define

$$\mathcal{L}_{\mathcal{P}}^{0} := \mathcal{W}_{\mathcal{P}} \times \mathcal{H}_{\mathcal{P}}^{T}, \tag{77}$$

which is the group formed by all "genuine" line defects living on $\mathcal{P}$ that do not arise at the ends of any non-trivial bulk surface defect.

In summary, the defect group associated to a puncture is uniquely determined by the magnetic data which fits into a nested pair of short exact sequences as

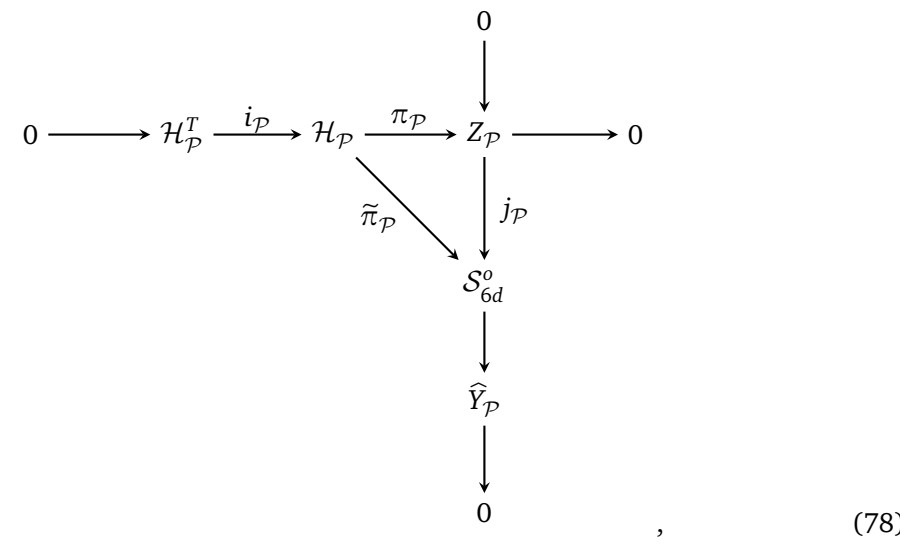

$$\tag{78}$$

with the electric data given by Pontryagin dual diagram. The horizontal sequence relates line defects living on the codimension-two defect $\mathcal{P}$. The vertical sequence relates surface defects of the 6d bulk theory.

## 4.2 Defect Groups of Special Punctures

**Regular Punctures.**    For any regular puncture $\mathcal{P}$, we can rephrase the claims of [31] as stating that $\mathcal{L}_{\mathcal{P}}^T = 0$ and $Y_{\mathcal{P}} = \mathcal{S}_{6d,o}$, which taken together imply that

$$\mathcal{L}_{\mathcal{P}} = 0\,. \tag{79}$$

$\mathcal{P}_0$ **Puncture.**    Consider $6d$ $\mathcal{N} = (2,0)$ theory of type $\mathfrak{g}$. This theory carries a special untwisted puncture that we call $\mathcal{P}_0$. This puncture has the property that

$$\mathcal{H}_{\mathcal{P}_0} = Z_{\mathcal{P}_0} = \mathcal{S}_{6d}\,. \tag{80}$$

This implies that it has

$$\mathcal{W}_{\mathcal{P}_0}^T = \mathcal{H}_{\mathcal{P}_0}^T = Y_{\mathcal{P}_0} = 0 \tag{81}$$

and

$$\mathcal{W}_{\mathcal{P}_0} = \mathcal{W}_{\mathcal{P}_0}^{\mathcal{S}} = \mathcal{S}_{6d}\,. \tag{82}$$

The $\mathcal{P}_0$ puncture is obtained by starting from the maximal untwisted regular puncture, which carries flavor symmetry $\mathfrak{g}$, and then gauging this flavor symmetry by adding $4d$ $\mathcal{N} = 2$ $\mathfrak{g}$ vector multiplet along the codimension-two defect. To show that the defect data of the puncture obtained after $\mathfrak{g}$ gauging is as claimed above, we simply notice that squeezing bulk surface defects onto the maximal untwisted regular puncture leads to flavor Wilson lines valued in $\widehat{Z}_G = \mathcal{S}_{6d}$. This means that, after the gauging, squeezing bulk surface defects onto the $\mathcal{P}_0$ puncture leads to gauge Wilson line defects valued in $\mathcal{S}_{6d}$, implying that

$$\mathcal{W}_{\mathcal{P}_0}^{\mathcal{S}} = \mathcal{S}_{6d}\,. \tag{83}$$

Moreover, the full set of electric line defects on $\mathcal{P}_0$ puncture is given by

$$\mathcal{W}_{\mathcal{P}_0} = \mathcal{S}_{6d}\,. \tag{84}$$

These two properties reproduce all the other properties claimed above.

$\mathcal{P}_0^o$ **Puncture.**    Consider $6d$ $\mathcal{N} = (2,0)$ theory of type $\mathfrak{g}$ and an outer-automorphism $o$. Then we have a special twisted puncture $\mathcal{P}_0^o$ attached to the outer-automorphism $o$. This puncture has the property that

$$\mathcal{H}_{\mathcal{P}_0^o} = Z_{\mathcal{P}_0^o} = \mathcal{S}_{6d}^o\,. \tag{85}$$

This implies that it has

$$\mathcal{W}_{\mathcal{P}_0^o}^T = \mathcal{H}_{\mathcal{P}_0^o}^T = Y_{\mathcal{P}_0^o} = 0 \tag{86}$$

and

$$\mathcal{W}_{\mathcal{P}_0^o} = \mathcal{W}_{\mathcal{P}_0^o}^{\mathcal{S}} = \mathcal{S}_{6d,o}\,. \tag{87}$$

The $\mathcal{P}_0^o$ puncture is obtained by starting from the maximal twisted regular puncture associated to $o$, which carries flavor symmetry $\mathfrak{h}_o^\vee$ which is the Langlands dual of the subalgebra[12] $\mathfrak{h}_o$ of $\mathfrak{g}$ left invariant by the action of $o$. Let us note that, except for the case of $\mathfrak{g} = \mathfrak{su}(2n+1)$ and non-trivial $o$, there is an isomorphism between $\mathcal{S}_{6d,o}$ and the group $\widehat{Z}_{H_o^\vee}$ Pontryagin dual to the center $Z_{H_o^\vee}$ of the simply connected group $H_o^\vee$ associated to the algebra $\mathfrak{h}_o^\vee$. Depending on $\mathfrak{g}$ and $o$, both $\mathcal{S}_{6d,o}$ and $\widehat{Z}_{H_o^\vee}$ are either trivial or form a $\mathbb{Z}_2$. In both cases, there is a unique possible isomorphism between the two groups. For the case of $\mathfrak{g} = \mathfrak{su}(2n+1)$ and non-trivial $o$, $\mathcal{S}_{6d,o} = 0$ and $\widehat{Z}_{H_o^\vee} = \mathbb{Z}_2$.

---

[12]The choices of $\mathfrak{h}_o$ for various $\mathfrak{g}$ and $o$ can be found in Table 1 of [104].

This flavor symmetry $\mathfrak{h}_o^\vee$ is then gauged by adding 4d $\mathcal{N}=2$ $\mathfrak{h}_o^\vee$ vector multiplet along the codimension-two defect. To show that the defect data of the puncture obtained after $\mathfrak{h}_o^\vee$ gauging is as claimed above, we simply notice that squeezing bulk surface defects valued in $\mathcal{S}_{6d,o}$ onto the maximal twisted regular puncture leads to flavor Wilson lines valued in $\mathcal{S}_{6d,o} \simeq \widehat{Z}_{H_o^\vee}$. Now we divide further analysis into two cases:

- For all cases except $\mathfrak{g} = \mathfrak{su}(2n+1)$ and non-trivial $o$, the flavor Wilson lines in $\widehat{Z}_{H_o^\vee}$ are not screened by any genuine local operators, or in other words, there are no genuine local operators carrying non-trivial flavor center charges valued in $\widehat{Z}_{H_o^\vee}$. This means that, after the gauging, squeezing bulk surface defects onto the $\mathcal{P}_0^o$ puncture leads to gauge Wilson line defects valued in $\mathcal{S}_{6d,o} \simeq \widehat{Z}_{H_o^\vee}$, implying that

$$\mathcal{W}_{\mathcal{P}_0^o}^{\mathcal{S}} = \mathcal{S}_{6d,o}. \tag{88}$$

  Moreover, the full set of electric line defects on $\mathcal{P}_0^o$ puncture is given by

$$\mathcal{W}_{\mathcal{P}_0^o} = \widehat{Z}_{H_o^\vee} \simeq \mathcal{S}_{6d,o}. \tag{89}$$

  These two properties reproduce all the other properties claimed above.

- For the case of $\mathfrak{g} = \mathfrak{su}(2n+1)$ and non-trivial $o$, the flavor Wilson lines in $\widehat{Z}_{H_o^\vee} = \mathbb{Z}_2$ are screened by genuine local operators [104]. Thus, after gauging, there are no non-trivial electric line operators (modulo screenings) carried by such a $\mathcal{P}_0^o$ puncture. This leads to the properties claimed above.

### 4.3 Defect Groups of Class S Theories from Defect Groups of Punctures

We now combine the defect groups of punctures discussed in the last subsection together with the surface defect contributions discussed in [31], to determine the defect group of an arbitrary class S theory, containing an arbitrary number of twisted and untwisted, irregular and regular punctures, on an arbitrary genus Riemann surface.

**Various types of possible lines.** Consider a general compactification of a 6d $\mathcal{N}=(2,0)$ theory of type $\mathfrak{g}$, leading to a 4d $\mathcal{N}=2$ theory $\mathfrak{T}$. Let $\mathcal{P}_i$ with $i=1,2,\cdots,k$ label the various punctures on the compactification manifold $\Sigma_g$ of genus $g$. Additionally, we have an outer-automorphism background $[B] \in H^1(\Sigma_g^*, \mathcal{O}_{\mathfrak{g}})$, where $\Sigma_g^*$ is the compactification manifold with punctures removed, and $\mathcal{O}_{\mathfrak{g}}$ is the group of outer-automorphisms of $\mathfrak{g}$. Near a puncture $\mathcal{P}_i$, the background $[B]$ has a holonomy $o_i \in \mathcal{O}_{\mathfrak{g}}$ around $\mathcal{P}_i$, where $o_i$ is the outer-automorphism element associated to $\mathcal{P}_i$.

First of all, wrapping bulk surface defects along compact 1-cycles of $\Sigma_g$, we generate 4d line defects in

$$\mathcal{K} := H_1^{[B]}(\Sigma_g^*, \mathcal{S}_{6d}), \tag{90}$$

which is the first homology group of $\Sigma_g^*$ twisted by $[B]$.

We moreover have the line defects living on each $\mathcal{P}_i$ that are not attached to any bulk surface defect. These form the group

$$\prod_i \mathcal{H}_{\mathcal{P}_i}^T \times \prod_i \mathcal{W}_{\mathcal{P}_i}. \tag{91}$$

Finally, we include line defects arising from the bulk surface defects ending on punctures. For this purpose, we pick a non-self-intersecting path $C_{i,i+1}$ from puncture $i$ to puncture $i+1$ for

each $1 \leq i \leq k-1$. We additionally require that there is no intersection between two paths $C_{i,i+1}$ and $C_{j,j+1}$ for $i \neq j$. Let us also define for $1 \leq i < j \leq k$

$$C_{i,j} = \sum_{p=i}^{j-1} C_{p,p+1} \,. \tag{92}$$

More precisely, $C_{i,j}$ is a path from $i$ to $j$ obtained by taking the $\sum_{p=i}^{j-1} C_{p,p+1}$ and perturbing it slightly away from the punctures $i+1, i+2, \cdots, j-1$, so that $C_{i,j}$ doesn't hit those punctures. Moreover, let $Z_{i,j}$ be the subgroup of $\mathcal{S}_{6d}$ generated by $0 \in \mathcal{S}_{6d}$ and non-zero elements $\alpha \in Y_{i,j} \subseteq \mathcal{S}_{6d}$ having the property that either $\pi_{o_i}(\alpha)$ or $\pi_{o_j}(\alpha)$ is non-zero, where

$$Y_{i,j} := \pi_{o_i}^{-1}(Z_{\mathcal{P}_i}) \cap \pi_{o_j}^{-1}(Z_{\mathcal{P}_j}) \tag{93}$$

$4d$ line defects arising from bulk surface defects wrapped along the path $C_{i,j}$ form a group $\mathcal{H}_{i,j}$ which as a set is

$$\mathcal{H}_{i,j} = \bigsqcup_{\alpha \in Z_{i,j}} \mathcal{H}_{i,j}^{\alpha} \,, \tag{94}$$

where

$$\mathcal{H}_{i,j}^{\alpha} = \left\{ (\beta, \gamma) \in \mathcal{H}_{\mathcal{P}_i} \times \mathcal{H}_{\mathcal{P}_j} \,\middle|\, \pi_{\mathcal{P}_i}(\beta) = \pi_{o_i}(\alpha), \pi_{\mathcal{P}_j}(\gamma) = \pi_{o_j}(-\alpha) \right\}. \tag{95}$$

The group structure on $\mathcal{H}_{i,j}$ is as follows. Take an element $(\beta_1, \gamma_1) \in \mathcal{H}_{i,j}^{\alpha_1}$ and an element $(\beta_2, \gamma_2) \in \mathcal{H}_{i,j}^{\alpha_2}$. Then we have

$$(\beta_1, \gamma_1) + (\beta_2, \gamma_2) = (\beta_1 + \beta_2, \gamma_1 + \gamma_2) \in \mathcal{H}_{i,j}^{\alpha_1 + \alpha_2} \,. \tag{96}$$

Notice that any other $4d$ line defect can be written as a linear combination of the $4d$ line defects discussed above, which form the group

$$\mathcal{K}_{\mathfrak{I}} := \mathcal{K} \times \prod_{i=1}^{k} \mathcal{H}_{\mathcal{P}_i}^T \times \prod_{i=1}^{k} \mathcal{W}_{\mathcal{P}_i} \times \prod_{1 \leq i < j \leq k} \mathcal{H}_{i,j} \,. \tag{97}$$

**Equivalences between lines and the defect group.** Not all the elements lying in $\mathcal{K}_{\mathfrak{I}}$ give rise to distinct line defects modulo screenings. First of all, $\mathcal{H}_{i,j}$ contains line defects lying in $\mathcal{H}_{\mathcal{P}_i}^T \times \mathcal{H}_{\mathcal{P}_j}^T$. This imposes the identification

$$\mathcal{H}_{i,j} \supseteq \mathcal{H}_{i,j}^{\alpha=0} \ni (\beta, \gamma) \sim (i_{\mathcal{P}_i}^{-1}\beta, i_{\mathcal{P}_j}^{-1}\gamma) \in \mathcal{H}_{\mathcal{P}_i}^T \times \mathcal{H}_{\mathcal{P}_j}^T \,. \tag{98}$$

Second type of identification arises if $o_i = o_j^{-1}$:

$$\mathcal{H}_{i,j} \supset \mathcal{H}_{i,j}^{\alpha} \ni (\gamma, \delta) \sim i_{\mathcal{P}_i}^{-1}(\gamma) + i_{\mathcal{P}_j}^{-1}(\delta) + \beta[C] \in \mathcal{H}_{\mathcal{P}_i}^T \times \mathcal{H}_{\mathcal{P}_j}^T \times \mathcal{K} \,, \tag{99}$$

where $\alpha = \beta - o_i \cdot \beta$ for some $\beta \in \mathcal{S}_{6d}$, and $\beta[C] \in \mathcal{K}$ is obtained by wrapping $\beta$ along the loop $C$ encircling the two punctures $\mathcal{P}_i$ and $\mathcal{P}_j$ in a frame obtained by choosing a representative $B$ of the outer-automorphism background $[B]$ in which the outer-automorphism twist lines can be represented in the vicinity of $C_{i,j}$ as shown in figure 23.

To describe the third identification, pick punctures $1 \leq i_1 < i_2 < i_3 \leq k$. Define

$$Z_{i_1,i_2,i_3} := Z_{i_1,i_2} \cap Z_{i_2,i_3} \tag{100}$$

and pick $\alpha \in Z_{i_1,i_2,i_3}$. Then, we have the identification

$$\mathcal{H}_{i_1,i_2}^{\alpha} \times \mathcal{H}_{i_2,i_3}^{\alpha} \ni (\beta, \gamma) + (-\gamma, \delta) \sim (\beta, \delta) \in \mathcal{H}_{i_1,i_3}^{\alpha} \,, \tag{101}$$

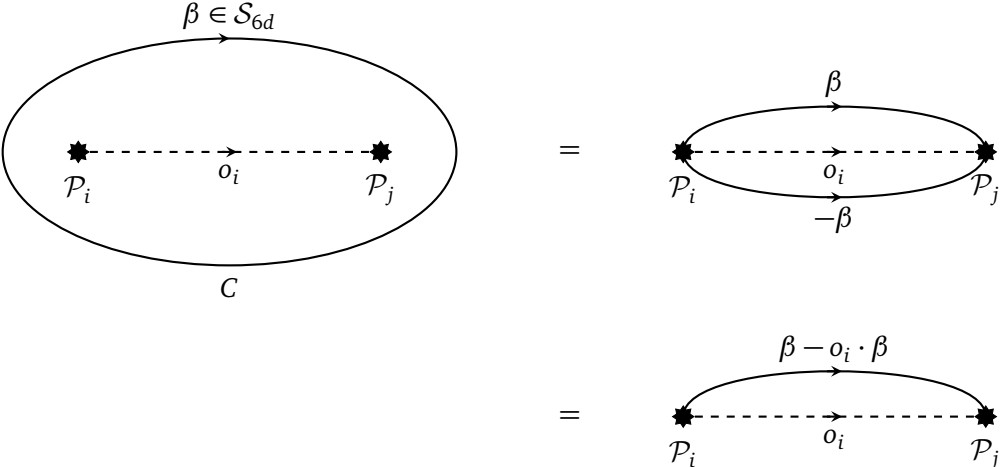

Figure 23: Topological move showing that a $4d$ line defect obtained by wrapping the surface defect $\beta \in \mathcal{S}_{6d}$ along the loop $C$ encircling punctures $\mathcal{P}_i$ and $\mathcal{P}_j$ is equivalent to a $4d$ line defect obtained by wrapping $\beta - o_i \cdot \beta$ along the path $C_{i,j}$ going from the puncture $\mathcal{P}_i$ to the puncture $\mathcal{P}_j$. Here the dashed line denotes the outer-automorphism twist line.

for elements $(\beta, \gamma) \in \mathcal{H}^\alpha_{i_1, i_2}$ and $(-\gamma, \delta) \in \mathcal{H}^\alpha_{i_2, i_3}$.

Finally, consider the $4d$ line defect $\alpha[C_i] \in \mathcal{K}$ obtained by wrapping an element $\alpha \in \mathcal{S}_{6d, o_i}$ along a loop $C_i$ linking the puncture $\mathcal{P}_i$ as shown below

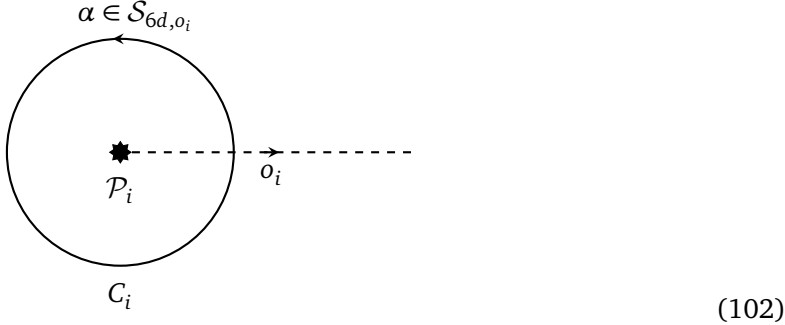

$$\tag{102}$$

As we have discussed in the previous subsection, we have the following identification

$$\mathcal{K} \ni \alpha[C_i] \sim \pi^\mathcal{S}_{\mathcal{P}_i}(\alpha) \in \mathcal{W}^\mathcal{S}_{\mathcal{P}_i} \subseteq \mathcal{W}_{\mathcal{P}_i}. \tag{103}$$

In particular, the $4d$ line defect $\alpha[C_i] \in \mathcal{K}$ is equivalent to trivial line defect for $\alpha \in Y_{\mathcal{P}_i}$.

Thus, in total, we find that the defect group of line defects of the $4d$ $\mathcal{N} = 2$ Class S theory $\mathfrak{T}$ is

$$\mathcal{L}_\mathfrak{T} = \mathcal{K}_\mathfrak{T} / \sim, \tag{104}$$

with the identifications $\sim$ generated by (98), (99), (101) and (103).

**Pairing on the defect group.** The pairing on $\mathcal{L}_\mathfrak{T}$ descends from the pairing on $\mathcal{K}_\mathfrak{T}$, which can be concretely described after choosing a representative co-chain $B \in C^1(\Sigma^*_g, \mathcal{O}_\mathfrak{g})$ of the outer-automorphism background $[B] \in H^1(\Sigma^*_g, \mathcal{O}_\mathfrak{g})$. Let $b \in C_1(\Sigma^*_g, \mathcal{O}_\mathfrak{g})$ be the chain obtained from $B$ by applying Poincaré duality. Notice that the chain $b$ ends on each twisted puncture $\mathcal{P}_i$ carrying the element $o_i$ in the neighborhood of $\mathcal{P}_i$. We additionally require that $b$ does not intersect the paths $C_{i,j}$.

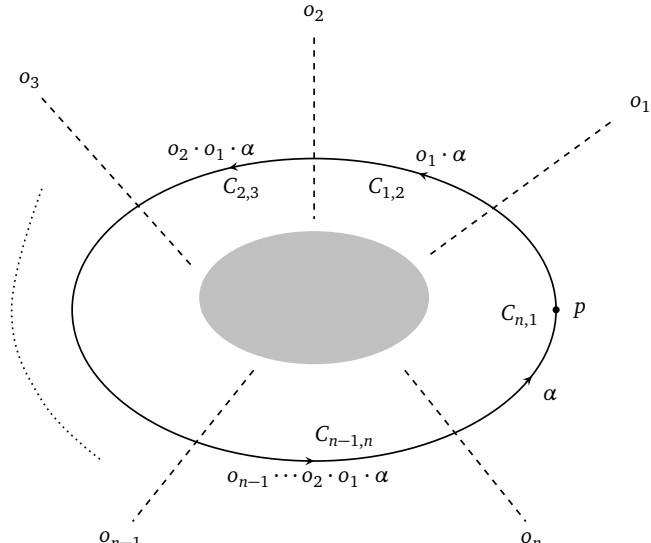

Figure 24: A loop $C$ on $\Sigma_g$. As one traverses the loop counter-clockwise starting from point $p$, the loop intersects various outer-automorphism twist lines $o_1, o_2, \cdots, o_n$. The sub-segment of $C$ between the locations of intersections with $o_i$ and $o_{i+1}$ is called $C_{i,i+1}$. Once the element $\alpha \in \mathcal{S}_{6d}$ carried by $C$ at point $p$ is specified, the element carried by the sub-segment $C_{i,i+1}$ is fixed to be $o_i \cdot o_{i-1} \cdots o_2 \cdot o_1 \cdot \alpha$.

Picking such a representative allows us to construct elements of $\mathcal{K}$ as follows. Pick a non-self-intersecting loop $C$ and a point $p$ in $C$. As we traverse the loop $C$ starting from $p$, we find that it intersects $b$ at $n$ number of points $p_1, p_2, \cdots p_n$. Thus the loop is divided into segments $C_{i,i+1}$ lying between points $p_i, p_{i+1}$. Let $b$ carry the element $o_i$ at the intersection point $p_i$. Then, we obtain an element $\alpha_p[C] \in \mathcal{K}$ by inserting an element $\alpha \in \mathcal{S}_{6d}$ at point $p$. The element $\alpha$ must satisfy $o_n \cdot o_{n-1} \cdots o_2 \cdot o_1 \cdot \alpha = \alpha$. The element of $\mathcal{S}_{6d}$ inserted at any other point $q$ along the loop is then determined uniquely in terms of $\alpha$. Along the segment between $p_n$ and $p_1$, it is $\alpha$. And along the segment $C_{i,i+1}$ it is $o_i \cdot o_{i-1} \cdots o_2 \cdot o_1 \cdot \alpha$. See figure 24. Such elements $\alpha_p[C]$ for all choices of $\alpha, p, C$ generate via linear combinations the whole of $\mathcal{K}$.

The pairing can now be concretely described as follows:

- There is a self-pairing on the $\mathcal{K}$ subfactor of $\mathcal{K}_{\mathfrak{T}}$ provided by combining the pairing on $\mathcal{S}_{6d}$ with the intersection pairing on $H_1(\Sigma_g, \mathbb{Z})$. More concretely, consider two elements $\alpha_p[C], \beta_q[D] \in \mathcal{K}$. Let $C$ and $D$ intersect at points $r_1, r_2, \cdots, r_m$. Let $\alpha_{r_i}$ and $\beta_{r_i}$ be the elements of $\mathcal{S}_{6d}$ carried by $\alpha_p[C]$ and $\beta_q[D]$ respectively at the point $r_i$. Then the pairing can be described as

$$\langle \alpha_p[C], \beta_q[D] \rangle_{\mathcal{K}_{\mathfrak{T}}} = \sum_{i=1}^{m} (-1)^{\sigma_i} \langle \alpha_{r_i}, \beta_{r_i} \rangle_{\mathcal{S}_{6d}}, \tag{105}$$

where $(-1)^{\sigma_i}$ captures the intersection pairing of $C$ and $D$ in the neighborhood of the point $r_i$.

- Consider elements $\alpha_p[C] \in \mathcal{K}$ and $(\gamma, \delta) \in \mathcal{H}_{i,j}^{\beta}$. Let $C$ intersect $C_{i,j}$ at the points $r_1, r_2, \cdots, r_m$. Let $\alpha_{r_a}$ be the element of $\mathcal{S}_{6d}$ carried by $\alpha_p[C]$ at the point $r_a$. Then the pairing between the two elements is given by

$$\langle \alpha([C]), (\gamma, \delta) \rangle_{\mathcal{K}_{\mathfrak{T}}} = \sum_{a=1}^{m} (-1)^{\sigma_a} \langle \alpha_{r_a}, \beta \rangle_{\mathcal{S}_{6d}}, \tag{106}$$

where $(-1)^{\sigma_a}$ captures the intersection pairing of $C$ and $C_{i,j}$ in the neighborhood of the point $r_a$.

- There is a pairing between $\alpha \in \mathcal{W}_{\mathcal{P}_i}$ and $(\gamma, \delta) \in \mathcal{H}_{i,j}^{\beta}$ given by

$$\langle \alpha, (\gamma, \delta) \rangle_{\mathcal{K}_{\mathfrak{T}}} = \langle \alpha, \gamma \rangle_{\mathcal{L}_{\mathcal{P}_i}} \tag{107}$$

and a pairing between $\alpha \in \mathcal{W}_{\mathcal{P}_j}$ and $(\gamma, \delta) \in \mathcal{H}_{i,j}^{\beta}$ given by

$$\langle \alpha, (\gamma, \delta) \rangle_{\mathcal{K}_{\mathfrak{T}}} = \langle \alpha, \delta \rangle_{\mathcal{L}_{\mathcal{P}_j}}. \tag{108}$$

- Finally, there is a pairing between $\mathcal{H}_{\mathcal{P}_i}^{T}$ and $\mathcal{W}_{\mathcal{P}_i}$ given by the pairing on $\mathcal{L}_{\mathcal{P}_i}$.

- All other pairings are trivial.

The elements of $\mathcal{K}_{\mathfrak{T}}$ identified by (98), (99), (101) and (103) have the same pairings with all the elements of $\mathcal{K}_{\mathfrak{T}}$. Thus $\mathcal{L}_{\mathfrak{T}}$ obtains a well-defined pairing $\langle \cdot, \cdot \rangle_{\mathcal{L}_{\mathfrak{T}}}$ descending from the pairing $\langle \cdot, \cdot \rangle_{\mathcal{K}_{\mathfrak{T}}}$ on $\mathcal{K}_{\mathfrak{T}}$. Conversely, if we have two elements $\alpha, \beta \in \mathcal{K}_{\mathfrak{T}}$ such that

$$\langle \alpha, \gamma \rangle_{\mathcal{K}_{\mathfrak{T}}} = \langle \beta, \gamma \rangle_{\mathcal{K}_{\mathfrak{T}}}, \tag{109}$$

for all $\gamma \in \mathcal{K}_{\mathfrak{T}}$, then $\alpha$ and $\beta$ are identified by the identifications (98), (99), (101) and (103). Thus, there is no element $\alpha \in \mathcal{L}_{\mathfrak{T}}$ such that

$$\langle \alpha, \beta \rangle_{\mathcal{L}_{\mathfrak{T}}} = 0, \tag{110}$$

for all $\beta \in \mathcal{L}_{\mathfrak{T}}$.

**1-form symmetry from the defect group.** So far we have been discussing the 4d $\mathcal{N} = 2$ theory $\mathfrak{T}$ obtained directly using the Class S construction, without any additional choices. Such a 4d theory is relative and has a defect group that we discussed in detail above in this subsection. An absolute 4d $\mathcal{N} = 2$ Class S theory $\mathfrak{T}_{\Lambda}$ can be chosen from this relative Class S theory $\mathfrak{T}$ by choosing a polarization $\Lambda$, which is a maximal subgroup of $\mathcal{L}_{\mathfrak{T}}$ on which the pairing trivializes. The 1-form symmetry $\mathcal{O}_{\mathfrak{T}_{\Lambda}}$ of the absolute 4d $\mathcal{N} = 2$ Class S theory $\mathfrak{T}_{\Lambda}$ can then be described as

$$\mathcal{O}_{\mathfrak{T}_{\Lambda}} = \widehat{\Lambda}. \tag{111}$$

# 5 Computing Defect Groups of Punctures

## 5.1 Special Class S Theories Associated to a Puncture

In the previous section we discussed how the defect group of a Class S theory can be computed in terms of defect groups of punctures participating in the compactification. In this subsection, we discuss how the defect group of a puncture can be computed in terms of the defect groups associated to some special Class S theories whose construction involves the puncture under consideration. If an alternative way of computing the defect groups of the special Class S theories is known, then one obtains the defect group associated to the puncture. We will discuss such alternative ways in subsequent sections.

**Computing the trapped defect group.** Consider an untwisted puncture $\mathcal{P}$ of 6d $\mathcal{N} = (2,0)$ theory of type $\mathfrak{g}$. Our first claim is that the trapped defect group $\mathcal{L}_{\mathcal{P}}^T$ associated to $\mathcal{P}$ is obtained as the defect group of line defects of the 4d $\mathcal{N} = 2$ Class S theory $\mathfrak{T}_{\mathcal{P}}$ obtained by compactifying the 6d $\mathcal{N} = (2,0)$ theory on a sphere carrying only a single puncture, with the puncture being of type $\mathcal{P}$. Let us compute the defect group of this Class S theory $\mathfrak{T}_{\mathcal{P}}$ using the formalism developed in the previous section. First of all, we have

$$\mathcal{K} = H_1(\Sigma_g^*, \mathcal{S}_{6d}) = 0. \tag{112}$$

Moreover, since we have a single puncture $\mathcal{P}$, we can write

$$\mathcal{K}_{\mathfrak{T}_{\mathcal{P}}} = \mathcal{H}_{\mathcal{P}}^T \times \mathcal{W}_{\mathcal{P}}. \tag{113}$$

The loop $C_i$ linking $\mathcal{P}_i$ is homologically trivial. So the identification (103) imposes that

$$\widehat{Z}_{\mathcal{P}} \ni \alpha \sim 0 \in \mathcal{L}_{\mathfrak{T}_{\mathcal{P}}}. \tag{114}$$

Modding out by this identification, we find that the defect group of this Class S theory is

$$\mathcal{L}_{\mathfrak{T}_{\mathcal{P}}} = \widetilde{\mathcal{L}}_{\mathfrak{T}_{\mathcal{P}}} / \sim \, = \mathcal{H}_{\mathcal{P}}^T \times \mathcal{W}_{\mathcal{P}}^T = \mathcal{L}_{\mathcal{P}}^T, \tag{115}$$

as claimed above.

Note that if we add untwisted regular punctures on the sphere, then the defect group of the resulting Class S theory is also $\mathcal{L}_{\mathcal{P}}^T$. In particular, we will often consider the Class S theory $\mathfrak{T}_{\mathcal{P}}^*$ obtained by compactifying $\mathcal{N} = (2,0)$ theory on a sphere containing a single puncture of type $\mathcal{P}$ and a single untwisted maximal regular puncture, for computing $\mathcal{L}_{\mathcal{P}}^T$ via

$$\mathcal{L}_{\mathfrak{T}_{\mathcal{P}}^*} = \mathcal{L}_{\mathcal{P}}^T. \tag{116}$$

For a twisted puncture $\mathcal{P}$, we define $\mathfrak{T}_{\mathcal{P}}$ to be the 4d $\mathcal{N} = 2$ Class S theory obtained by compactifying 6d $(2,0)$ theory on a sphere with two punctures: one of them being of type $\mathcal{P}$, and the other being a minimal regular twisted puncture. One can easily show, in a similar fashion as above, that

$$\mathcal{L}_{\mathfrak{T}_{\mathcal{P}}} = \mathcal{L}_{\mathcal{P}}^T. \tag{117}$$

We can replace the minimal regular twisted puncture by any other regular twisted puncture, and the defect group of the resulting Class S theory is also $\mathcal{L}_{\mathcal{P}}^T$. In particular, we will often consider the Class S theory $\mathfrak{T}_{\mathcal{P}}^*$ obtained by replacing the minimal regular twisted puncture by a maximal regular twisted puncture, for computing $\mathcal{L}_{\mathcal{P}}^T$ via

$$\mathcal{L}_{\mathfrak{T}_{\mathcal{P}}^*} = \mathcal{L}_{\mathcal{P}}^T. \tag{118}$$

**Computing the full defect group.** Our second claim is that $\mathcal{L}_{\mathcal{P}}$ for a puncture $\mathcal{P}$ associated to outer-automorphism $o^{-1}$ is the defect group of the Class S theory $\mathfrak{T}_{\mathcal{P}}^{\mathcal{P}_0^o}$ obtained by considering compactification of $(2,0)$ theory on a sphere with a single puncture $\mathcal{P}_i$ of type $\mathcal{P}$ and a single puncture $\mathcal{P}_j$ of type $\mathcal{P}_0^o$. We have

$$\mathcal{K}_{\mathfrak{T}_{\mathcal{P}}^{\mathcal{P}_0^o}} = \mathcal{K} \times \mathcal{H}_{\mathcal{P}}^T \times \mathcal{W}_{\mathcal{P}} \times \mathcal{W}_{\mathcal{P}_0^o} \times \mathcal{H}_{i,j}, \tag{119}$$

with

$$\mathcal{W}_{\mathcal{P}_0^o} \simeq \mathcal{S}_{6d,o} \tag{120}$$

and

$$\mathcal{K} \simeq \mathcal{S}_{6d,o}, \tag{121}$$

generated by wrapping bulk surface defects on a loop $C$ on the sphere that divides the sphere into two hemispheres such that each hemisphere contains exactly one puncture. We choose $C_{ij}$ to be a path from $\mathcal{P}$ to $\mathcal{P}_0$ such that $C_{ij}$ intersects $C$ at one point. We have

- For trivial $o$, i.e. for an untwisted puncture $\mathcal{P}$, $Z_{i,j} = Z_{\mathcal{P}}$ and $\mathcal{H}_{i,j} = \mathcal{H}_{\mathcal{P}}$. The identification (98) identifies $\mathcal{H}_{\mathcal{P}}^T$ subfactor of $\mathcal{K}_{\mathfrak{T}_{\mathcal{P}}^{\mathcal{P}_0}}$ as the $\mathcal{H}_{\mathcal{P}}^T$ subgroup of $\mathcal{H}_{i,j} = \mathcal{H}_{\mathcal{P}}$. The identification (103) identifies $\mathcal{K} \simeq \mathcal{S}_{6d}$ with $\mathcal{W}_{\mathcal{P}_0}$ when squeezed onto $\mathcal{P}_0$, and with the $\widehat{Z}_{\mathcal{P}}$ subgroup of $\mathcal{W}_{\mathcal{P}}$ when squeezed onto $\mathcal{P}$. In total, we obtain

$$\mathcal{L}_{\mathfrak{T}_{\mathcal{P}}^{\mathcal{P}_0}} = \frac{\mathcal{K}_{\mathfrak{T}_{\mathcal{P}}^{\mathcal{P}_0}}}{\sim} = \mathcal{W}_{\mathcal{P}} \times \mathcal{H}_{i,j} = \mathcal{W}_{\mathcal{P}} \times \mathcal{H}_{\mathcal{P}} = \mathcal{L}_{\mathcal{P}}, \tag{122}$$

as claimed.

- For non-trivial $o$ and $Z_{\mathcal{P}} \neq 0$, $Z_{i,j} = \pi_o^{-1}(Z_{\mathcal{P}})$ and so

$$\mathcal{H}_{i,j} = \bigsqcup_{\alpha \in \pi_o^{-1}(Z_{\mathcal{P}})} \mathcal{H}_{i,j}^{\alpha}, \tag{123}$$

with

$$\mathcal{H}_{i,j}^{\alpha} = \pi_{\mathcal{P}}^{-1} \pi_o(\alpha). \tag{124}$$

Using (99) and (98), we see that $\mathcal{H}_{i,j}^{\alpha}$ is equivalent to $\mathcal{H}_{\mathcal{P}}^T$ subfactor of $\mathcal{K}_{\mathfrak{T}_{\mathcal{P}}^{\mathcal{P}_0^o}}$ for all $\alpha \in \pi_o^{-1}(0)$. These identifications project $\mathcal{H}_{\mathcal{P}}^T \times \mathcal{H}_{i,j}$ to $\mathcal{H}_{\mathcal{P}}$. The identification (103) identifies $\mathcal{K} \simeq \mathcal{S}_{6d,o}$ with $\mathcal{W}_{\mathcal{P}_0^o}$ when squeezed onto $\mathcal{P}_0^o$, and with the $\widehat{Z}_{\mathcal{P}}$ subgroup of $\mathcal{W}_{\mathcal{P}}$ when squeezed onto $\mathcal{P}$. In total, we obtain

$$\mathcal{L}_{\mathfrak{T}_{\mathcal{P}}^{\mathcal{P}_0^o}} = \frac{\mathcal{K}_{\mathfrak{T}_{\mathcal{P}}^{\mathcal{P}_0^o}}}{\sim} = \mathcal{W}_{\mathcal{P}} \times \mathcal{H}_{\mathcal{P}} = \mathcal{L}_{\mathcal{P}}, \tag{125}$$

as claimed.

- For non-trivial $o$ and $Z_{\mathcal{P}} = 0$, $Z_{i,j} = 0$ and so

$$\mathcal{H}_{i,j} \simeq \mathcal{H}_{\mathcal{P}}^T. \tag{126}$$

The identification (98) identifies $\mathcal{H}_{\mathcal{P}}^T$ subfactor of $\mathcal{K}_{\mathfrak{T}_{\mathcal{P}}^{\mathcal{P}_0^o}}$ with $\mathcal{H}_{i,j} \simeq \mathcal{H}_{\mathcal{P}}^T$. The identification (103) identifies $\mathcal{K} \simeq \mathcal{S}_{6d,o}$ with $\mathcal{W}_{\mathcal{P}_0^o}$ when squeezed onto $\mathcal{P}_0^o$, and with the $\widehat{Z}_{\mathcal{P}} = 0$ subgroup of $\mathcal{W}_{\mathcal{P}}$ when squeezed onto $\mathcal{P}$. In total, we obtain

$$\mathcal{L}_{\mathfrak{T}_{\mathcal{P}}^{\mathcal{P}_0^o}} = \frac{\mathcal{K}_{\mathfrak{T}_{\mathcal{P}}^{\mathcal{P}_0^o}}}{\sim} = \mathcal{W}_{\mathcal{P}}^T \times \mathcal{H}_{\mathcal{P}}^T = \mathcal{L}_{\mathcal{P}}^T = \mathcal{L}_{\mathcal{P}}. \tag{127}$$

The last equality $\mathcal{L}_{\mathcal{P}}^T = \mathcal{L}_{\mathcal{P}}$ follows from the fact that $Z_{\mathcal{P}} = \widehat{Z}_{\mathcal{P}} = 0$.

## 5.2 Generalized Quivers: Classes $\mathcal{G}\mathcal{Q}$ and $\mathcal{G}\mathcal{Q}'$

We will conjecture (and in some cases argue) that the defect groups of lines associated to a large class of (conformal and non-conformal) punctures can be computed as the defect groups of lines of generalized quiver gauge theories lying in a subclass $\mathcal{G}\mathcal{Q}$ of all 4d $\mathcal{N} = 2$ generalized quiver gauge theories.

**Relating defect groups of punctures and generalized quivers.** A generalized quiver $\tau \in \mathcal{GQ}$ takes the following form

$$\mathfrak{g}_0 \underline{\quad M_1 \quad} \mathfrak{g}_1 \underline{\quad M_2 \quad} \mathfrak{g}_2 \underline{\quad} \cdots \underline{\quad} \mathfrak{g}_{k-1} \underline{\quad M_k \quad} [\mathfrak{g}_k] \ , \tag{128}$$

where $\mathfrak{g}_i$ are gauge algebras for $0 \le i \le k-1$ and $\mathfrak{g}_k$ is a flavor algebra. Moreover, we have

$$\mathfrak{g}_0 = \mathfrak{h}_o^\vee \,, \tag{129}$$

where $\mathfrak{h}_o^\vee$ is the Langlands dual to the subalgebra $\mathfrak{h}_o$ of the 6d A,D,E algebra $\mathfrak{g}$ left invariant by the outer-automorphism $o$ associated to the puncture under study. Each edge $M_i$ for $1 \le i \le k$ denotes a 4d $\mathcal{N} = 2$ 'matter' SCFT whose flavor algebra is gauged by the neighboring gauge algebra nodes.

Then, the conjecture states that there is a large family $\mathcal{F}_\tau$ of $o$-twisted punctures of 6d $\mathcal{N} = (2,0)$ theory of type $\mathfrak{g}$ such that the defect group associated to a puncture $\mathcal{P} \in \mathcal{F}_\tau$ is the same as the defect group associated to $\tau$

$$\mathcal{L}_\mathcal{P} = \mathcal{L}_\tau \,. \tag{130}$$

Moreover, the trapped part of the defect group associated to $\mathcal{P}$ is computed as the defect group associated to the generalized quiver $\widetilde{\tau}$ obtained from $\tau$ by treating $\mathfrak{g}_0 = \mathfrak{g}$ as a flavor algebra

$$\mathcal{L}_\mathcal{P}^T = \mathcal{L}_{\widetilde{\tau}} \,, \tag{131}$$

where $\widetilde{\tau}$ takes the form

$$[\mathfrak{g}_0] \underline{\quad M_1 \quad} \mathfrak{g}_1 \underline{\quad M_2 \quad} \mathfrak{g}_2 \underline{\quad} \cdots \underline{\quad} \mathfrak{g}_{k-1} \underline{\quad M_k \quad} [\mathfrak{g}_k] \ . \tag{132}$$

**Relating generalized quivers and special Class S theories.** A subclass $\mathcal{GQ}' \subseteq \mathcal{GQ}$ of generalized quivers has the property that $\tau' \in \mathcal{GQ}'$ can be realized as the 4d $\mathcal{N} = 2$ Class S theory $\mathfrak{T}_{\mathcal{P}'}^{\mathcal{P}_0^o}$

$$\tau' = \mathfrak{T}_{\mathcal{P}'}^{\mathcal{P}_0^o} \tag{133}$$

associated to an $o$-twisted puncture $\mathcal{P}'$ of type $\mathfrak{g}$ $(2,0)$ theory. In such a situation, we also have the relationship

$$\widetilde{\tau}' = \mathfrak{T}_{\mathcal{P}'}^* \,. \tag{134}$$

**Structure of the defect group of a generalized quiver.** The line defects participating in the defect group of the quiver theory $\tau$ arise from 't Hooft-Wilson line defects associated to the gauge algebras. Each gauge factor $\mathfrak{g}_i$ provides

$$\mathcal{L}_i = \mathcal{H}_i \times \mathcal{W}_i \tag{135}$$

defect group of lines, with $\mathcal{H}_i$ being the 't Hooft lines and $\mathcal{W}_i$ being the Wilson lines. We can take

$$\mathcal{H}_i = Z_{G_i} \tag{136}$$

and

$$\mathcal{W}_i = \widehat{Z}_{G_i} \,, \tag{137}$$

where $Z_{G_i}$ is the center of the simply connected group $G_i$ associated to the simple Lie algebra $\mathfrak{g}_i$, and $\widehat{Z}_{G_i}$ is the Pontryagin dual of $Z_{G_i}$. The matter SCFTs for $\tau \in \mathcal{GQ}$ have the special property

that they do not provide any line defects modulo screenings. Thus, before accounting for local operators coming from the matter SCFTs, we have an initial defect group

$$\mathcal{L} = \mathcal{H} \times \mathcal{W} = \prod_{i=0}^{k-1} \mathcal{H}_i \times \prod_{i=0}^{k-1} \mathcal{W}_i. \tag{138}$$

Let us now include the local operators. The genuine local operators of the matter SCFT $M_i$ charged non-trivially under the gauged subgroup of the flavor symmetry $\mathfrak{f}_i$ of $M_i$ become non-genuine local operators of the quiver theory obtained after the gauging procedure. The gauge center charges of non-genuine local operators arising from $M_i$ span a sub-lattice

$$\Gamma_i \subseteq \widehat{Z}_{G_{i-1}} \times \widehat{Z}_{G_i}. \tag{139}$$

Let

$$\Gamma \subseteq \prod_{i=0}^{k-1} \widehat{Z}_{G_i} \tag{140}$$

be the sub-lattice generated by combining the contributions $\Gamma_i$ from all matter SCFTs $M_i$. Then, we can write the defect group of lines of the quiver theory $\tau$ as

$$\mathcal{L}_\tau = \mathcal{H}_\tau \times \mathcal{W}_\tau, \tag{141}$$

where the Wilson line contribution $\mathcal{W}_\tau$ is obtained by screening the Wilson lines in $\mathcal{W}$ by $\Gamma$

$$\mathcal{W}_\tau = \mathcal{W}/\Gamma = \frac{\prod_{i=0}^{k-1} \mathcal{W}_i}{\Gamma} \tag{142}$$

and the 't Hooft line contribution $\mathcal{H}_\tau$ is constrained to be a subgroup of the 't Hooft lines in $\mathcal{H}$ as they have to be mutually local with $\Gamma$. Notice that this is just the Pontryagin dual of $\mathcal{W}_\tau$. Thus, we have

$$\mathcal{H}_\tau = \widehat{\mathcal{W}}_\tau. \tag{143}$$

**Computing defect groups using 1-form symmetry.** In fact, the defect group $\mathcal{L}_\tau$ can be computed in terms of the 1-form symmetry group of an absolute theory obtained by choosing the electric polarization

$$\Lambda_\tau^e = \mathcal{W}_\tau \subset \mathcal{L}_\tau. \tag{144}$$

This polarization corresponds to choosing the gauge group $\mathcal{G}_\tau$ of the quiver theory $\tau$ as

$$\mathcal{G}_\tau = \prod_{i=0}^{k-1} G_i, \tag{145}$$

where each $G_i$ is simply connected. The 1-form symmetry $\mathcal{O}_\tau^e$ of this absolute theory is the Pontryagin dual of the corresponding polarization

$$\mathcal{O}_\tau^e = \widehat{\Lambda}_\tau^e = \mathcal{H}_\tau. \tag{146}$$

Thus

$$\mathcal{H}_\mathcal{P} = \mathcal{H}_\tau, \tag{147}$$

for a puncture $\mathcal{P} \in \mathcal{F}_\tau$ can be computed by computing the 1-form symmetry $\mathcal{O}_\tau^e$ of the generalized quiver theory $\tau$ with all gauge groups chosen to be simply connected. Then,

$$\mathcal{W}_\mathcal{P} = \mathcal{W}_\tau \tag{148}$$

is readily computed as the Pontryagin dual $\widehat{\mathcal{O}}_\tau^e$ of the 1-form symmetry group $\mathcal{O}_\tau^e$. In total, the defect group associated to puncture $\mathcal{P}$ can be computed as

$$
\begin{aligned}
\mathcal{L}_\mathcal{P} &= \mathcal{H}_\mathcal{P} \times \mathcal{W}_\mathcal{P}, \\
\mathcal{H}_\mathcal{P} &= \mathcal{O}_\tau^e, \\
\mathcal{W}_\mathcal{P} &= \widehat{\mathcal{O}}_\tau^e
\end{aligned}
\tag{149}
$$

and the pairing on $\mathcal{L}_\mathcal{P}$ is obtained simply as the natural pairing of $\widehat{\mathcal{O}}_\tau^e$ with $\mathcal{O}_\tau^e$.

To compute the trapped part $\mathcal{L}_\mathcal{P}^T$ we use the quiver theory $\widetilde{\tau}$

$$
[\mathfrak{g}_0] \;\rule[0.5ex]{0.6cm}{0.4pt}\; {\scriptstyle M_1} \;\rule[0.5ex]{0.6cm}{0.4pt}\; \mathfrak{g}_1 \;\rule[0.5ex]{0.6cm}{0.4pt}\; {\scriptstyle M_2} \;\rule[0.5ex]{0.6cm}{0.4pt}\; \mathfrak{g}_2 \;\rule[0.5ex]{0.6cm}{0.4pt}\; \cdots \;\rule[0.5ex]{0.6cm}{0.4pt}\; \mathfrak{g}_{k-1} \;\rule[0.5ex]{0.6cm}{0.4pt}\; {\scriptstyle M_k} \;\rule[0.5ex]{0.6cm}{0.4pt}\; [\mathfrak{g}_k] \;.
\tag{150}
$$

Let us write the defect group associated to this theory as

$$
\mathcal{L}_{\widetilde{\tau}} = \mathcal{H}_{\widetilde{\tau}} \times \mathcal{W}_{\widetilde{\tau}},
\tag{151}
$$

which we can easily compute using the same trick as above. We choose the absolute theory with gauge group

$$
\mathcal{G}_{\widetilde{\tau}} = \prod_{i=1}^{k-1} G_i
\tag{152}
$$

being the product of simply connected groups associated to the simple factors in the gauge algebra. Let $\mathcal{O}_{\widetilde{\tau}}^e$ be the 1-form symmetry group of this absolute theory. Then we can identify

$$
\begin{aligned}
\mathcal{L}_\mathcal{P}^T &= \mathcal{H}_\mathcal{P}^T \times \mathcal{W}_\mathcal{P}^T, \\
\mathcal{H}_\mathcal{P}^T &= \mathcal{O}_{\widetilde{\tau}}^e, \\
\mathcal{W}_\mathcal{P}^T &= \widehat{\mathcal{O}}_{\widetilde{\tau}}^e,
\end{aligned}
\tag{153}
$$

with the pairing on $\mathcal{L}_{\mathcal{P}'}^T$ being simply the natural pairing of $\widehat{\mathcal{O}}_{\widetilde{\tau}}^e$ with $\mathcal{O}_{\widetilde{\tau}}^e$.

**Computing maps participating in the defect group.** Let us now describe the computation of various maps appearing in (78) for the puncture $\mathcal{P} \in \mathcal{F}_\tau$. $\mathcal{O}_\tau^e$ is a subgroup of the center

$$
Z_{\mathcal{G}_\tau} := \prod_{i=0}^{k-1} Z_{G_i}
\tag{154}
$$

of the gauge group $\mathcal{G}_\tau$ appearing in (145). Let us define maps

$$
\pi^i : Z_{\mathcal{G}_\tau} \to Z_{G_i}
\tag{155}
$$

that project $Z_{\mathcal{G}_\tau}$ onto its $Z_{G_i}$ subfactor. We can then describe

$$
\mathcal{O}_{\widetilde{\tau}}^e = \left\{ \alpha \in \mathcal{O}_\tau^e \,\middle|\, \pi^0(\alpha) = 0 \in Z_{G_0} \right\}
\tag{156}
$$

identifying $\mathcal{H}_\mathcal{P}^T = \mathcal{O}_{\widetilde{\tau}}^e$ as a subgroup of $\mathcal{H}_\mathcal{P} = \mathcal{O}_\tau^e$. This implies that we can identify

$$
\pi_\mathcal{P} = \pi^0
\tag{157}
$$

and hence

$$
Z_\mathcal{P} = \pi^0\left(\mathcal{O}_\tau^e\right) \subseteq Z_{G_0} = Z_{H_o^\vee}.
\tag{158}
$$

To obtain $Z_\mathcal{P}$ as a subgroup of $\mathcal{S}_{6d}^o$, we employ the following isomorphism between $Z_{H_o^\vee}$ and $\mathcal{S}_{6d}^o$:

- For trivial $o$, we have $Z_{H_o^\vee} = Z_G$ which can be mapped to $\mathcal{S}_{6d} = \widehat{Z}_G$ by using the pairing on $\mathcal{S}_{6d}$.

- For trivial $o$, we have only two possibilities: either $Z_{H_o^\vee} \simeq \mathbb{Z}_2$ and $\mathcal{S}_{6d}^o \simeq \mathbb{Z}_2$, or $Z_{H_o^\vee} = \mathcal{S}_{6d}^o = 0$. In both cases, there is a unique isomorphism between $Z_{H_o^\vee}$ and $\mathcal{S}_{6d}$.

Thus we have computed all the maps appearing in (78), which capture the magnetic part of $\mathcal{L}_\mathcal{P}$. As discussed earlier, the data about the electric part is obtained simply by taking Pontryagin dual of the data about the magnetic part.

## 5.3 Spectral Cover Monodromies and ALE Fibrations in IIB

We will now discuss how the part $\mathcal{L}_\mathcal{P}^0$ of the full defect group, as defined in (77), can be easily computed using the monodromy of the Hitchin field $\phi$ as one encircles the puncture $\mathcal{P}$. Note that the information about $\mathcal{L}_\mathcal{P}^0$ is insufficient in providing the full information about $\mathcal{L}_\mathcal{P}$.

**Higgs field for a general Class S compactification.** The Higgs field $\phi$ of a relative theory of class S with bulk Lie algebra $\mathfrak{g}$ and UV curve $C$ is a $\mathfrak{g}$ valued meromorphic section of the canonical bundle $K_C$ modulo gauge transformations. We consider Higgs fields which are globally diagonalizable by some gauge transformation, that is their profile lies along a Cartan subalgebra $\mathfrak{h} \subset \mathfrak{g}$. Gauge transformations bringing Higgs fields into this diagonal form are fixed up to conjugation by elements in the Weyl group $\mathfrak{w}_\mathfrak{g}$ which maps $\mathfrak{h}$ onto itself. The Higgs field is therefore a meromorphic section of $K_C \otimes (\mathfrak{h}/\mathfrak{w}_\mathfrak{g})$.

The spectral curve of a Higgs field $\phi$ with respect to a representation $\mathbf{r}$ of $\mathfrak{g}$ is[13]

$$\Sigma_\mathbf{r} = \{(z, \lambda_z) \in K_C \mid \det(\lambda_z - \phi_\mathbf{r}(z)) = 0\} \subset K_C, \tag{159}$$

where $\phi_\mathbf{r}$ is the presentation of the Higgs field $\phi$ as acting on $\mathbf{r}$. Denote the dimension of the representation by $\dim \mathbf{r} = r$, then the spectral curve $\Sigma_\mathbf{r}$ is a ramified $r$-fold covering of $C$ away from the poles of $\phi_\mathbf{r}$. The $r$ sheets of the covering are permuted by Weyl transformations when encircling untwisted punctures and branch points and by outer automorphisms $\mathcal{O}_\mathfrak{g}$ when crossing twist lines.

**Higgs field in a generic small open set.** Consider the spectral curve $\Sigma_\mathbf{r}$ restricted to a local patch $U \subset C$ away from punctures, branch points and twist lines. Across $U$ the sheets of $\Sigma_\mathbf{r}$ can be distinguished and labelled by weights of $\mathbf{r}$ and the Higgs field $\phi$ lifts to a meromorphic section of $K_U \otimes \mathfrak{h}$. The weights of the representation $\mathbf{r}$ are $r$ elements in the dual space $\mathfrak{h}^*$ of the Cartan subalgebra $\mathfrak{h}$ and we denote these by $w_i$ with $i = 1, \dots, r$. To label the sheets of $\Sigma_\mathbf{r}$ across $U$ we consider the canonical Weyl-invariant pairing $(\cdot, \cdot) : \mathfrak{h}^* \otimes \mathfrak{h} \to \mathbb{C}$ and define

$$\lambda_{z,i} = x_i(z)dz = (w_i, \phi(z)), \tag{160}$$

which gives precisely $r$ solutions (sheets) to the spectral equation $\det(\lambda_z - \phi_\mathbf{r}(z)) = 0$. Given gauge equivalent Higgs fields as sections in $K_C \otimes \mathfrak{h}$ their labelling of sheets by weights are related by Weyl transformations. There are $|\mathfrak{w}_\mathfrak{g}|$ equivalent labelings of spectral cover sheets by weights.

---

[13]Here we abbreviate by $K_C$ the total space of the canonical bundle over $C$.

**Higgs field near a puncture and monodromy.** Next consider the spectral curve $\Sigma_{\mathbf{r}}$ restricted to a local patch $V \subset C$ containing a single puncture $\mathcal{P}$. Choose complex coordinate $t$ centered on the puncture and coordinate $v$ on the cotangent fiber such that $\lambda_i = v_i \, dt/t$ which will prove convenient later in relation to IIA Hanay-Witten brane constructions. In a subset $U \subset V$ we can label the sheets $v_i$ by weights $w_i$. Encircling $\mathcal{P}$ the sheets and weights are permuted by a Weyl transformation and possibly an outer-automorphism if the puncture is twisted. This gives a monodromy action on the weight lattice $\Lambda_{\text{weight}}$. The weight lattice contains the root lattice $\Lambda_{\text{root}} \subset \Lambda_{\text{weight}}$ and therefore the monodromy lifts to an action on the root lattice. We obtain a monodromy action $M_{\mathcal{P}}$ of the puncture $\mathcal{P}$ on the root lattice of the bulk algebra $\mathfrak{g}$

$$M_{\mathcal{P}} : \Lambda_{\text{root}} \to \Lambda_{\text{root}}. \tag{161}$$

This monodromy action can be read off from any spectral curve $\Sigma_{\mathbf{r}}$ for which the weight system of the representation $\mathbf{r}$ spans $\Lambda_{\text{weight}}$ or equivalently that contains the set of fundamental weights. For Lie algebras $\mathfrak{g} = A_{n-1}, D_{n \geq 4}, E_6, E_7$ the lowest-dimensional representations with this property are the $n$-dimensional fundamental representation, the $2n$-dimensional vector representation and the representations $\mathbf{27}, \mathbf{56}$ respectively.

**ALE-fibration in Type IIB.** We discuss the physical consequences of the monodromy action $M_{\mathcal{P}}$ in the IIB dual description where the above configurations are recast in a purely geometric framework. In this picture the Higgs field is the period map for the ALE fibration

$$\widetilde{\mathbb{C}^2/\Gamma_{\mathfrak{g}}} \hookrightarrow X_3' \to V', \tag{162}$$

with respect to the holomorphic top form $\Omega$ of the Calabi-Yau three-fold $X_3'$. Here $\widetilde{\mathbb{C}^2/\Gamma_{\mathfrak{g}}}$ denotes the hyperkahler unfolding of the ADE singularity $\mathbb{C}^2/\Gamma_{\mathfrak{g}}$ to an ALE space and $V'$ denotes $V$ with the puncture $\mathcal{P}$ at $t = 0$ excised. The ALE fibration degenerates approaching the puncture. We obtain a non-degenerate ALE fibration by restricting to $V^*$ which is defined as $V$ with an open disk containing $t = 0$ removed. The geometry modelling each puncture is

$$\widetilde{\mathbb{C}^2/\Gamma_{\mathfrak{g}}} \hookrightarrow X_3 \to V^*. \tag{163}$$

The boundary of this geometry has two components $B_{\mathcal{P}}$ and $B_X$. These are fibered as

$$\begin{aligned}
\widetilde{\mathbb{C}^2/\Gamma_{\mathfrak{g}}} &\hookrightarrow B_{\mathcal{P}} \to S^1, \\
S^3/\Gamma_{\mathfrak{g}} &\hookrightarrow B_X \to V^*.
\end{aligned} \tag{164}$$

Here $S^1$ bounds the open disk excised from $V$. See figure 25. The two five-dimensional boundary components intersect along a four-dimensional corner $B_X \cap B_{\mathcal{P}} = \partial B_{\mathcal{P}}$ which is fibered as

$$S^3/\Gamma_{\mathfrak{g}} \hookrightarrow \partial B_{\mathcal{P}} \to S^1. \tag{165}$$

We are interested in the torsional two-cycles of the boundary components $B_{\mathcal{P}}$ and $B_X$ which can arise as the intersection of non-compact three-cycles in the full IIB ALE fibration. D3 branes wrapping such three-cycles engineer line defects. Both boundary components contribute to the spectrum of such two-cycles.

We begin by considering $B_X$ which is fibered over $V^*$. The base $V^*$ is topologically an annulus and deformation retracts onto a circle. The homology groups of $B_X$ therefore follow from the monodromy action on $H_k(S^3/\Gamma_{\mathfrak{g}}, \mathbb{Z})$. A non-trivial monodromy can only occur for $H_1(S^3/\Gamma_{\mathfrak{g}}, \mathbb{Z}) \cong \widehat{Z}_G \cong \Lambda_{\text{weights}}/\Lambda_{\text{roots}}$. The monodromy action on $H_1(S^3/\Gamma_{\mathfrak{g}}, \mathbb{Z})$ can therefore be inferred from the action on weights and we find it to be trivial, this implies

$$H_2(B_X, \mathbb{Z}) \cong \Gamma_{\mathfrak{g}}^{\text{ab}} \cong \widehat{Z}_G. \tag{166}$$

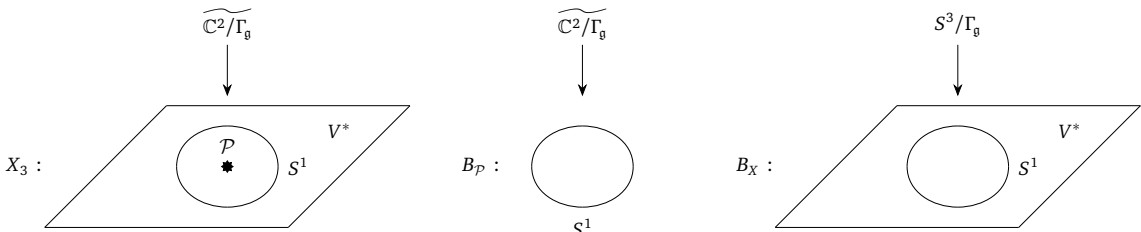

Figure 25: We depict the IIB geometry $X_3$ associated with a neigbourhood of the puncture $\mathcal{P}$. The boundary of the ALE fibration $X_3$ is given by $\partial X_3 = B_{\mathcal{P}} \cup B_X$.

Here, the superscript ab denotes the abelianization of the group.

Next consider $B_{\mathcal{P}}$. Its ALE fibers contain rank $\mathfrak{g}$ rational curves. These rational curves decompactify when $S^1$ shrinks onto the puncture. Therefore, any three-cycle in the IIB ALE fibration which intersects $B_{\mathcal{P}}$ in such a rational curve is in fact non-compact. The monodromy action associated with the puncture introduces redundancies among these rational curves with independent classes counted by $H_2(B_{\mathcal{P}}, \mathbb{Z})$. According to the results of [13], the contribution of the boundary component $B_{\mathcal{P}}$ to the defect group $\mathcal{L}_{\mathcal{P}}$ takes the form $\mathrm{Tor}\, H_2(B_{\mathcal{P}}, \mathbb{Z})$. The rational curves are associated with the simple roots of the Lie algebra $\mathfrak{g}$ by the McKay correspondence. The monodromy action on rational curves is therefore the one described in (161) which is now geometrized as

$$M_{\mathcal{P}} : H_2(\widetilde{\mathbb{C}^2/\Gamma_{\mathfrak{g}}}, \mathbb{Z}) \to H_2(\widetilde{\mathbb{C}^2/\Gamma_{\mathfrak{g}}}, \mathbb{Z}) \,. \tag{167}$$

As all smooth manifolds fibered over a circle the homology groups of $B_{\mathcal{P}}$ are determined by the monodromy mappings

$$M_k : \quad H_k(\widetilde{\mathbb{C}^2/\Gamma_{\mathfrak{g}}}, \mathbb{Z}) \to H_k(\widetilde{\mathbb{C}^2/\Gamma_{\mathfrak{g}}}, \mathbb{Z}) \,, \tag{168}$$

which enter into the short exact sequence

$$0 \to \mathrm{coker}(M_k - 1) \to H_k(B_{\mathcal{P}}, \mathbb{Z}) \to \ker(M_{k-1} - 1) \to 0 \,. \tag{169}$$

Of these mappings only $M_2 \equiv M_{\mathcal{P}}$ differs from the identity. With this we derive

$$\mathrm{Tor}\, H_2(B_{\mathcal{P}}, \mathbb{Z}) \cong \mathrm{Tor}\, \mathrm{coker}(M_{\mathcal{P}} - 1) \,. \tag{170}$$

Upon checking against predictions for defect groups via other methods, we find that

$$\mathrm{Tor}\, \mathrm{coker}(M_{\mathcal{P}} - 1) \cong \mathcal{L}_{\mathcal{P}}^0 = \mathcal{W}_{\mathcal{P}} \times \mathcal{H}_{\mathcal{P}}^T \,. \tag{171}$$

Notice that all trapped contributions are accounted for.

Finally we require a map from the boundary component $B_X$ into the total boundary $\partial X$ which determines the projection $H_2(B_X, \mathbb{Z}) \cong \widehat{Z}_G$ onto $\widehat{Z}_{\mathcal{P}}$. The cycles in the image of this map are expected to link with the cycles associated to $\mathcal{L}_{\mathcal{P}}^0$ defining a Dirac pairing from which the full data of the puncture (78) can be derived as follows. The pairing defines a Pontryagin dual pairing between $\widehat{\mathcal{L}}_{\mathcal{P}}^0 = \mathcal{H}_{\mathcal{P}} \times \mathcal{W}_{\mathcal{P}}^T$ and $Z_{\mathcal{P}}$. The elements of $\widehat{\mathcal{L}}_{\mathcal{P}}^0$ that have trivial pairing with all the elements of $Z_{\mathcal{P}}$ form the subgroup $\mathcal{L}_{\mathcal{P}}^T = \mathcal{H}_{\mathcal{P}}^T \times \mathcal{W}_{\mathcal{P}}^T$. From this we can determine

$$\mathcal{H}_{\mathcal{P}}^T = \sqrt{\mathcal{L}_{\mathcal{P}}^T} \tag{172}$$

and we obtain the injection

$$i_{\mathcal{P}} : \mathcal{H}_{\mathcal{P}}^T \to \mathcal{H}_{\mathcal{P}} \tag{173}$$

allowing us to determine all the relevant data about the full defect group $\mathcal{L}_{\mathcal{P}}$ of the puncture $\mathcal{P}$.

When discussing the ALE-fibration, we will in the following restrict our attention only to $\mathcal{L}_{\mathcal{P}}^0$, while leaving the geometric determination of the above-mentioned projection map $\widehat{Z}_G \to \widehat{Z}_{\mathcal{P}}$ to future work. Morally, this projection map is the geometric avatar of the squeezing map (62).

# 6 Type IIB on Canonical Singularities and Irregular Punctures

There is a special class of punctures (that we refer to as IHS punctures) for which the $4d$ $\mathcal{N}=2$ theory $\mathfrak{T}_{\mathcal{P}}$ defined in section 5.1 can be constructed as IIB compactified on an isolated hypersurface singularity (IHS). This provides a map between IHS singularities and irregular punctures of IHS type. The subsection 6.1 discusses this map for untwisted irregular punctures of IHS type, which is quite well-known in the literature. The following subsections 6.2–6.4 discuss the map for twisted irregular punctures. Some of the twisted cases have been discussed in prior literature, while some others are being discussed for the first time. Using the IHS description presented here, we can compute the defect groups $\mathcal{L}_{\mathfrak{T}_{\mathcal{P}}}$ associated to the theories $\mathfrak{T}_{\mathcal{P}}$, which as discussed in section 5.1, coincide with the trapped parts $\mathcal{L}_{\mathcal{P}}^T$ of the defect groups $\mathcal{L}_{\mathcal{P}}$ associated to the IHS punctures $\mathcal{P}$. Thus the IIB IHS description provides a concrete and simple way of computing the trapped defect groups of IHS punctures, which can be used as strong counter-checks for the various proposals regarding the defect groups of general punctures made in this paper.

Finally, in subsection 6.5, we discuss another class of IHS singularities that have the property that compactifying IIB on such IHS singularities leads to $4d$ $\mathcal{N}=2$ "trinion" SCFTs that are obtained by gauging a diagonal flavor symmetry of three $\mathfrak{T}_{\mathcal{P}_i}^*$ (with $i=1,2,3$) theories, where $\mathcal{P}_i$ are IHS punctures. Thus, one can compute the defect groups of these trinion theories using the IHS techniques. On the other hand, as discussed in this subsection, the same defect group can also be expressed in terms of the defect groups $\mathcal{L}_{\mathcal{P}_i}$ associated to the punctures $\mathcal{P}_i$. It turns out that the defect group of the trinion theory sees not only the trapped parts $\mathcal{L}_{\mathcal{P}_i}^T$ of $\mathcal{L}_{\mathcal{P}_i}$, but also some of the non-trapped parts. Thus, matching the defect group of trinion theories as obtained using IHS, against the defect groups predicted using our proposal, provides strong checks for the proposed non-trapped parts of the defect groups of punctures.

## 6.1 Untwisted Punctures

Let us begin the discussion by considering untwisted irregular punctures discussed in [81, 88]. Such punctures are characterized by two integers $k, b$, where $k$ takes infinitely many values, while $b$ takes either two or three possible values. We call such punctures *IHS punctures*. This is because the $4d$ $\mathcal{N}=2$ Class S theory $\mathfrak{T}_{\mathcal{P}}$, obtained by compactifying the 6d (2,0) theory on a sphere with a single puncture $\mathcal{P}$, can be constructed by compactifying Type IIB on an IHS singularity, for almost all punctures $\mathcal{P}$ discussed in [81, 88], except for the punctures with extremely small values of $k$.

The defining equation is given by an ADE singularity, of the same type as the 6d $\mathcal{N}=(2,0)$ theory relevant for the Class S realization, fibered over a complex plane. This is not surprising since 6d $\mathcal{N}=(2,0)$ theories are engineered in Type IIB by compactification on the corresponding ADE singularity and the complex plane parametrizes the punctured sphere, with the puncture located at infinity. The defining equation for the IHS is

$$P(x_1, x_2, x_3, z) = x_1^2 + F(x_2, x_3, z) = 0\,,$$
$$\Omega_3 = \frac{dx_1 \wedge dx_2 \wedge dx_3 \wedge dz}{dP}\,, \tag{174}$$

Table 2: Untwisted irregular punctures of IHS type and their hypersurface realization. We provide the hypersurface equations in $\mathbb{C}^4$, which correspond to class S theories with one irregular puncture of type $(G, b, k)$. Note that in order for the IHS to be well-defined $k$ needs to be large enough. The Type indicates the description of the singularity in KS-YY classification of IHS (see [55] for our conventions).

| 6d G | $b$ | Singularity after closure | Type($a, b, c, d$) | AD[$G, G'$] |
|------|-----|---------------------------|---------------------|--------------|
| $A_{N-1}$ | $N$ | $x_1^2 + x_2^2 + x_3^N + z^{k-N}$ | $\{1,1\}(2,2,N,k-b)$ | $(A_{N-1}, A_{k-N-1})$ |
|  | $N-1$ | $x_1^2 + x_2^2 + x_3^N + x_3 z^{k-N+1}$ | $\{2,1\}(2,2,N,k-N+1)$ |  |
| $D_N$ | $2N-2$ | $x_1^2 + x_2^{N-1} + x_2 x_3^2 + z^{k-2N+2}$ | $\{2,1\}(2,k-2N+2,N-1,2)$ | $(A_{k-2N+1}, D_N)$ |
|  | $N$ | $x_1^2 + x_2^{N-1} + x_2 x_3^2 + x_3 z^{k-N}$ | $\{7,1\}(2,N-1,2,k-N)$ |  |
| $E_6$ | $12$ | $x_1^2 + x_2^3 + x_3^4 + z^{k-12}$ | $\{1,1\}(2,3,4,k-12)$ | $(A_{k-13}, E_6)$ |
|  | $9$ | $x_1^2 + x_2^3 + x_3^4 + x_3 z^{k-9}$ | $\{2,1\}(2,3,4,k-9)$ |  |
|  | $8$ | $x_1^2 + x_2^3 + x_3^4 + x_2 z^{k-8}$ | $\{2,1\}(2,4,3,k-8)$ |  |
| $E_7$ | $18$ | $x_1^2 + x_2^3 + x_2 x_3^3 + z^{k-18}$ | $\{2,1\}(2,k-18,3,3)$ | $(A_{k-19}, E_7)$ |
|  | $14$ | $x_1^2 + x_2^3 + x_2 x_3^3 + x_3 z^{k-14}$ | $\{7,1\}(2,3,3,k-14)$ |  |
| $E_8$ | $30$ | $x_1^2 + x_2^3 + x_3^5 + z^{k-30}$ | $\{1,1\}(2,3,5,k-30)$ | $(A_{k-31}, E_8)$ |
|  | $24$ | $x_1^2 + x_2^3 + x_3^5 + x_3 z^{k-24}$ | $\{2,1\}(2,3,5,k-24)$ |  |
|  | $20$ | $x_1^2 + x_2^3 + x_3^5 + x_2 z^{k-20}$ | $\{2,1\}(2,5,3,k-20)$ |  |

where we use the coordinate $z$ to parametrize the sphere and $\Omega_3$ denotes the holomorphic three-form.

We now summarize briefly the properties and types of such IHS punctures. Concerning untwisted irregular punctures, we are interested in the case when the Higgs field for a 6d $(2,0)$ theory of type $G$ has the form

$$\Phi = \frac{1}{z^{1+\frac{k}{b}}} \cdots . \tag{175}$$

The values of $(G, k, b)$ are constrained and lead to the hypersurfaces in table 2, as was discussed in [82]. In table 2 we also provide the identifications with AD theories when appropriate.

We will also consider the closely-related family of theories $\mathfrak{T}_{\mathcal{P}}^*$ whose class S description is in terms of a sphere with one irregular puncture $\mathcal{P}$ of type IHS and a full regular puncture. Also these have a Type IIB description since they can be realized as hypersurfaces in $\mathbb{C}^3 \times \mathbb{C}^*$. The defining equations are essentially the same as in table 2. The only difference is that the coordinate $z$ is now $\mathbb{C}^*$-valued (and accordingly we replace $dz$ with $d(\log z)$ in $\Omega_3$) and the parameter $k$ should be replaced by $k+b$ (see [105]). Said differently, we see that geometrically closing the regular puncture amounts to shifting $k \to k-b$. Note that $k$ has to be large enough for the equation in table 2 to be regular. In the class S description this corresponds to the fact that unless $k$ is large enough, the full regular puncture cannot be closed [105].

## 6.2 Twisted Punctures

The situation is more subtle in the case of twisted punctures since the twist line must necessarily end at a second puncture, invalidating the above argment. We will however find that, once the second puncture is taken to be regular and minimal, the resulting SCFT can still be described by a IHS. In order to derive this statement, we start from the theory with a twisted full puncture whose Type IIB geometric description in terms of a hypersurface in $\mathbb{C}^3 \times \mathbb{C}^*$ is known explicitly [89].

Our strategy is to show that upon closure of the regular puncture we find a model which admits a IHS description in Type IIB. We provide the corresponding equations in the fourth

Table 3: Twisted irregular punctures of IHS type: In the fourth column the relevant hypersurface singularity in $\mathbb{C}^4$ is provided and in the last column we report the corresponding singularity type according to the KS-YY classification (see [55] for our conventions). Entries in red are those already identified in [106]. In the first column we provide the corresponding 6d $\mathcal{N} = (2, 0)$ theory and in the second column we indicate the outer-automorphism twist considered.

| 6d Type | Twist | $b_t$ | Singularity after closure | Type $(a, b, c, d)$ |
|---|---|---|---|---|
| $A_{2N}$ | $\mathbb{Z}_2$ | $4N + 2$ | $u^2 + x^{2N+1} + y^{\kappa-N} + yz^2 = 0$ | $\{2,1\}(2, 2N+1, \kappa-N, 2)$ |
| | | $2N$ | $u^2 + x^{2N+1} + xy^{\kappa-N} + yz^2 = 0$ | $\{7,1\}(2, 2N+1, \kappa-N, 2)$ |
| $A_{2N-1}$ | $\mathbb{Z}_2$ | $2N$ | $u^2 + x^{\kappa-N-1} + xy^2 + yz^N = 0$ | $\{7,1\}(2, \kappa-N-1, 2, N)$ |
| | | $4N-2$ | $u^2 + xy^{\kappa-N} + yz^2 + zx^N = 0$ | $\{10,1\}(2, \kappa-N, N, 2)$ |
| $D_{N+1}$ | $\mathbb{Z}_2$ | $2N$ | $u^2 + x^{\kappa+1-2N} + xy^N + yz^2 = 0$ | $\{7,1\}(2, \kappa+1-2N, N, 2)$ |
| | | $2N+2$ | $u^2 + xy^2 + yz^{\kappa-N} + zx^N = 0$ | $\{10,1\}(2, 2, N, \kappa-N)$ |
| $D_4$ | $\mathbb{Z}_3$ | $12$ | $u^2 + x^3 + yz^3 + zy^{\kappa-2} = 0$ | $\{3,1\}/(2, 3, 3, \kappa-2)$ |
| | | $12$ | $u^2 + x^3 + yz^3 + xy^{\kappa-3} = 0$ | $\{7,1\}(2, 3, \kappa-3, 3)$ |
| | | $6$ | $u^2 + x^3 + yz^3 + y^{\kappa-4} = 0$ | $\{2,1\}(2, 3, \kappa-4, 3)$ |
| $E_6$ | $\mathbb{Z}_2$ | $18$ | $u^2 + x^3 + yz^4 + zy^{\kappa-6} = 0$ | $\{3,1\}(2, 3, 4, \kappa-6)$ |
| | | $12$ | $u^2 + x^3 + yz^4 + y^{\kappa-9} = 0$ | $\{2,1\}(2, 3, \kappa-9, 4)$ |
| | | $8$ | $u^2 + x^3 + yz^4 + xy^{\kappa-6} = 0$ | $\{7,1\}(2, 3, \kappa-6, 4)$ |

column of table 3. Furthermore, in the last column of the table we include the singularity type in the notation of [55].

The defining equation of the hypersurface is derived as follows: We assume that, as in the untwisted case, the singularity is of the form $u^2 + F(x, y, z) = 0$ and then we determine the explicit form of $F$ by matching the Coulomb Branch (CB) spectra of the two theories. This will now be exemplified for a few twisted irregular punctures.

Before entering the details of the derivation, let us remind the reader how to determine CB operators and physical parameters of a $\mathcal{N} = 2$ SCFT geometrically engineered in Type IIB string theory. The procedure exploits the fact that (extended) CB moduli are associated with complex structure deformations of the geometry and therefore what we need to do is to deform the hypersurface singularity and determine the scaling dimension of the corresponding parameters. This information is extracted via the following procedure [78]:

1. We impose the normalization condition that the holomorphic three-form $\Omega_3$ has dimension one. This must be required since its periods compute the mass of BPS states in the theory.

2. We require that all the terms appearing in the equation describing the deformed singularity have the same dimension.

Once we have the full list of parameters together with their scaling dimension is known, we can exploit the rule that parameters whose dimension is smaller than one correspond to coupling constants of the theory, those with dimension exactly one are mass parameters and those with dimension larger than one describe the expectation value of Coulomb branch operators[14]. We therefore see that we can extract the CB spectrum of the theory with this method.

---
[14]There is potentially an exception to this rule in the case of hypersurfaces in $\mathbb{C}^3 \times \mathbb{C}^*$, since some parameters

## 6.3 Extracting the IHS for twisted $A_3$

Since the procedure is rather involved, let us start by illustrating our method with a specific example, namely a $A_3$ twisted theory described by the Type IIB geometry

$$W(u,v,x,z) \equiv uv + x^4 + z^\kappa + \text{subleading} = 0; \quad \Omega_3 = \frac{du \wedge dv \wedge dx \wedge dz}{z \, dW}. \tag{176}$$

The subleading terms encode all the physical parameters of the theory and can be uniformly described as $z$-dependent versal deformations of the ADE singularity (in this case $A_3$). When the corresponding Casimir is invariant under the action of the outer-automorphism, the subleading terms are proportional to integer powers of $z$; otherwise they are proportional to a fractional (half-integer in all cases apart from $\mathbb{Z}_3$ twisted $D_4$) power of $z$. In the case at hand the Casimirs of degree 2 and 4 lead therefore to terms of the form $x^2 z^2 u_{2,n}$ and $z^n u_{4,n}$ respectively, whose scaling dimensions are

$$D(u_{2,n}) = 2 - n\frac{4}{\kappa}; \quad D(u_{4,n}) = 4 - n\frac{4}{\kappa}, \tag{177}$$

from (176). The cubic Casimir instead leads to terms of the form $x z^{n-1/2} u_{3,n}$ and their dimension is

$$D(u_{3,n}) = 3 - \left(n - \frac{1}{2}\right)\frac{4}{\kappa}. \tag{178}$$

From (176) we see that allowing all terms with $n$ a positive integer, the degree $k$ differentials appearing in the spectral curve have at $z = 0$ a pole of order $1, \frac{5}{2}, 3$ for $k = 2, 3, 4$ respectively. The leading singular terms are those with $n = 0$. Let us now explain how the spectrum changes upon closure of the regular puncture.

Since for a minimal twisted puncture the pole degrees of the $k$-differentials are reduced to $1, \frac{1}{2}, 2$ (see [71]), we conclude that we need to remove $u_{3,1}, u_{3,2}$ and $u_{4,1}$. This is not the end of the story though, since we also need to take into account constraints: It is known that for each k-differential a puncture can introduce a constraint which can be either of a-type or of c-type. The latter says that the leading singular term is actually the product of other terms appearing in the spectral curve, therefore reducing by one the number of independent parameters. A constraint of a-type instead tells us that the leading term is the square of a more fundamental object. Full punctures never exhibit constraints however, a minimal twisted $A_3$ puncture has a c-constraint for $k = 4$. This tells us that $u_{4,2}$ is not an independent operator and therefore we can discard it from the spectrum of the theory. If it had been a constraint of type a we should have replaced $u_{4,2}$, whose dimension is $4 - 8/\kappa$, with another of dimension $2 - 4/\kappa$. Overall, upon closure of the puncture we find the following spectrum from the quadratic differential

$$2 - \frac{4n}{\kappa} \quad (n \ge 1), \tag{179}$$

and from cubic and quartic differentials

$$3 - \frac{4n-2}{\kappa}; \quad 4 - \frac{4n}{\kappa} \quad (n \ge 3). \tag{180}$$

As we have explained before, parameters with dimension larger than one are Coulomb branch operators, those with dimension exactly one are mass parameters and the others are coupling constants.

---

with dimension larger than one might actually correspond to mass parameters rather than CB operators. These can be singled out by looking at the behaviour of $\Omega_3$, since residues for the holomorphic three-form always correspond to mass parameters and not CB operators, regardless of their dimension.

From these data we will now try to guess the IHS equation, assuming, as we have mentioned before, that the general structure is

$$W(u,x,y,z) \equiv u^2 + F(x,y,z) = 0; \quad \Omega_3 = \frac{du \wedge dx \wedge dy \wedge dz}{dW}. \tag{181}$$

A useful observation at this stage is that the CB operator of largest dimension is $u_{4,3}$. Since for a singularity of the form (181) a constant term (i.e. which does not depend on any of the four coordinates) is always an allowed deformation and the corresponding parameter is clearly the CB operator of highest dimension, we identify it with $u_{4,3}$ and therefore we conclude that $D(u) = \frac{D(u_{4,3})}{2} = 2 - \frac{6}{\kappa}$. We can also notice that the parameters coming from a given $k$-differential have scaling dimension spaced by $4/m$ and therefore it is natural to guess that one of the coordinates (say $x$) has precisely that dimension. We can further guess the dimension of $y$ by noticing that $D(u_{4,n}) - D(u_{3,n}) = D(u_{3,n}) - D(u_{2,n-2})$ for every $n$ and therefore the most natural option is that the corresponding terms in the IHS equation are obtained by gradually increasing the power of one of the coordinates, whose dimension is therefore necessarily $D(u_{4,n}) - D(u_{3,n}) = 1 - \frac{2}{\kappa}$. Now that we have a guess for the scaling dimension of all the coordinates except $z$, we can notice that requiring $\Omega_3$ to have dimension one we find the equation

$$D(x) + D(y) + D(z) = 1 + D(u), \tag{182}$$

from which we conclude that $D(z) = 2 - \frac{8}{\kappa}$. The last step is to construct a homogeneous singularity of the form (181) compatible with our prediction for the scaling dimensions. We find that

$$F = x^{\kappa-3} + xz^2 + zy^2 \tag{183}$$

does the job. We can indeed check that the spectrum derived from this IHS reproduces (179) and (180). We therefore recover the result reported in Table 3.

## 6.4 IHS for twisted irregular punctures

The strategy we will follow to identify the IHS equations is to implement for all cases the analysis presented in 6.3 for the $A_3$ case. We will now review for each case the relevant properties of twisted punctures and then explain how to construct the relevant IHS singularity. We have checked in all cases that the CB spectrum of the IHS reproduces that of the class $\mathcal{S}$ theory on the sphere with minimal and irregular twisted punctures.

### 6.4.1 Twisted $A_{\mathbf{odd}}$ theories

In this case all Casimirs of odd degree change sign under the action of the outer-automorphism, therefore all the k-differentials with k odd are proportional to half-integer powers of $z$ and those with k even are proportional to integer powers of $z$. The properties of twisted $A_{odd}$ regular punctures were derived in [71]. The full puncture introduces at $z = 0$ poles of order

$$\frac{3k}{2} - \left\lfloor \frac{k}{2} \right\rfloor - 1; \quad k = 2, \ldots, 2N, \tag{184}$$

where $\lfloor \ \rfloor$ denotes the integer part. It turns out that there are no a-type constraints for the minimal twisted puncture. Since we are only interested in determining the spectrum of the theory, we can directly consider the combined effect of c-type constraints and the different pole orders. Overall, the order of poles minus the number of c-constraints is

$$\frac{k}{2} - \left\lfloor \frac{k-1}{N} \right\rfloor; \quad k = 2, \ldots, 2N. \tag{185}$$

Therefore, starting from the SW geometry of the theory with a full twisted puncture, if we remove from the spectrum the first $k - 1 - \lfloor \frac{k}{2} \rfloor + \lfloor \frac{k-1}{N} \rfloor$ for every k (those with largest scaling dimension as in 6.3) we find the spectrum of the theory with a minimal puncture.

**Models with $b_t = 2N$.**   The relevant SW geometry is

$$uv + x^{2N} + z^\kappa + \text{subleading} = 0 \tag{186}$$

and generalizes the $A_3$ family we have discussed before. As in 6.3 we expect in the IHS one coordinate with dimension $\frac{2N}{\kappa}$ since this is the spacing between operators from each k-differential. We also expect a second coordinate which allows us to go e.g. from the CB operator of largest dimension in $\phi_k$ to the one in $\phi_{k-1}$. This has dimension $1 - \frac{N}{\kappa}$. Finally, since $\Delta_{max}$ (the highest dimension in the CB spectrum) after closure becomes $2N - \frac{2N^2 + 2N}{\kappa}$ we conclude that the third coordinate has dimension $N - \frac{N^2 + N}{\kappa}$. Finally, using the normalization condition $D(\Omega_3) = 1$ we find that the fourth coordinate has dimension $N - \frac{N^2 + 2N}{\kappa}$. Using these data we find as desired the IHS singularity

$$u^2 + x^{\kappa - N - 1} + xy^2 + yz^N = 0. \tag{187}$$

**Models with $b_t = 4N - 2$.**   The relevant SW geometry is

$$uv + x^{2N} + xz^{\kappa + 1/2} + \text{subleading} = 0. \tag{188}$$

The spacing between operators from the same k-differential is always a multiple of $\frac{4N-2}{2\kappa+1}$ and this quantity will therefore become the dimension of one of the coordinates. Interpolating between differentials of degree $k$ and $k-1$ requires instead a coordinate with dimension $1 - \frac{2N-1}{2\kappa+1}$. The coordinate $u$ will have dimension $\Delta_{max}/2$, which in the case at hand is equal to $N - \frac{(2N-1)(N+1)}{2\kappa+1}$. Finally, the normalization condition $D(\Omega_3) = 1$ implies that the fourth coordinate has dimension $N - \frac{(2N-1)(N+2)}{2\kappa+1}$. From these results we find the IHS

$$u^2 + xy^{\kappa - N} + yz^2 + zx^N = 0. \tag{189}$$

### 6.4.2   Twisted $A_{\text{even}}$ theories

Also in this case the outer-automorphism changes the sign of $k$-differentials with $k$ odd. The corresponding monomials will therefore involve half-integer powers of the coordinate $z$. Unfortunately the detailed data of twisted $A_{even}$ punctures are not known at present, but we have sufficient information to determine how the spectrum changes upon closure of the regular puncture [106]. The change in pole orders and the implementation of c-type constraints[15] instructs us to remove for each $k$-differential the $\lfloor \frac{k}{2} \rfloor - 1$ terms with largest dimension from the spectrum of the theory with a full puncture. Furthermore, there is a constraint of a-type whenever $k$ is odd and therefore we should modify the spectrum accordingly.

Let us give an example of this procedure since this is the first time we come across a-type constraints. Let us consider the case $k = 5$ (and $b_t = 2N$ for definiteness). The corresponding operators in the theory with a full puncture have dimension

$$-\frac{2N}{\kappa}, 5 - \frac{4N}{\kappa}, 5 - \frac{6N}{\kappa}, \dots . \tag{190}$$

Since $\lfloor \frac{k}{2} \rfloor - 1$ is 1 for $k = 5$, in order to implement the closure we first remove the operator with largest dimension, therefore the first in the sequence above and only afterwards we implement the a-constraint and trade the second operator for another with the same dimension divided by two. We therefore get an operator with dimension $\frac{5}{2} - \frac{2N}{\kappa}$.

---

[15]It is not known at present how to disentangle these two pieces of information.

**Models with $b_t = 4N + 2$.**   The relevant SW geometry is

$$uv + x^{2N+1} + z^{\kappa+1/2} + \text{subleading} = 0. \tag{191}$$

The spacing between operators from the same $k$-differential is a multiple of $\frac{4N+2}{2\kappa+1}$ which becomes the dimension of one of the coordinates. Moving between differentials with consecutive degree ($k$ and $k-1$) requires instead a coordinate with dimension $1-\frac{2N+1}{2\kappa+1}$. The coordinate $u$ has dimension $\Delta_{max}/2$, namely $N+\frac{1}{2}-\frac{(2N+1)^2}{4\kappa+2}$. Finally, the normalization condition $D(\Omega_3) = 1$ implies that the fourth coordinate has dimension $N + \frac{1}{2} - \frac{(2N+1)^2}{4\kappa+2} - \frac{2N+1}{2\kappa+1}$. From these results we find the IHS

$$u^2 + x^{2N+1} + y^{\kappa-N} + yz^2 = 0. \tag{192}$$

**Models with $b_t = 2N$**   The relevant SW geometry is

$$uv + x^{2N+1} + xz^{\kappa} + \text{subleading} = 0. \tag{193}$$

The spacing between operators from the same $k$-differential is a multiple of $\frac{2N}{\kappa}$ which becomes the dimension of one of the coordinates. Moving between differentials with consecutive degree ($k$ and $k-1$) requires a coordinate with dimension $1-\frac{N}{\kappa}$. The coordinate $u$ has dimension $\Delta_{max}/2$, namely $N+\frac{1}{2}-\frac{2N^2+N}{2\kappa}$. Finally, the normalization condition $D(\Omega_3) = 1$ implies that the fourth coordinate has dimension $N + \frac{1}{2} - \frac{2N^2+3N}{2\kappa}$. From these results we find the IHS

$$u^2 + x^{2N+1} + xy^{\kappa-N} + yz^2 = 0. \tag{194}$$

### 6.4.3   Twisted $D_{N+1}$ theories

The class $\mathcal{S}$ description includes $k$-differentials with $k$ even from 2 to 2N plus another differential with $k = N + 1$ associated with the pfaffian of the Hitchin field. For this family of theories the outer-automorphism acts nontrivially (changing the sign) of the pfaffian only. The corresponding monomials will therefore involve half-integer powers of the coordinate $z$. In this case both the full and minimal punctures do not exhibit any constraints [69] and therefore we just need to know the pole orders. For the differentials $(\phi_2, \phi_4, \ldots, \phi_{2N}; \phi_{N+1})$ the pole orders for the full puncture are $(1, 3, \ldots, 2N-1; \frac{2N+1}{2})$ whereas for the minimal puncture we have $(1, 1, \ldots, 1; \frac{1}{2})$. The number of operators we need to remove from each $k$-differential to implement the closure is therefore $(0, 2, 4, \ldots, 2N-2; N)$. Let us now discuss the two cases separately.

**Models with $b_t = 2N + 2$.**   The relevant SW geometry is

$$uv + x^{N} + xy^2 + yz^{\kappa+1/2} + \text{subleading} = 0. \tag{195}$$

The spacing between operators from the same $k$-differential is a multiple of $\frac{2N+2}{2\kappa+1}$ which becomes the dimension of one of the coordinates. Moving between differentials with consecutive degree ($k$ and $k-2$) requires instead a coordinate with dimension $2-\frac{4N+4}{2\kappa+1}$. The coordinate $u$ has dimension $\Delta_{max}/2$, namely $N-\frac{2N^2+N-1}{2\kappa+1}$. Finally, the normalization condition $D(\Omega_3) = 1$ implies that the fourth coordinate has dimension $N-1-\frac{2N^2+N-1}{2\kappa+1} + \frac{2N+2}{2\kappa+1}$. From these results we find the IHS

$$u^2 + xy^2 + yz^{\kappa-N} + zx^N = 0. \tag{196}$$

**Models with $b_t = 2N$.** The relevant SW geometry is

$$u^2 + x^N + xy^2 + z^\kappa + \text{subleading} = 0. \tag{197}$$

The spacing between operators from the same $k$-differential is a multiple of $\frac{2N}{\kappa}$ which becomes the dimension of one of the coordinates. Moving between differentials with consecutive degree ($k$ and $k-2$) requires a coordinate with dimension $2 - \frac{4N}{\kappa}$. The coordinate $u$ has dimension $\Delta_{max}/2$, namely $N - \frac{2N^2-N}{\kappa}$. Finally, the normalization condition $D(\Omega_3) = 1$ implies that the fourth coordinate has dimension $N - 1 - \frac{2N^2-3N}{\kappa}$. From these results we find the IHS

$$u^2 + x^{\kappa+1-2N} + xy^N + yz^2 = 0. \tag{198}$$

### 6.4.4 $\mathbb{Z}_3$-twisted $D_4$ theories

This is the only case in which the outer-automorphism is not $\mathbb{Z}_2$. There are a quadratic and a sextic differentials which are invariant under outer-automorphisms and two quartic differentials which transform as $\phi_4 \to \omega \phi_4$ and $\tilde{\phi}_4 \to \omega^{-1} \tilde{\phi}_4$ with $\omega^3 = 1$. This implies that all terms in $\phi_2$ and $\phi_6$ contain integer powers of $z$, whereas $\phi_4$ contains terms of the form $z^{n+1/3}$ and $\tilde{\phi}_4$ contains terms of the form $z^{n+2/3}$. The pole orders (see [63]) for $(\phi_2, \phi_4, \tilde{\phi}_4, \phi_6)$ at the full puncture are $(1, 10/3, 11/3, 5)$ and at the minimal puncture are $(1, 7/3, 8/3, 4)$. In the case of the minimal puncture we have a constraint of a-type for $\phi_4$ and a c-constraint for $\tilde{\phi}_4$. There are instead two constraints of c-type for $\phi_6$.

**Models with $b_t = 12$.** The relevant SW geometry is

$$u^2 + x^3 + xy^2 + yz^{\kappa\pm1/3} + \text{subleading} = 0. \tag{199}$$

The spacing between operators from the same $k$-differential is a multiple of $\frac{12}{3\kappa\pm1}$, which becomes the dimension of one of the coordinates. The new feature in this case is that the sign ambiguity leads to two different families of singularities. Moving between the differentials $\phi_6$ and $\phi_4$ requires a coordinate with dimension $2 - \frac{20}{3\kappa\pm1}$ and going instead from $\phi_6$ to $\tilde{\phi}_4$ leads to a coordinate with dimension $2 - \frac{16}{3\kappa\pm1}$. The coordinate $u$ has as always dimension $\Delta_{max}/2$, i.e. $3 - \frac{24}{3\kappa\pm1}$. Notice that at this stage we have already determined the scaling dimensions of all the coordinates and therefore the normalization condition $D(\Omega_3) = 1$ should be automatically satisfied for the consistency of our picture. It is satisfactory to see this is indeed the case. From these assignments of dimensions we find the following two families of singularities:

$$u^2 + x^3 + yz^3 + zy^{\kappa-2} = 0, \tag{200}$$

when we choose the $+$ sign and

$$u^2 + x^3 + yz^3 + xy^{\kappa-3} = 0, \tag{201}$$

when we choose the $-$ sign.

**Models with $b_t = 6$.** The relevant SW geometry is

$$u^2 + x^3 + xy^2 + z^\kappa + \text{subleading} = 0. \tag{202}$$

We have no sign ambiguity this time, but as in the $b_t = 12$ case we will be able to identify the scaling dimension of all the coordinates without exploiting the normalization condition. The spacing between operators from the same $k$-differential is a multiple of $\frac{6}{\kappa}$, which therefore becomes the dimension of one of the coordinates. Moving between the differentials $\phi_6$ and

$\phi_4$ requires a coordinate with dimension $2 - \frac{10}{\kappa}$ and going instead from $\phi_6$ to $\tilde{\phi}_4$ leads to a coordinate with dimension $2 - \frac{8}{\kappa}$. The dimension of $u$, which is always equal to $\Delta_{max}/2$, in this case reads $3 - \frac{12}{\kappa}$. Again the normalization condition $D(\Omega_3) = 1$ is automatically satisfied. From these assignments of dimensions we find the following family of singularities:

$$u^2 + x^3 + yz^3 + y^{\kappa-4} = 0. \tag{203}$$

### 6.4.5 Twisted $E_6$ theories

In the $E_6$ class $\mathcal{S}$ theory we have $k$-differentials with $k = 2, 5, 6, 8, 9, 12$ and only those with $k = 5, 9$ are odd under the action of the outer-automorphism. We therefore conclude that terms in $\phi_5$ and $\phi_9$ are proportional to half-integer powers of $z$ whereas all the others are proportional to integer powers of $z$. Finally, let us consider the properties of the full and minimal punctures [66]. The pole orders for $(\phi_2, \phi_5, \phi_6, \phi_8, \phi_9, \phi_{12})$ are $(1, 9/2, 5, 7, 17/2, 11)$ for the full puncture and $(1, 5/2, 3, 4, 9/2, 6)$ for the minimal. Moreover, the minimal puncture exhibits one constraint of a-type for $\phi_6$ and several c-type constraints: one for $\phi_5$, two for $\phi_8$ and $\phi_9$ and three for $\phi_{12}$.

**Models with $b_t = 18$.** The relevant SW geometry is

$$u^2 + x^3 + y^4 + yz^{\kappa+1/2} + \text{subleading} = 0. \tag{204}$$

The spacing between operators from the same $k$-differential is a multiple of $\frac{18}{2\kappa+1}$, which therefore becomes the dimension of one of the coordinates. Moving between the differentials $\phi_{12}$ and $\phi_9$ requires a coordinate with dimension $3 - \frac{45}{2\kappa+1}$[16] and going instead from $\phi_{12}$ to $\phi_8$ leads to a coordinate with dimension $4 - \frac{54}{2\kappa+1}$. Finally, in this case $\Delta_{max} = 12 - \frac{162}{2\kappa+1}$ and therefore the dimension of $u$ is equal to $6 - \frac{81}{2\kappa+1}$. Again the normalization condition $D(\Omega_3) = 1$ is automatically satisfied. From these assignments of dimensions we find the singularity

$$u^2 + x^3 + yz^4 + zy^{\kappa-6} = 0. \tag{205}$$

**Models with $b_t = 12$.** The relevant SW geometry is

$$u^2 + x^3 + y^4 + z^\kappa + \text{subleading} = 0. \tag{206}$$

The spacing between operators from the same $k$-differential is a multiple of $\frac{12}{\kappa}$, which therefore becomes the dimension of one of the coordinates. Moving between the differentials $\phi_{12}$ and $\phi_9$ requires a coordinate with dimension $3 - \frac{30}{\kappa}$ and going instead from $\phi_{12}$ to $\phi_8$ leads to a coordinate with dimension $4 - \frac{36}{\kappa}$. Finally, in this case $\Delta_{max} = 12 - \frac{108}{\kappa}$ and therefore the dimension of $u$ is equal to $6 - \frac{54}{\kappa}$. Again the normalization condition $D(\Omega_3) = 1$ is automatically satisfied. From these assignments of dimensions we find the singularity

$$u^2 + x^3 + yz^4 + y^{\kappa-9} = 0. \tag{207}$$

**Models with $b_t = 8$.** The relevant SW geometry is

$$u^2 + x^3 + y^4 + xz^\kappa + \text{subleading} = 0. \tag{208}$$

The spacing between operators from the same $k$-differential is a multiple of $\frac{8}{\kappa}$, which therefore becomes the dimension of one of the coordinates. Moving between the differentials $\phi_{12}$ and

---

[16]We can also notice, as a further check of our method, that this assignment of scaling dimension also allows us to recover from the term with highest dimension in $\phi_6$ the term generated by the a-type constraint.

$\phi_9$ requires a coordinate with dimension $3 - \frac{20}{\kappa}$ and going instead from $\phi_{12}$ to $\phi_8$ leads to a coordinate with dimension $4 - \frac{24}{\kappa}$. In this case $\Delta_{max} = 12 - \frac{72}{\kappa}$ and therefore the dimension of $u$ is equal to $6 - \frac{36}{\kappa}$. The normalization condition $D(\Omega_3) = 1$ is automatically satisfied as in the previous cases and from these assignments of dimensions we find the singularity

$$u^2 + x^3 + yz^4 + xy^{\kappa-6} = 0. \tag{209}$$

## 6.5 IHS for Trinions

As is well known, the class S formalism does not allow to describe the gauging of more than two matter sectors, in particular it does not provide a realization of unitary quivers with exceptional shape. This is not a restriction for geometric engineering and indeed IHS descriptions in Type IIB for quivers with exceptional shape are known [76]. This is just a special case of systems (which we call trinions) involving the gauging of three $D_p^b(G)$ theories through a $G$ vector multiplet.

A general procedure to construct IHS descriptions for trinions of $D_p^b(G)$ theories with $G$ simply-laced was proposed in [29], and we will now briefly review it. With this result at hand, we can use the Type IIB description to determine the defect group of the trinion theory, therefore providing a highly nontrivial check of our results. Notice that this computation is sensitive to the non trapped part of the defect group.

### 6.5.1 Trinions of $D_p^b(G)$ Theories from Type IIB

In the general case of gaugings of three $D_p^b(G)$ theories, we just have a Landau-Ginzburg (LG) description instead of a threefold singularity. The LG superpotential reads

$$\mathcal{W} = w^2 + f(t, x) + \sum_{i=1}^{3} z_i^{p_i} f_{b_i}(x, t), \tag{210}$$

where $W_G(w, x, t) \equiv w^2 + f(t, x) = 0$ is the ADE singularity of type $G$ and $f_{b_i}(x, t) z_i^{p_i}$ denotes the $z$-dependent part of the threefold singularity describing $D_{p_i}^{b_i}(G)$, as in table 3. From a LG model point of view, $w$ in (210) is a massive field and can be integrated out, leaving five fields. The conformality condition reads

$$\frac{b_1}{p_1} + \frac{b_2}{p_2} + \frac{b_3}{p_3} = h^\vee(G). \tag{211}$$

Importantly, if the singularity $\mathcal{W} = 0$ is non-isolated, one needs to add marginal terms to (210) in order to have an isolated singularity at the origin, so that the LG model is well-defined (see [29] for a detailed discussion on this point).

We will focus on the special case in which the LG model is equivalent to a system with at most four fields, since this is the class of theories for which the LG superpotential reduces to the defining equation of a IHS. This is always the case when $G$ is special unitary since $W_G = w^2 + t^2 + x^N$ and therefore also $t$ is massive and can be integrated out, leaving us with the four fields $x, z_1, z_2, z_3$. We then conclude that a trinion involving the gauging of $D_{p_i}^{b_i}(SU(N))$ for $i = 1, 2, 3$ is described by the IHS

$$x^N + x^{N-b_1} z_1^{p_1} + x^{N-b_2} z_2^{p_2} + x^{N-b_3} z_3^{p_3} = 0. \tag{212}$$

The case in which $G$ is special orthogonal or exceptional is more subtle since we cannot integrate out either $t$ or $x$. In order to reduce to a system with four fields, we should restrict

to cases in which at least one of the fields $z_i$ is massive, so that it can be integrated out. This happens whenever

$$z^p f_b(x,t) = z^2 \qquad \text{or} \qquad z^p f_b(x,t) = tz, \; xz, \tag{213}$$

for $p = 2$ or $p = 1$ and an appropriate choice of $b$. Such massive fields can be integrated out. If we choose $p = 2$ for one of the legs, we are considering a trinion of the form

$$
\begin{array}{c}
D_2^{h^\vee}(G) \\
| \\
D_{p_2}^{b_2}(G) \;\text{------}\; G \;\text{------}\; D_{p_3}^{b_3}(G)
\end{array}
\tag{214}
$$

The constraint (211) now implies that the other two $D_p^b(G)$ sectors should satisfy the relation

$$\frac{b_2}{p_2} + \frac{b_3}{p_3} = \frac{h^\vee(G)}{2}. \tag{215}$$

Since we are now left with only four fields, the system can be understood as a threefold compactification in Type IIB. The second option is $p = 1$, which gives a mass term (213) for $b \neq h^\vee(G)$, since in that case the $f_b(x,t)$ function is always linear in either $x$ or $t$, we can simply choose the $D_{p_1}^{b_1}(G)$ theory to be $D_1^b(G)$ with $b \neq h^\vee(G)$ (we remind the reader that $D_1^b(G)$ is trivial for $b = h^\vee(G)$) and, due to (211), we take the other two $D_p^b(G)$ models to satisy the constraint

$$\frac{b_2}{p_2} + \frac{b_3}{p_3} = h^\vee(G) - b. \tag{216}$$

Under these conditions, we can integrate out both $z_1$ and either $x$ or $t$ (the variable appearing in $f_{b_1}(x,t)$). We can now simply introduce a new massive field $y$ which enters quadratically in the superpotential. The threefold singularity is now given by a hypersurface in the four variables $z_2, z_3, y$, and $x$ or $t$ (the variable we have not integrated out). We can therefore also consider the family of trinions

$$
\begin{array}{c}
D_1^b(G) \\
| \\
D_{p_2}^{b_2}(G) \;\text{------}\; G \;\text{------}\; D_{p_3}^{b_3}(G)
\end{array}
\tag{217}
$$

satisfying the condition (216).

### 6.5.2  The Defect Group of Trinions from Punctures

Let us consider general 4d $\mathcal{N} = 2$ SCFTs of the form

$$
\begin{array}{c}
\mathfrak{T}_{\mathcal{P}_1}^* \\
| \\
\mathfrak{T}_{\mathcal{P}_2}^* \;\text{------}\; \mathfrak{g} \;\text{------}\; \mathfrak{T}_{\mathcal{P}_3}^*
\end{array},
\tag{218}
$$

where $\mathcal{P}_i$ are *untwisted* conformal punctures, so that each $\mathfrak{T}_{\mathcal{P}_i}^*$ is a matter SCFT [29]. Moreover, $\mathfrak{T}_{\mathcal{P}_i}^*$ carries a $\mathfrak{g}$ flavor symmetry associated to the untwisted maximal regular puncture used in

the Class S construction of $\mathfrak{T}^*_{\mathcal{P}_i}$. In the trinion theory (218), this $\mathfrak{g}$ flavor symmetry of all three $\mathfrak{T}^*_{\mathcal{P}_i}$ is gauged by a single gauge algebra $\mathfrak{g}$ in the center of the trinion.

We can compute the defect group $\mathcal{L}_{\mathcal{P}_1,\mathcal{P}_2,\mathcal{P}_3}$ of the above trinion theory in terms of defect groups $\mathcal{L}_{\mathcal{P}_i}$ associated to the punctures $\mathcal{P}_i$. This is easily found to be

$$\mathcal{L}_{\mathcal{P}_1,\mathcal{P}_2,\mathcal{P}_3} = \mathcal{H}_{\mathcal{P}_1,\mathcal{P}_2,\mathcal{P}_3} \times \widehat{\mathcal{H}}_{\mathcal{P}_1,\mathcal{P}_2,\mathcal{P}_3} \,, \tag{219}$$

with

$$\mathcal{H}_{\mathcal{P}_1,\mathcal{P}_2,\mathcal{P}_3} = \left\{ (\alpha,\beta,\gamma) \in \mathcal{H}_{\mathcal{P}_1} \times \mathcal{H}_{\mathcal{P}_2} \times \mathcal{H}_{\mathcal{P}_3} \middle| \pi_{\mathcal{P}_1}(\alpha) = \pi_{\mathcal{P}_2}(\beta) = \pi_{\mathcal{P}_3}(\gamma) \right\} \tag{220}$$

and the pairing provided by the pairing of $\widehat{\mathcal{H}}_{\mathcal{P}_1,\mathcal{P}_2,\mathcal{P}_3}$ with $\mathcal{H}_{\mathcal{P}_1,\mathcal{P}_2,\mathcal{P}_3}$. In fact, we find that for all the trinions that can be realized by IHS, the short exact sequence (57) splits for each participating puncture $\mathcal{P}_i$, which allows us to simplify $\mathcal{H}_{\mathcal{P}_1,\mathcal{P}_2,\mathcal{P}_3}$ as

$$\mathcal{H}_{\mathcal{P}_1,\mathcal{P}_2,\mathcal{P}_3} = \prod_{i=1}^{3} \mathcal{H}^T_{\mathcal{P}_i} \times Z_{\mathcal{P}_1,\mathcal{P}_2,\mathcal{P}_3} \,, \tag{221}$$

where

$$Z_{\mathcal{P}_1,\mathcal{P}_2,\mathcal{P}_3} = Z_{\mathcal{P}_1} \cap Z_{\mathcal{P}_2} \cap Z_{\mathcal{P}_3} \,. \tag{222}$$

# 7 Untwisted A

In this section, we describe the first of our proposals for the defect groups associated to punctures. We discuss untwisted punctures of 6$d$ A-type $(2,0)$ theories. The key information that we use about these punctures is discussed in subsection 7.1. We not only consider conformal punctures of IHS type, but also numerous other classes of conformal punctures. We also include numerous classes of non-conformal punctures.

In subsection 7.2, we make our proposal. For each puncture, we assign a 4$d$ $\mathcal{N} = 2$ quiver gauge theory, and propose that the defect groups associated to the quiver gauge theory capture the defect groups of the associated punctures. See section 5.2. Using this proposal, we explicitly computed the defect groups associated to all punctures considered in subsection 7.1. As a result of this proposal, untwisted punctures of A-type do not carry any trapped 1-form symmetries (but can carry non-trapped parts).

In subsection 7.3, we expand on our motivation for making the proposal discussed in subsection 7.2. We describe a large sub-class of punctures that admits constructions in terms of Hanany-Witten brane setups in IIA. The IIA construction implies that, for a puncture $\mathcal{P}$ in this subclass, the 4$d$ $\mathcal{N} = 2$ theory $\mathfrak{T}^{\mathcal{P}_0}_{\mathcal{P}}$ associated to $\mathcal{P}$ is precisely the quiver gauge theory assigned to $\mathcal{P}$. Hence, the defect group $\mathcal{L}_{\mathcal{P}}$ of the puncture $\mathcal{P}$ must match the defect group of the quiver theory.

In subsection 7.4, we provide a very strong check of our proposal. We compactify the 4$d$ $\mathcal{N} = 2$ theory $\mathfrak{T}^{\mathcal{P}_0}_{\mathcal{P}}$ for an arbitrary puncture $\mathcal{P}$ on a circle, which leads us to a quiver gauge theory in 3$d$. The defect group of $\mathfrak{T}^{\mathcal{P}_0}_{\mathcal{P}}$ must now match the defect group of the 3$d$ quiver theory, and we show that the latter defect group matches the defect group of 4$d$ quiver theory associated to $\mathcal{P}$.

In subsection 7.5, we use the IIB ALE fibration construction of an arbitrary puncture $\mathcal{P}$ to compute the genuine part $\mathcal{L}^0_{\mathcal{P}}$ of the defect group $\mathcal{L}_{\mathcal{P}}$ associated to $\mathcal{P}$. In subsection 7.6, we use the IHS realization to compute the trapped defect groups of IHS punctures, and find that there cannot be any trapped contribution, which is in line with our proposal. The 4$d$ $\mathcal{N} = 2$ theories constructed by the IHS include the well-studied (A,A) type Argyres-Douglas theories, which are known to have trivial 1-form symmetry.

## 7.1 Punctures

In this section, we discuss untwisted punctures of $A_{n-1}$ $\mathcal{N} = (2,0)$ theory. For simplicity and uniformity of presentation, we will specify them using a $\mathfrak{u}(n)$ valued Higgs field, from which the $\mathfrak{su}(n)$ Higgs field can be obtained by removing the center of mass. Place the puncture at $t = 0$ on a complex plane $\mathbb{C}$ with coordinate $t$. Near the puncture, the Higgs field can be decomposed into blocks, which we label as $B_i$. Each block $B_i$ is singular at $t = 0$, with the leading singular piece being

$$B_i = \frac{1}{t^{1+r_i}} A_i + \cdots , \qquad (223)$$

where $A_i$ is a non-singular diagonal matrix and $r_i \geq 0$ is a rational number. We order the blocks such that $r_i \geq r_{i+1}$. Let $s$ be biggest value of $i$ for which $r_i > 0$. For each $i \leq s$, we write

$$r_i = \frac{p_i}{q_i} , \qquad (224)$$

where $1 \leq q_i \leq n$ and $\gcd(p_i, q_i) = 1$. For such a block, the sheets comprising the block are permuted by a $\mathbb{Z}_{q_i}$ monodromy as one encircles the puncture.

It is known [88] that irregular punctures (which in our notation are characterized as the punctures having $s > 0$) lead to $4d$ $\mathcal{N} = 2$ conformal theories only if the compactification manifold is a sphere and if the sphere carries either a single puncture that is irregular or two punctures such that one is irregular and the other is regular. However, not every irregular puncture can be used in this way to obtain a superconformal theory in $4d$. Let us call the irregular punctures that lead to superconformal theories as *conformal punctures*.

Extending the arguments of [88], we propose that an untwisted A-type puncture $\mathcal{P}$ is conformal only if $r_i = r_j$ for all $i, j \in \{1, 2, \cdots, s\}$. These have $0 \leq n_s \leq n/2$. Out of these, the IHS punctures discussed in [81, 88] are the ones for which $n_s = 0, 1$.

## 7.2 1-Form Symmetry from Class $\mathcal{GQ}$

Consider such a puncture $\mathcal{P}$. Define for it

$$n_i := n - \sum_{j=1}^{i} q_j , \qquad (225)$$

for $1 \leq i \leq s$. Then, we claim that $\mathcal{P} \in \mathcal{F}_\tau$ (see section 5.2) with

$$\tau = \quad \mathfrak{su}(n) \; \rule{1cm}{0.4pt} \; \mathfrak{su}(n_1) \; \rule{1cm}{0.4pt} \; \mathfrak{su}(n_2) \; \rule{1cm}{0.4pt} \; \cdots \; \rule{1cm}{0.4pt} \; \mathfrak{su}(n_{s-1}) \; \rule{1cm}{0.4pt} \; [\mathfrak{u}(n_s)] \quad , \qquad (226)$$

where each edge denotes a hyper in bifundamental between the adjacent (gauge or flavor) algebras. From this we can compute

$$\mathcal{L}_{\mathcal{P}}^T = 0 \qquad (227)$$

and

$$\begin{aligned} \mathcal{L}_{\mathcal{P}} &= \mathbb{Z}_g \times \mathbb{Z}_g \qquad \text{if } n_s = 0 , \\ \mathcal{L}_{\mathcal{P}} &= 0 \qquad\qquad\;\; \text{if } n_s \neq 0 , \end{aligned} \qquad (228)$$

with

$$g = \gcd(n, n_1, n_2, \cdots, n_{s-1}) \qquad (229)$$

and the pairing on $\mathcal{L}_{\mathcal{P}} = \mathbb{Z}_g \times \mathbb{Z}_g$ is obtained by regarding the two $\mathbb{Z}_g$ factors as Pontryagin duals of each other. Since $\mathcal{H}_{\mathcal{P}}^T = 0$, we find that

$$Z_{\mathcal{P}} = \mathbb{Z}_g \subseteq \mathbb{Z}_n = Z_G , \qquad (230)$$

for $n_s = 0$, and

$$Z_\mathcal{P} = 0,$$ (231)

for $n_s \neq 0$.

## 7.3 Class $\mathcal{G}\mathcal{Q}'$ – Type IIA Punctures

Consider a puncture $\mathcal{P}'$ having $p_i = 1$ for all $i$ having $r_i > 0$. The 4d $\mathcal{N} = 2$ Class S theory $\mathfrak{T}^*_{\mathcal{P}'}$ associated to $\mathcal{P}'$ admits a Hanany-Witten type brane construction in Type IIA superstring theory [39, 90, 93]

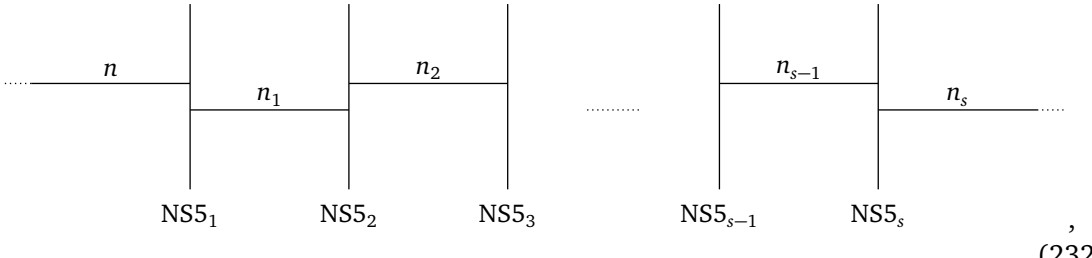

(232)

where the segment between $NS5_i$ and $NS5_{i+1}$ carries $n_i$ D4 branes, and there are $n$ semi-infinite D4 branes on the left and $n_s$ semi-infinite D4 branes on the right. The $n$ semi-infinite D4 branes on the left give rise to the untwisted maximal regular puncture in the Class S construction. All the NS5 branes bend to the right and (along with the other D4 branes) give rise to the irregular puncture $\mathcal{P}'$ in the Class S construction. The resulting 4d theory $\mathfrak{T}^*_{\mathcal{P}'}$ can be identified as the quiver

$$\mathfrak{T}^*_{\mathcal{P}'} = [\mathfrak{su}(n)] - \mathfrak{su}(n_1) - \mathfrak{su}(n_2) - \cdots - \mathfrak{su}(n_{s-1}) - [\mathfrak{u}(n_s)]$$ (233)

and the 4d theory $\mathfrak{T}^{\mathcal{P}_0}_{\mathcal{P}'}$ can be identified as the quiver

$$\mathfrak{T}^{\mathcal{P}_0}_{\mathcal{P}'} = \mathfrak{su}(n) - \mathfrak{su}(n_1) - \mathfrak{su}(n_2) - \cdots - \mathfrak{su}(n_{s-1}) - [\mathfrak{u}(n_s)].$$ (234)

The Hanany-Witten brane construction associated to $\mathfrak{T}^{\mathcal{P}_0}_{\mathcal{P}'}$ is

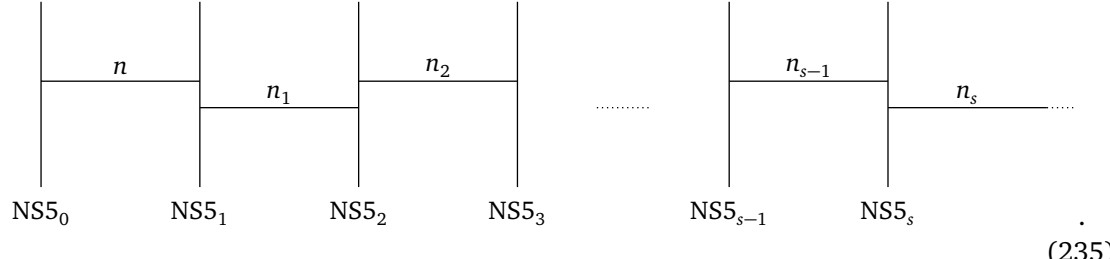

(235)

The $NS5_0$ brane bends to the left and gives rise to the $\mathcal{P}_0$ puncture.

## 7.4 Checks Via Circle Reduction – Electric Quiver $EQ_4$

For a general puncture $\mathcal{P}$, consider the 4d theory $\mathfrak{T}^{\mathcal{P}_0}_{\mathcal{P}}$ and compactify this theory on a circle. A particular zero radius limit of the 4d theory leads to the 3d theory $EQ_4(\mathfrak{T}^{\mathcal{P}_0}_{\mathcal{P}})$ which is the quiver [29]

$$\mathfrak{su}(n) - \cdots - \mathfrak{su}(n_{i-1}) - \mathfrak{u}(n_{i,1}) - \cdots - \mathfrak{u}(n_{i,p_i-1}) - \mathfrak{su}(n_i) - \cdots - [\mathfrak{u}(n_s)],$$ (236)

which is constructed from (226) by inserting between $\mathfrak{su}(n_{i-1})$ and $\mathfrak{su}(n_i)$ algebras, a $p_i - 1$ number of unitary gauge algebras $\mathfrak{u}(n_{i,j})$ with $1 \le j \le p_i - 1$. We have

$$n_{i,j} = \lfloor n_{i-1} - j/r_i \rfloor. \tag{237}$$

The defect group $\mathcal{L}_{\mathcal{P}} = \mathcal{L}_{\mathfrak{T}_{\mathcal{P}}^{\mathcal{P}_0}}$ of lines in the 4d theory $\mathfrak{T}_{\mathcal{P}}^{\mathcal{P}_0}$ should be identified with the defect group formed by lines and local operators in the 3d theory $EQ_4(\mathfrak{T}_{\mathcal{P}}^{\mathcal{P}_0})$. The reader can easily verify from the above quiver description of $EQ_4(\mathfrak{T}_{\mathcal{P}}^{\mathcal{P}_0})$, that the defect group of $EQ_4(\mathfrak{T}_{\mathcal{P}}^{\mathcal{P}_0})$ is given by (228). This provides a strong check for our proposal for computing defect group $\mathcal{L}_{\mathcal{P}} = \mathcal{L}_{\mathfrak{T}_{\mathcal{P}}^{\mathcal{P}_0}}$ as the defect group $\mathcal{L}_{\tau}$ of the quiver $\tau$ shown in (226).

Similarly, the 3d theory $EQ_4(\mathfrak{T}_{\mathcal{P}}^*)$ is the quiver

$$[\mathfrak{su}(n)] \!-\!\!\cdots\!\!-\! \mathfrak{su}(n_{i-1}) \!-\! \mathfrak{u}(n_{i,1}) \!-\!\cdots\!-\! \mathfrak{u}(n_{i,p_i-1}) \!-\! \mathfrak{su}(n_i) \!-\!\cdots\!-\! [\mathfrak{u}(n_s)] , \tag{238}$$

which confirms that $\mathcal{L}_{\mathcal{P}}^T = 0$.

## 7.5 Spectral Cover Monodromies

In this subsection, we derive the defect group (228) of an arbitrary untwisted A-type puncture using the monodromy of the Higgs field near the puncture, under the assumption that the puncture does not carry any trapped part. As discussed earlier, the monodromy of the Higgs field allows one to compute the part

$$\mathcal{L}_{\mathcal{P}}^0 = \mathcal{H}_{\mathcal{P}}^T \times \mathcal{W}_{\mathcal{P}} \subseteq \mathcal{L}_{\mathcal{P}} \tag{239}$$

of the defect group $\mathcal{L}_{\mathcal{P}}$ associated to a puncture $\mathcal{P}$. If one knows via other methods that $\mathcal{H}_{\mathcal{P}}^T = 0$, then the whole defect group $\mathcal{L}_{\mathcal{P}}$ can be recovered from the knowledge of the piece $\mathcal{L}_{\mathcal{P}}^0$ as then one has the relationship

$$\mathcal{L}_{\mathcal{P}} = \left( \mathcal{L}_{\mathcal{P}}^0 \right)^2 . \tag{240}$$

Notice that in this situation we also have $\mathcal{L}_{\mathcal{P}}^0 \cong \widehat{Z}_{\mathcal{P}}$

For this computation, we only need the profile of the spectral curve near the puncture placed at $t = 0$. We work in the frame where the differential associated to the curve is taken to be $\lambda = v \, dt/t$. Then, the profile of the spectral curve very close to the puncture can be described as

$$v^n + \sum_{i=1}^{s} \frac{v^{n_i}}{t^{P_i}} = 0, \tag{241}$$

where

$$P_j := \sum_{i=1}^{j} p_i . \tag{242}$$

This is just the leading behavior of the spectral curve following from the Higgs field singularity (223) describing the puncture.

The sheets of the spectral cover form $s$ number of orbits[17] (under monodromy around the puncture) parametrized by the index $i$. The number of sheets in the orbit $i$ are $q_i$. This means that the integers $p_i$ do not enter the monodromy, and hence the defect group associated to the puncture. Indeed the result (228) depends only on $q_i$ but not on $p_i$.

---

[17]Here we regard all sheets having trivial monodromy to be in the same "orbit" for uniformity of presentation.

Let $e_a$ with $1 \leq a \leq n$ denote various sheets. The first $q_1$ sheets are cyclically permuted amongst themselves, the next $q_2$ sheets are cyclically permuted amongst themselves, and so on. The roots can be expressed as

$$\alpha_a = e_a - e_{a+1},\tag{243}$$

for $1 \leq a \leq n-1$. The above monodromy action on the sheets $e_a$ descends to a monodromy action $M_\mathcal{P}$ on the roots $\alpha_a$. We now compute and find if $n_s = 0$

$$T_\mathcal{P} = M_\mathcal{P} - 1, \qquad \mathrm{SNF}(T_\mathcal{P}) = \mathrm{diag}(g, 1, \ldots, 1, 0, \ldots, 0),\tag{244}$$

where $g = \gcd(n, n_1, \ldots, n_s)$ and we have $s-1$ vanishing entries. With this we compute

$$\mathcal{L}_\mathcal{P}^0 = \mathrm{Tor}\left(\frac{\mathbb{Z}^{n-1}}{T_\mathcal{P}\mathbb{Z}^{n-1}}\right) = \mathbb{Z}_g.\tag{245}$$

When $n_s \neq 0$, we instead find $\mathcal{L}_\mathcal{P}^0 = 0$.

**Example:** $\mathcal{P}_0$ **puncture.** Let us consider the example of $\mathcal{P}_0$ puncture, which has $s = 1, p_1 = 1, q_1 = n$. There is a $\mathbb{Z}_n$ monodromy around the $\mathcal{P}_0$ puncture. That is, all the $n$ sheets are cyclically permuted amongst themselves. We label the sheets by weights $w_i$ of the $n$-dimensional fundamental representation such that these are permuted as $w_i \to w_{i+1}$ with $n + 1 \equiv 1$. We have the roots $\alpha_a = w_a - w_{a+1}$ where $a = 1, \ldots, n-1$. With respect to this labelling the monodromy action on sheets leads to the following action on the roots

$$\alpha_a \to \alpha_{a+1} \quad \forall\, 1 \leq a \leq n-2, \qquad \alpha_{n-1} \to -\sum_{a=1}^{n-1} \alpha_a.\tag{246}$$

The mondromy matrix is thus

$$M_{\mathcal{P}_0} = \begin{pmatrix} 0 & 0 & \cdots & 0 & 0 & 0 & -1 \\ 1 & 0 & \cdots & 0 & 0 & 0 & -1 \\ 0 & 1 & \ddots & 0 & 0 & 0 & -1 \\ \vdots & \ddots & \ddots & \ddots & \vdots & \vdots & \vdots \\ 0 & 0 & \ddots & 1 & 0 & 0 & -1 \\ 0 & 0 & \ldots & 0 & 1 & 0 & -1 \\ 0 & 0 & \ldots & 0 & 0 & 1 & -1 \end{pmatrix}.\tag{247}$$

From here we define

$$T_{\mathcal{P}_0} = M_{\mathcal{P}_0} - 1\tag{248}$$

and find

$$\mathrm{SNF}\left(T_{\mathcal{P}_0}\right) = \mathrm{diag}(n, 1, \ldots, 1),\tag{249}$$

implying that

$$\mathcal{L}_{\mathcal{P}_0}^0 = \mathbb{Z}_n.\tag{250}$$

## 7.6 Trapped Defect Group from Type IIB on CY3

As we discussed earlier, for a puncture $\mathcal{P}$ of IHS type we have an alternate geometric method for computing the trapped part of the defect group $\mathcal{L}_{\mathcal{P}}^{T}$ based on the associated isolated hypersurface singularity $X_{\mathcal{P}}$. The 4d $\mathcal{N} = 2$ SCFT $\mathfrak{T}_{\mathcal{P}}$ associated to an untwisted A-type IHS puncture $\mathcal{P}$ is an AD theory of type A, with the following isolated hypersurface realization:

$$\text{AD}[A_n, A_k]: \qquad x_1^2 + x_2^{n+1} + x_3^2 + x_4^{k+1} = 0. \tag{251}$$

The results in [17, 30, 55] show that for any $k, n$ these have a trivial defect group

$$\mathcal{L}_{X_{\text{AD}[A_n,A_k]}} = 0. \tag{252}$$

Similarly, for untwisted $A_{N-1}$-type IHS punctures $\mathcal{P}_k^b$ having $b = N - 1$ and arbitrary $k$, one finds using the IHS singularity, that the defect group $\mathcal{L}_{\mathfrak{T}_{\mathcal{P}_k^b}}$ of the theory $\mathfrak{T}_{\mathcal{P}_k^b}$ associated to the puncture $\mathcal{P}_k^b$ is trivial. Thus we find that there is no trapped defect group for all IHS punctures of untwisted type A

$$\mathcal{L}_{\mathcal{P}_k^b}^{T} = 0. \tag{253}$$

This is consistent with the result (227) stating that in fact no conformal (IHS or non-IHS) or non-conformal puncture of untwisted A-type carries any trapped defect group.

## 8 Untwisted D

In this section, we describe our proposal for computing the defect groups associated to untwisted punctures for 6d D-type (2, 0) theories. The key information that we use about these punctures is discussed in subsection 8.1. We not only consider conformal punctures of IHS type, but also numerous other classes of conformal and non-conformal punctures.

In subsection 8.2, we make our proposal. For each puncture, we assign a 4d $\mathcal{N} = 2$ generalized quiver gauge theory involving both perturbative and strongly coupled matter sectors, and propose that the defect groups associated to the generalized quiver gauge theory capture the defect groups of the associated punctures.

In subsection 8.2.1, we expand on our motivation for making the proposal discussed in subsection 8.2. We describe a large class of punctures $\mathcal{P}$ for which the 4d $\mathcal{N} = 2$ theories $\mathfrak{T}_{\mathcal{P}}^{\mathcal{P}_0}$ coincide with the proposed generalized quivers. A subclass of this class of punctures, whose associated generalized quivers involve only perturbative matter content, can be realized in terms of Hanany-Witten brane setups in IIA involving O4 planes.

Before using the proposal of subsection 8.2 to compute defect groups associated to untwisted D-type punctures, one needs to understand some properties of the strongly coupled matter sectors appearing in the generalized quivers associated to these punctures. These properties are derived in subsection 8.3, and subsequently used in subsection 8.4 to derive the defect groups associated to untwisted D-type punctures. Unlike untwisted A-type punctures, we find that untwisted D-type punctures can carry trapped 1-form symmetries.

Non-trivial checks of this proposal are provided using the IIB ALE fibrations in subsection 8.5. In subsection 8.6.1, we use the IHS realization to compute the trapped defect groups of IHS punctures, and find a perfect match with the trapped defect groups computed using our proposal. The 4d $\mathcal{N} = 2$ theories constructed by the IHS include the well-studied (A,D) type Argyres-Douglas theories, which are known to have non-trivial 1-form symmetry in general. In subsection 8.6.2, we study other IHS that construct trinion theories involving three untwisted D-type punctures. In all such cases, we find that the defect group of the trinion theory as computed using IHS maps the defect group as computed using our proposal.

## 8.1 Punctures

Now we consider punctures for 6d $D_n$ $\mathcal{N} = (2,0)$ theory. These are described in terms of a Hitchin field in vector representation of $D_n$.

The $i$-th block of the Hitchin field takes the form

$$B_i = \frac{1}{t^{1+r_i}} A_i + \cdots . \tag{254}$$

Again we order the blocks such that $r_i \geq r_{i+1}$ and let $s$ be biggest value of $i$ for which $r_i > 0$. For each $i \leq s$, we write

$$r_i = \frac{p_i}{q_i}, \tag{255}$$

where $1 \leq q_i \leq 2n$ and $\gcd(p_i, q_i) = 1$.

Let us again define

$$n_i := 2n - \sum_{j=1}^{i} q_j . \tag{256}$$

We impose extra conditions on the punctures that can be described in terms of $n_i$ as follows. If $n_i$ is even (including $n_0 := 2n$) and $q_{i+1}$ is odd, then we require

$$q_{i+2} = q_{i+1} . \tag{257}$$

This in particular ensures that $n_{i-1}$ and $n_{i+1}$ are even if $n_i$ is odd. Furthermore, we require that $n_s$ is even. If $s$ is odd then $n_s \geq 2$, and if $s$ is even then $n_s \geq 0$.

Similar to the untwisted A-type punctures, an untwisted D-type puncture is conformal only if $r_i = r_j$ for all $i, j \in \{1, 2, \cdots, s\}$. Out of these, the IHS punctures discussed in [81, 88] are those for which $n_s = 0, 2$.

The conformal untwisted-$D_N$ punctures $\mathcal{P}_k^b$ of IHS type have $b = 2N - 2$ or $b = N$. The theories $\mathfrak{T}_{\mathcal{P}_k^b}$ associated to $b = 2N - 2$ are the AD$[A, D]$ theories. Again, there is a prediction from the hyper-surface realization for the defect group of the theories $\mathfrak{T}_{\mathcal{P}_k^b}$, which we discuss in section 8.6.

## 8.2 Class $\mathcal{GQ}$ of Generalized Quivers

Consider a puncture $\mathcal{P}$ of the above type. Then we conjecture that $\mathcal{P} \in \mathcal{F}_\tau$ where $\tau$ is a generalized quiver that can be constructed using the data of $n_i$ as follows. The quiver takes the form

$$\mathfrak{so}(2n) \;\rule[0.5ex]{1.2em}{0.4pt}\; N_1 \;\rule[0.5ex]{1.2em}{0.4pt}\; N_2 \;\rule[0.5ex]{1.2em}{0.4pt}\; \cdots \;\rule[0.5ex]{1.2em}{0.4pt}\; N_{s-1} \;\rule[0.5ex]{1.2em}{0.4pt}\; [\mathfrak{g}_s] \; . \tag{258}$$

The node $N_i$ is either a gauge algebra or a matter SCFT, depending on whether $n_i$ is even or odd respectively. If $n_i$ is even, we write

$$N_i = \mathfrak{g}_i . \tag{259}$$

If $i$ is even, we have

$$\mathfrak{g}_i = \mathfrak{so}(n_i) \tag{260}$$

and if $i$ is odd, we have

$$\mathfrak{g}_i = \mathfrak{usp}(n_i - 2) . \tag{261}$$

The same applies to the flavor algebra $\mathfrak{g}_s$. If $s$ is even, we have

$$\mathfrak{g}_i = \mathfrak{so}(n_s) \tag{262}$$

and if $s$ is odd, we have

$$\mathfrak{g}_i = \mathfrak{usp}(n_s - 2). \tag{263}$$

If $n_i$ is odd and $i$ is odd, then we write

$$N_i = C(n_i), \tag{264}$$

where $C(n_i)$ is a matter SCFT having flavor algebra

$$\mathfrak{f}_{C(n_i)} = \mathfrak{so}(2n_i) \oplus \mathfrak{u}(1). \tag{265}$$

If $n_i$ is odd and $i$ is even, then we write

$$N_i = D(n_i), \tag{266}$$

where $D(n_i)$ is a matter SCFT having flavor algebra

$$\mathfrak{f}_{D(n_i)} = \mathfrak{usp}(2n_i - 4) \oplus \mathfrak{u}(1). \tag{267}$$

Let us now describe the edges in the quiver $\tau$. Consider the edge between $N_i$ and $N_{i+1}$ where both $n_i$ and $n_{i+1}$ are even. If $i$ is even, we have

$$\mathfrak{so}(n_i) \quad\text{———}\quad \mathfrak{usp}(n_{i+1} - 2) \quad. \tag{268}$$

If $i$ is odd, we have

$$\mathfrak{usp}(n_i - 2) \quad\text{———}\quad \mathfrak{so}(n_{i+1}) \quad. \tag{269}$$

In both cases, the edge denotes a half-hyper in bifundamental representation of the two gauge algebras. Now, consider the situation where $i$ is odd and $n_i$ is odd. If $i + 1 < s$, then locally the quiver looks like

$$\mathfrak{so}(n_{i-1}) \text{——} C(n_i) \text{——} \mathfrak{so}(n_{i+1}) \quad = \quad \mathfrak{so}(n_i + q_i) \text{——} C(n_i) \text{——} \mathfrak{so}(n_i - q_i) \quad, \tag{270}$$

where the edges denote that $\mathfrak{so}(n_{i-1}) \oplus \mathfrak{so}(n_{i+1}) = \mathfrak{so}(n_i + q_i) \oplus \mathfrak{so}(n_i - q_i) \subset \mathfrak{so}(2n_i) \subset \mathfrak{f}_{C(n_i)}$ subalgebra of the flavor algebra $\mathfrak{f}_{C(n_i)}$ of the matter SCFT $C(n_i)$ has been gauged. If $i + 1 = s$, then locally the quiver looks like

$$\mathfrak{so}(n_{s-2}) \text{——} C(n_{s-1}) \text{——} [\mathfrak{so}(n_s)] \quad = \quad \mathfrak{so}(n_s + 2q_s) \text{——} C(n_s + q_s) \text{——} [\mathfrak{so}(n_s)] \quad, \tag{271}$$

where the edges denote that $\mathfrak{so}(n_{s-2}) = \mathfrak{so}(n_s + 2q_s) \subset \mathfrak{so}(2n_s + 2q_s) \subset \mathfrak{f}_{C(n_{s-1})}$ subalgebra of the flavor algebra $\mathfrak{f}_{C(n_{s-1})}$ of the matter SCFT $C(n_{s-1})$ has been gauged. Now, consider the situation where $i$ is even and $n_i$ is odd. If $i + 1 < s$, then locally the quiver looks like

$$\mathfrak{usp}(n_{i-1} - 2) \text{——} D(n_i) \text{——} \mathfrak{usp}(n_{i+1} - 2)$$

$$= \quad \mathfrak{usp}(n_i + q_i - 2) \text{——} D(n_i) \text{——} \mathfrak{usp}(n_i - q_i - 2) \quad, \tag{272}$$

where the edges denote that
$\mathfrak{usp}(n_{i-1} - 2) \oplus \mathfrak{usp}(n_{i+1} - 2) = \mathfrak{usp}(n_i + q_i - 2) \oplus \mathfrak{usp}(n_i - q_i - 2) \subset \mathfrak{usp}(2n_i - 4) \subset \mathfrak{f}_{D(n_i)}$
subalgebra of the flavor algebra $\mathfrak{f}_{D(n_i)}$ of the matter SCFT $D(n_i)$ has been gauged. If $i + 1 = s$, then locally the quiver looks like

$$\mathfrak{usp}(n_{s-2} - 2) \text{——} D(n_{s-1}) \text{——} [\mathfrak{usp}(n_s - 2)]$$

$$= \quad \mathfrak{usp}(n_s + 2q_s - 2) \text{——} D(n_s + q_s) \text{——} [\mathfrak{usp}(n_s - 2)] \quad, \tag{273}$$

where the edges denote that $\mathfrak{usp}(n_{s-2} - 2) = \mathfrak{usp}(n_s + 2q_s - 2) \subset \mathfrak{usp}(2n_s + 2q_s - 4) \subset \mathfrak{f}_{D(n_{s-1})}$ subalgebra of the flavor algebra $\mathfrak{f}_{D(n_{s-1})}$ of the matter SCFT $D(n_{s-1})$ has been gauged.

### 8.2.1 Class $\mathcal{G}\mathcal{Q}'$ and Type IIA Punctures

The Class $\mathcal{G}\mathcal{Q}'$ of punctures is given by punctures having $p_i = 1$ for all $1 \le i \le s$. The Type IIA punctures form a subclass of $\mathcal{G}\mathcal{Q}'$ for which all $q_i$ and hence all $n_i$ are even. In such a situation, every node $N_i$ in (5.2) is a gauge algebra, which is $\mathfrak{so}(n_i)$ for $i$ even and $\mathfrak{usp}(n_i - 2)$ for $i$ odd. The Type IIA brane construction corresponding to a Type IIA puncture $\mathcal{P}'$ is

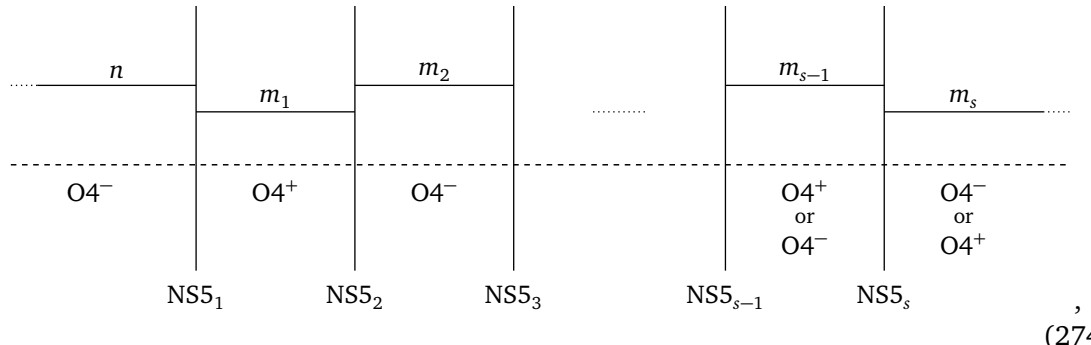

$$\tag{274}$$

where $m_i$ number of physical D4 branes have been placed in the $i^{\text{th}}$ interval, where $m_i = \frac{n_i}{2}$ for $i$ even and $m_i = \frac{n_i}{2} - 1$ for $i$ odd. Moreover, the $i^{\text{th}}$ interval carries an O4$^-$ plane if $i$ is even and an O4$^+$ plane if $i$ is odd.

## 8.3 The $C(m)$ and $D(m)$ SCFTs

### 8.3.1 The $C(m)$ SCFT

The theory $C(m)$ is a 4d $\mathcal{N} = 2$ SCFT which is better known as the $D_2^{2m-4}\big(SO(2m-2)\big)$ theory lying in the class $D_p^b(G)$ of 4d $\mathcal{N} = 2$ SCFTs. This theory can be recognized as the 4d $\mathcal{N} = 2$ theory $\mathfrak{T}_\mathcal{P}^*$ associated to an untwisted puncture $\mathcal{P}$ of 6d $\mathcal{N} = (2,0)$ theory of type $\mathfrak{so}(2m-2)$. The puncture $\mathcal{P}$ is characterized by

$$s = 2, \quad q_1 = q_2 = m - 2, \quad p_1 = p_2 = 1. \tag{275}$$

For even $m$, it can also be recognized as a Lagrangian 4d $\mathcal{N} = 2$ theory carrying $\mathfrak{usp}(m-2)$ gauge algebra and $m$ full hypermultiplets in the fundamental representation. On the other hand, for odd $m$, there is no such Lagrangian description available. This is the case that arises as matter SCFT in the generalized quivers discussed above.

In order to determine the defect group associated to the generalized quivers discussed above, we need to determine the defect group of $C(2m+1)$, and additionally also determine the flavor center charges of the genuine local operators in the $C(2m+1)$ SCFT. For this purpose, we utilize another Class S construction of $C(2m + 1)$ discussed in [62] (where this model was called $R_{2,2m-1}$) that uses only regular punctures. This involves the compactification of 6d $\mathcal{N} = (2,0)$ theory of type $\mathfrak{su}(2m - 1)$ on a sphere with three regular punctures. One of the punctures is maximal regular (whose associated partition is $[1^{2m-1}]$). The other two punctures are of the same type, described by partition $[m-1, m-1, 1]$. Since the Class S compactification involves only regular punctures, the defect group of lines of $C(2m + 1)$ must be trivial [31]. Thus there is no contribution to defect group from a node of the form (264) in the generalized quivers discussed above.

The manifest flavor symmetry (visible from the regular punctures) is only

$$\mathfrak{f}_{\text{man}} = \mathfrak{su}(2m - 1) \oplus \mathfrak{u}(2) \oplus \mathfrak{u}(2), \tag{276}$$

with $\mathfrak{su}(2m - 1)$ subfactor arising from the maximal puncture, and the two $\mathfrak{u}(2)$ subfactors arising from the two other punctures. For our purposes, it would be sufficient to keep track of

only the non-abelian part of the manifest flavor symmetry which is

$$\mathfrak{f}_{\text{man}}^{\text{na}} = \mathfrak{su}(2m-1) \oplus \mathfrak{so}(4), \tag{277}$$

which embeds into $\mathfrak{so}(4m-2) \oplus \mathfrak{so}(4)$ and then further into $\mathfrak{so}(4m+2)$ part of the full flavor symmetry algebra

$$\mathfrak{f}_{C(2m+1)} = \mathfrak{so}(4m+2) \oplus \mathfrak{u}(1). \tag{278}$$

We can easily study representations of local operators under $\mathfrak{f}_{\text{man}}$ or $\mathfrak{f}_{\text{man}}^{\text{na}}$ using the method of [104]. We find that there exist local operators charged in spinor and cospinor irreps of $\mathfrak{so}(4)$, while being uncharged under $\mathfrak{su}(2m-1)$. This implies that the theory $C(2m+1)$ must contain a local operator charged in spinor irrep $S$ of $\mathfrak{so}(4m+2)$.

This local operator is charged under

$$SC \oplus CS \tag{279}$$

representation of $\mathfrak{so}(2p) \oplus \mathfrak{so}(4m+2-2p)$ subalgebra of $\mathfrak{so}(4m+2)$. Thus, in the generalized quivers discussed above, we can replace a sub-quiver of the form

$$\mathfrak{so}(n_{i-1}) \mathrel{\rule[0.5ex]{2em}{0.4pt}} C(n_i) \mathrel{\rule[0.5ex]{2em}{0.4pt}} \mathfrak{so}(n_{i+1}) \quad, \tag{280}$$

with

$$\mathfrak{so}(n_{i-1}) \overset{S}{\underset{C}{\rule[0.8ex]{6em}{0.4pt}\rule[-0.3ex]{0pt}{0pt}}\overset{C}{\underset{S}{}}} \mathfrak{so}(n_{i+1}) \quad, \tag{281}$$

as far as computation of defect group is concerned. The peculiar double edge denotes that we have matter content transforming as $SC \oplus CS$ under the gauge algebra $\mathfrak{so}(n_{i-1}) \oplus \mathfrak{so}(n_{i+1})$. Similarly, we can replace a sub-quiver of the form

$$\mathfrak{so}(n_{s-2}) \mathrel{\rule[0.5ex]{2em}{0.4pt}} C(n_{s-1}) \mathrel{\rule[0.5ex]{2em}{0.4pt}} [\mathfrak{so}(n_s)] \quad, \tag{282}$$

with

$$\mathfrak{so}(n_{s-2}) \overset{S}{\underset{C}{\rule[0.8ex]{6em}{0.4pt}}\overset{C}{\underset{S}{}}} [\mathfrak{so}(n_s)] \quad, \tag{283}$$

as far as computation of defect group is concerned. The double edge denotes that we have matter content transforming as $SC \oplus CS$ under the algebra $\mathfrak{so}(n_{s-2}) \oplus \mathfrak{so}(n_s)$, where $\mathfrak{so}(n_s-2)$ is a gauge algebra while $\mathfrak{so}(n_s)$ is a flavor algebra.

### 8.3.2 The $D(m)$ SCFT

The theory $D(m)$ is a 4d $\mathcal{N} = 2$ SCFT which is better known as the $D_2^{2m-4}\big(SO(2m-2)/\mathbb{Z}_2\big)$ theory lying in the class $D_p^b(G/\mathbb{Z}_i)$ of 4d $\mathcal{N} = 2$ SCFTs. This theory can be recognized as the 4d $\mathcal{N} = 2$ theory $\mathfrak{T}_{\mathcal{P}}^*$ associated to a twisted puncture $\mathcal{P}$ of 6d $\mathcal{N} = (2,0)$ theory of type $\mathfrak{so}(2m-2)$. The puncture $\mathcal{P}$ is characterized by

$$s = 2, \quad q_1 = q_2 = m-2, \quad p_1 = p_2 = 1, \tag{284}$$

where the data characterized twisted punctures of D type is discussed in section 9.1. For even $m$, it can also be recognized as a Lagrangian 4d $\mathcal{N} = 2$ theory carrying $\mathfrak{so}(m)$ gauge algebra and $m-2$ hypermultiplets in the fundamental representation. On the other hand, for odd $m$, there is no such Lagrangian description available. This is the case that arises as matter SCFT in the generalized quivers discussed above.

In order to determine the defect group associated to the generalized quivers discussed above, we need to determine the defect group of $D(2m+1)$, and additionally also determine the flavor center charges of the genuine local operators in the $D(2m+1)$ SCFT. For this purpose, we utilize another construction of the $D(2m+1)$. We propose that $D(2m+1)$ can be obtained as a circle reduction of the $5d$ SCFT that admits a relevant deformation to $5d$ $\mathcal{N}=1$ gauge theory carrying $\mathfrak{su}(2m)$ gauge algebra at CS level 0 and $4m-2$ fundamental hypers. The circle reduction involves a twist by charge conjugation symmetries associated to $\mathfrak{su}(2m)$ gauge algebra and $\mathfrak{su}(4m-2)$ flavor algebra. Our proposal is an extension of the proposal made in [107] for the case of $m=2$.

Since the $5d$ theory has no line and surface defects modulo screening, we deduce that $D(2m+1)$ has no defect group of lines. Thus there is no contribution to defect group from a node of the form (266) in the generalized quivers discussed above.

To compute the flavor center charges of the genuine local operators in the $D(2m+1)$ SCFT, we look at the magnetic quiver of the $5d$ theory, which is [108]

$$\mathfrak{u}(1) \longrightarrow \mathfrak{u}(1)$$

$$\mathfrak{u}(1) \longrightarrow \mathfrak{u}(2) \longrightarrow \cdots \longrightarrow \mathfrak{u}(2m-2) \longrightarrow \mathfrak{u}(2m-1) \longrightarrow \mathfrak{u}(2m-2) \longrightarrow \cdots \longrightarrow \mathfrak{u}(2) \longrightarrow \mathfrak{u}(1),$$

(285)

where the balanced nodes and the edges connecting them are shown in blue. The balanced sub-quiver forms an $\mathfrak{su}(4m-2)$ Dynkin diagram corresponding to the $\mathfrak{su}(4m-2)$ flavor symmetry of the $5d$ theory. Following the proposal of [109], because the circle reduction involves the outer automorphism of $\mathfrak{su}(4m-2)$, the magnetic quiver for the $4d$ theory can be obtained by folding the magnetic quiver of the $5d$ theory by the action of the outer-automorphism. Thus, the magnetic quiver for the $4d$ $D(2m+1)$ SCFT is

$$\mathfrak{u}(1) \longrightarrow \mathfrak{u}(1)$$

$$\mathfrak{u}(2m-1) \Longrightarrow \mathfrak{u}(2m-2) \longrightarrow \cdots \longrightarrow \mathfrak{u}(2) \longrightarrow \mathfrak{u}(1) \ .$$

(286)

Non simply-laced quivers like (286) were first introduced in [110], where a prescription for computing their Coulomb branch Hilbert seris was proposed. The Coulomb branch is essentially obtained from that of the underlying simply-laced quiver (285) by restricting to monopole configurations invariant under the outer-automorphism action (see also [111]). From the quiver (286) we manifestly see the $\mathfrak{usp}(4m-2)$ flavor symmetry subalgebra of the $D(2m+1)$ theory. The fact that we have an unbalanced node attached to the $(2m-1)^{\text{th}}$ node of the $\mathfrak{usp}(4m-2)$ algebra implies that we have a genuine (Higgs branch) local operator in the $D(2m+1)$ theory that transforms in the representation $\Lambda^{2m-1}$ obtained by antisymmetrizing the $(2m-1)$th power of the fundamental irrep of $\mathfrak{usp}(4m-2)$. The local operator carries a non-trivial charge under the $\mathbb{Z}_2$ center of the simply connected group $USp(4m-2)$ associated to the flavor algebra $\mathfrak{usp}(4m-2)$. This implies that we must have another genuine local operator in the theory carrying the fundamental irrep $\mathsf{F}$ of $\mathfrak{usp}(4m-2)$.

This local operator is charged under

$$\mathsf{F} \oplus \mathsf{F} \tag{287}$$

representation of $\mathfrak{usp}(2p) \oplus \mathfrak{usp}(4m-2-2p)$ subalgebra of $\mathfrak{usp}(4m-2)$. Thus, in the generalized quivers discussed above, we can replace a sub-quiver of the form

$$\mathfrak{usp}(n_{i-1}) \longrightarrow D(n_i) \longrightarrow \mathfrak{usp}(n_{i+1}) \ , \tag{288}$$

with

$$\mathfrak{usp}(n_{i-1}) \ \text{---} \ \mathsf{F} \qquad \mathsf{F} \ \text{---} \ \mathfrak{usp}(n_{i+1}) \ , \tag{289}$$

as far as computation of defect group is concerned, thus splitting the quiver into two pieces. An edge joining $\mathsf{F}$ to $\mathfrak{usp}(n)$ denotes matter transforming as $\mathsf{F}$ of $\mathfrak{usp}(n)$. Similarly, we can replace a sub-quiver of the form

$$\mathfrak{usp}(n_{s-2}) \ \text{---} \ D(n_{s-1}) \ \text{---} \ [\mathfrak{usp}(n_s)] \ , \tag{290}$$

with

$$\mathfrak{usp}(n_{s-2}) \ \text{---} \ \mathsf{F} \ , \tag{291}$$

as far as computation of defect group is concerned, where we have thrown away a piece that does not involve gauge algebras.

## 8.4 1-Form Symmetries

As proposed above, we can compute the defect group associated to an untwisted puncture $\mathcal{P}$ by computing the defect group associated to a generalized quiver of the form (258). We compute the defect group of the quiver in terms of its 1-form symmetry as explained in section 5.2.

Before describing the result, we introduce some notation. Let us call an $\mathfrak{so}$ gauge node to be of Type I if exactly one of its neighbors is a $C(\text{odd})$ SCFT. Let $\mu_1$ be the number of $\mathfrak{so}$ gauge nodes of Type I and let $\nu_1 = \lfloor \frac{\mu_1}{2} \rfloor$. Similarly, let us call an $\mathfrak{so}$ gauge node to be of Type II if none of its neighbors is a $C(\text{odd})$ SCFT. Let $\mu_2$ be the number of $\mathfrak{so}$ gauge nodes of Type II. Let us define $\nu_2 = 1$ if $\nu_1 + \mu_2 > 0$ and $\nu_2 = 0$ if $\nu_1 = \mu_2 = 0$. Moreover, let $\mu_3$ be the number of nodes carrying $D(\text{odd})$ SCFT. Finally, let us also define $\mu_s = n_s$ if $s$ is even, and $\mu_s = n_s - 2$ if $s$ is odd.

Then we can divide the result into the following cases:

- Consider the case when $\mu_1 + \mu_3 + \mu_s > 0$. Then we have

$$\begin{aligned} \mathcal{H}_{\mathcal{P}}^T &= \mathbb{Z}_2^{\nu_1 + \mu_2 - \nu_2} \,, \\ Z_{\mathcal{P}} &= \mathbb{Z}_2^{\nu_2} \,, \\ \mathcal{H}_{\mathcal{P}} &= \mathcal{H}_{\mathcal{P}}^T \times Z_{\mathcal{P}} \,, \end{aligned} \tag{292}$$

  with $i_{\mathcal{P}}$ mapping $\mathcal{H}_{\mathcal{P}}^T$ identically to the $\mathcal{H}_{\mathcal{P}}^T$ subfactor of $\mathcal{H}_{\mathcal{P}}$.

- Consider the case when $\mu_1 = \mu_3 = \mu_s = 0$ and $n = 2m + 1$. Then we have

$$\begin{aligned} \mathcal{H}_{\mathcal{P}}^T &= \mathbb{Z}_2^{\mu_2 - 1} \,, \\ Z_{\mathcal{P}} &= \mathbb{Z}_4 \,, \\ \mathcal{H}_{\mathcal{P}} &= \mathcal{H}_{\mathcal{P}}^T \times Z_{\mathcal{P}} \,, \end{aligned} \tag{293}$$

  with $i_{\mathcal{P}}$ mapping $\mathcal{H}_{\mathcal{P}}^T$ identically to the $\mathcal{H}_{\mathcal{P}}^T$ subfactor of $\mathcal{H}_{\mathcal{P}}$.

- Consider the case when $\mu_1 = \mu_3 = \mu_s = 0$, $n = 2m$ and $n_{2i} = 4m_{2i}$ for all integer $0 < i < s/2$. Then we have

$$\begin{aligned} \mathcal{H}_{\mathcal{P}}^T &= \mathbb{Z}_2^{\mu_2 - 1} \,, \\ Z_{\mathcal{P}} &= \mathbb{Z}_2^2 \,, \\ \mathcal{H}_{\mathcal{P}} &= \mathcal{H}_{\mathcal{P}}^T \times Z_{\mathcal{P}} \,, \end{aligned} \tag{294}$$

  with $i_{\mathcal{P}}$ mapping $\mathcal{H}_{\mathcal{P}}^T$ identically to the $\mathcal{H}_{\mathcal{P}}^T$ subfactor of $\mathcal{H}_{\mathcal{P}}$.

- Consider the case when $\mu_1 = \mu_3 = \mu_s = 0$, $n = 2m$ and $n_{2i} = 4m_{2i} + 2$ for at least one integer $0 < i < s/2$. Then we have

$$
\begin{aligned}
\mathcal{H}_{\mathcal{P}}^T &= \mathbb{Z}_2^{\mu_2 - 2} \times \mathbb{Z}_2 \,, \\
Z_{\mathcal{P}} &= \mathbb{Z}_2 \times \mathbb{Z}_2 \,, \\
\mathcal{H}_{\mathcal{P}} &= \mathbb{Z}_2^{\mu_2 - 2} \times \mathbb{Z}_4 \times \mathbb{Z}_2 \,.
\end{aligned}
\tag{295}
$$

Unlike the above cases, in this case, the short exact sequence

$$
0 \to \mathcal{H}_{\mathcal{P}}^T \to \mathcal{H}_{\mathcal{P}} \to Z_{\mathcal{P}} \to 0
\tag{296}
$$

does not split. The map $i_{\mathcal{P}}$ maps $\mathbb{Z}_2^{\mu_2 - 2}$ subfactor of $\mathcal{H}_{\mathcal{P}}^T$ identically to the $\mathbb{Z}_2^{\mu_2 - 2}$ subfactor of $\mathcal{H}_{\mathcal{P}}$, and the $\mathbb{Z}_2$ subfactor of $\mathcal{H}_{\mathcal{P}}^T$ to the $\mathbb{Z}_2$ subgroup of the $\mathbb{Z}_4$ subfactor of $\mathcal{H}_{\mathcal{P}}$. Correspondingly, the map $\pi_{\mathcal{P}}$ maps the $\mathbb{Z}_2^{\mu_2 - 2}$ subfactor of $\mathcal{H}_{\mathcal{P}}$ to zero in $Z_{\mathcal{P}}$, the generator of $\mathbb{Z}_4$ subfactor of $\mathcal{H}_{\mathcal{P}}$ to the generator of one of the $\mathbb{Z}_2$ subfactors of $Z_{\mathcal{P}}$, and the final $\mathbb{Z}_2$ subfactor of $\mathcal{H}_{\mathcal{P}}$ identically to the other $\mathbb{Z}_2$ subfactor of $Z_{\mathcal{P}}$.

## 8.5 Spectral Cover Monodromies

The behavior of the Higgs field near the puncture takes the form

$$
F(v, t) = v^{2n} + \sum_{i \in \mathcal{I}} \frac{v^{n_i}}{t^{P_i}} + \frac{1}{t^k} = 0 \,,
\tag{297}
$$

where $\mathcal{I}$ is the set of $1 \le i \le s$ for which $n_i$ is even, $k$ is an integer and

$$
P_j := \sum_{i=1}^{j} p_i \,.
\tag{298}
$$

This is similar to the behavior for untwisted A-type punctures, along with an extra constant term $v^0 t^0$. If the term $1/t^k$ is not added, then the $v$-values for two sheets coincide in the vicinity of the puncture. In such cases, the constant term provides a distance between the two sheets, allowing us to read the full monodromy. Different values of $k$ lead to distinct monodromies and which value of $k$ captures the generic monodromy depends on the (extended) Coulomb branch of the set-up.

Fix a point $t_0 \ne 0$ in the vicinity of the puncture. There are only $n$ independent values of $v$ solving $F(v, t_0) = 0$, as the other $n$ values of $v$ are fixed to be negatives of the former $n$ values, due to the symmetry $v \to -v$ of $F(v, t) = 0$. We can identify a set of $n$ independent values of $v$ solving $F(v, t_0) = 0$ as $n$ of the weights $e_a$ for $1 \le a \le n$. The other $n$ values of $v$ solving $F(v, t_0) = 0$ are identified with the weights $-e_a$.

As one goes around the puncture $t_0 \to e^{2\pi i} t_0$, the $2n$ weights are permuted into each other. This induces an action on the roots, which can be identified as

$$
\alpha_a = e_a - e_{a+1} \,,
\tag{299}
$$

for $1 \le a \le n-1$ and

$$
\alpha_n = e_{n-1} + e_n \,.
\tag{300}
$$

Let the action on the roots associated to a puncture $\mathcal{P}$ be denoted by an $n \times n$ matrix $\mathcal{M}_{\mathcal{P}}$. Then, as before, we have the identification

$$
\mathcal{L}_{\mathcal{P}}^0 = \mathrm{Tor}\, \frac{\mathbb{Z}^n}{\mathcal{T}_{\mathcal{P}} \cdot \mathbb{Z}^n} \,,
\tag{301}
$$

where

$$
\mathcal{T}_{\mathcal{P}} := \mathcal{M}_{\mathcal{P}} - 1 \,.
\tag{302}
$$

**Example:** $\mathcal{P}_0$ **puncture.** Let us consider the example of $\mathcal{P}_0$ puncture, which has $s = 2, p_1 = 1, q_1 = 2n - 2, p_2 = 0, q_2 = 1$. The curve is

$$v^{2n} + 1 + \frac{v^2}{t} = 0, \tag{303}$$

where we have moved onto the Coulomb branch to resolve an order two degeneracy with $v = 0$. There is a $\mathbb{Z}_{2n-2} \times \mathbb{Z}_2$ monodromy around the $\mathcal{P}_0$ puncture. That is, $2n - 2$ sheets are cyclically permuted amongst themselves while a pair of sheets are interchanged along the same path. We pick a set of $n$ pairwise linearly independent sheets and label these by weights $e_a$ of the $2n$-dimensional vector representation such that these are permuted as

$$e_a \to e_{a+1}, \qquad e_{n-1} \to -e_1, \qquad e_n \to -e_n, \tag{304}$$

where $a = 1, \ldots, n-2$. With respect to this labelling the monodromy action on sheets leads to the following action on the roots

$$\alpha_b \to \alpha_{b+1}, \qquad \alpha_{n-2} \to \sum_{a=1}^{n} \alpha_a, \qquad \alpha_{n-1} \to -\sum_{a=1}^{n-1} \alpha_a, \qquad \alpha_n \to -\alpha_n - \sum_{a=1}^{n-2} \alpha_a, \tag{305}$$

where $b = 1, \ldots, n-3$. The mondromy matrix is thus

$$M_{\mathcal{P}_0} = \begin{pmatrix} 0 & 0 & \cdots & 0 & 0 & 1 & -1 & -1 \\ 1 & 0 & \cdots & 0 & 0 & 1 & -1 & -1 \\ 0 & 1 & \ddots & 0 & 0 & 1 & -1 & -1 \\ \vdots & \ddots & \ddots & \ddots & \vdots & \vdots & \vdots & \vdots \\ 0 & 0 & \ddots & 1 & 0 & 1 & -1 & -1 \\ 0 & 0 & \ldots & 0 & 1 & 1 & -1 & -1 \\ 0 & 0 & \ldots & 0 & 0 & 1 & -1 & 0 \\ 0 & 0 & \ldots & 0 & 0 & 1 & 0 & -1 \end{pmatrix}. \tag{306}$$

From here we define

$$T_{\mathcal{P}_0} = M_{\mathcal{P}_0} - 1, \tag{307}$$

and the reader can check that

$$\text{SNF}\left(T_{\mathcal{P}_0}\right) = \begin{cases} \text{diag}(4, 1, 1, \ldots, 1) & \mathfrak{g} = D_{2n+1}, \\ \text{diag}(2, 2, 1, \ldots, 1) & \mathfrak{g} = D_{2n}, \end{cases} \tag{308}$$

implying that

$$\mathcal{L}_{\mathcal{P}_0}^0 = \begin{cases} \mathbb{Z}_4 & \mathfrak{g} = D_{2n+1}, \\ \mathbb{Z}_2 \times \mathbb{Z}_2 & \mathfrak{g} = D_{2n}. \end{cases} \tag{309}$$

**Example: Puncture with trapped defects and non-trivial extension.** Let us consider the puncture $\mathcal{P}'$ engineering the $\mathcal{GQ}'$ quiver

$$[\mathfrak{so}(12)] \,\text{------}\, \mathfrak{usp}(8) \,\text{------}\, \mathfrak{so}(6) \,, \tag{310}$$

which has $s = 3, p_i = 1, q_1 = 2, q_2 = 4, q_3 = 4$ and the curve

$$v^{12} + \frac{v^{10}}{t} + \frac{v^6 + 1}{t^2} + \frac{v^2}{t^3} = 0. \tag{311}$$

Here the term $1/t^2$ resolves a degeneracy at $v = 0$, it is the leading order contribution for generic Coulomb branch parameters. There is a $\mathbb{Z}_4^2 \times \mathbb{Z}_2^2$ monodromy around the $\mathcal{P}'$ puncture. We pick a set of 6 pairwise linearly independent sheets and label these by weights $e_a$ of the $2n$-dimensional vector representation such that these are permuted as

$$e_1 \to e_2 \to -e_1 \to -e_2\,, \qquad e_3 \to e_4 \to -e_3 \to -e_4\,, \qquad e_5 \to -e_5\,, \qquad e_6 \to -e_6\,. \tag{312}$$

With respect to this labelling the monodromy action on sheets leads to the action on the roots summarized by the monodromy matrix

$$M_{\mathcal{P}'} = \begin{pmatrix} 1 & -1 & 0 & 0 & 0 & 0 \\ 2 & -1 & 0 & 0 & 0 & 0 \\ 2 & -1 & 1 & -1 & 0 & 0 \\ 2 & -2 & 2 & -1 & 0 & 0 \\ 1 & -1 & 1 & 0 & -1 & 0 \\ 1 & -1 & 1 & 0 & 0 & -1 \end{pmatrix}. \tag{313}$$

From here we define

$$T_{\mathcal{P}'} = M_{\mathcal{P}'} - 1\,, \tag{314}$$

and the reader can check that

$$\mathrm{SNF}\,(T_{\mathcal{P}'}) = \mathrm{diag}\,(4, 2, 2, 1, 1, 1)\,, \tag{315}$$

implying that

$$\mathcal{L}_{\mathcal{P}'}^0 = \mathbb{Z}_2^2 \times \mathbb{Z}_4\,. \tag{316}$$

## 8.6 Trapped Defect Group from Type IIB on CY3

Again, the trapped part of the defect group can be computed from the Type IIB realization on IHS singularities.

### 8.6.1 Trapped Parts of IHS Punctures

For a IHS-puncture $\mathcal{P}$, we can compute the trapped defect group using their isolated hypersurface singularity $X_{\mathcal{P}}$ realization.

The theories of with untwisted D type class S with irregular punctures can be realized using the following hypersurface singularities: the AD$[A, D]$, which have the hypersurface realizations in type IIB as

$$X_{\mathrm{AD}[A_l, D_n]}: \qquad x_1^2 + x_2^{l+1} + x_3^{n-1} + x_3 x_4^2 = 0]\,. \tag{317}$$

This singularity has associated to it $b = 2n - 2$ and $k = l + 2n - 1$. And the second is of type VII $(\{7, 1\}, \{2, n-1, 2, l\})$ in the nomenclature of [55, 79, 85]

$$X_{\{7,1\}(2,n-1,2,l)}: \qquad x_1^2 + x_2^{n-1} + x_2 x_3^2 + x_3 x_4^l = 0\,. \tag{318}$$

This singularity has associated to it $b = n$ and $k = l + n$.

Table 4: Defect groups for the AD[$A_l, D_n$] theories.

| Tor $H_2(\partial X_{\text{AD}[A_l,D_n]})$ | $D_4$ | $D_5$ | $D_6$ | $D_7$ | $D_8$ | $D_9$ | $D_{10}$ | $D_{11}$ | $D_{12}$ | $D_{13}$ | $D_{14}$ | $D_{15}$ |
|---|---|---|---|---|---|---|---|---|---|---|---|---|
| $A_1$ | 0 | 0 | 0 | 0 | 0 | 0 | 0 | 0 | 0 | 0 | 0 | 0 |
| $A_2$ | $\mathbb{Z}_2^2$ | 0 | 0 | $\mathbb{Z}_2^2$ | 0 | 0 | $\mathbb{Z}_2^2$ | 0 | 0 | $\mathbb{Z}_2^2$ | 0 | 0 |
| $A_3$ | 0 | $\mathbb{Z}_2^2$ | 0 | 0 | 0 | $\mathbb{Z}_2^2$ | 0 | 0 | 0 | $\mathbb{Z}_2^2$ | 0 | 0 |
| $A_4$ | 0 | 0 | $\mathbb{Z}_2^4$ | 0 | 0 | 0 | 0 | $\mathbb{Z}_2^4$ | 0 | 0 | 0 | 0 |
| $A_5$ | 0 | 0 | 0 | $\mathbb{Z}_2^4$ | 0 | 0 | 0 | 0 | 0 | $\mathbb{Z}_2^4$ | 0 | 0 |
| $A_6$ | 0 | 0 | 0 | 0 | $\mathbb{Z}_2^6$ | 0 | 0 | 0 | 0 | 0 | 0 | $\mathbb{Z}_2^6$ |
| $A_7$ | 0 | 0 | 0 | 0 | 0 | $\mathbb{Z}_2^6$ | 0 | 0 | 0 | 0 | 0 | 0 |
| $A_8$ | $\mathbb{Z}_2^2$ | 0 | 0 | $\mathbb{Z}_2^2$ | 0 | 0 | $\mathbb{Z}_2^8$ | 0 | 0 | $\mathbb{Z}_2^2$ | 0 | 0 |

The defect group of the AD[A, D] theories was computed in [17, 30], shown for low values in table 4 and has as closed form

$$
\mathcal{L}_{X_{\text{AD}[A_l,D_n]}} = \begin{cases} \mathbb{Z}_2^{\gcd(l+1,n-1)-1} & \text{if } 2 \nmid (l+1), \\ \mathbb{Z}_2^{\gcd(l+1,n-1)-2} & \text{if } 2 | (n-1) \text{ and } \gcd(l+1,2(n-1))|(n-1), \\ 0 & \text{else}. \end{cases} \tag{319}
$$

For the hypersurface $X_{7,1(2,n-1,2,l)}$ the result is shown for low values of $l, n$ in table 5.

Below, we match these results with the trapped defect group of the IHS punctures computed using our proposal. An IHS puncture is characterized by two numbers $b, k$, where the possible values of $b$ are $n, 2n - 2$. These values of $b$ correspond respectively to $n_s = 0, 2$. The number $k$ is any positive integer. Since the IHS punctures are conformal, they have the property that all $p_i$ are equal and all $q_i$ are equal. Let $r = r_i, p = p_i, q = q_i$. Then, we can write $r$ as

$$
r = \frac{k}{b}, \tag{320}
$$

from which we find

$$
p = \frac{k}{\gcd(k,b)}, \\
q = \frac{b}{\gcd(k,b)}. \tag{321}
$$

We also compute

$$
s = \gcd(k,b) \qquad \text{if } b = 2n - 2, \\
s = 2 \cdot \gcd(k,b) \qquad \text{if } b = n. \tag{322}
$$

We divide the further analysis into the following three cases:

- Assume $q$ is odd. Then $\mu_1 = 1$ and hence $\nu_1 = 0$. Also, $\mu_2 = 0$, so we find that

$$
\mathcal{L}_{\mathcal{P}}^T = 0. \tag{323}
$$

- Assume $q$ is even and $b = 2n - 2$. Then $\mu_1 = 0$ and $\mu_2 = 1 + \left\lfloor \frac{\gcd(k,b)-1}{2} \right\rfloor$. So, we find that

$$
\mathcal{L}_{\mathcal{P}}^T = \mathbb{Z}_2^{2\left\lfloor \frac{\gcd(k,b)-1}{2} \right\rfloor}. \tag{324}
$$

Table 5: The 1-form symmetries for the hypersurfaces of type $X_{\{7,1\}(2,n-1,2,l)} : 0 = x^2 + y^{n-1} + yu^2 + uv^l$, $l$ increases downwards, $n$ increases to the right.

| $\operatorname{Tor} H_2(\partial X) : l \backslash n$ | 4 | 5 | 6 | 7 | 8 | 9 | 10 | 11 | 12 |
|---|---|---|---|---|---|---|---|---|---|
| 4 | 0 | 0 | 0 | 0 | 0 | 0 | 0 | 0 | 0 |
| 5 | 0 | 0 | 0 | 0 | 0 | 0 | 0 | 0 | 0 |
| 6 | $\mathbb{Z}_2^2$ | 0 | 0 | 0 | 0 | 0 | 0 | 0 | 0 |
| 7 | 0 | 0 | 0 | 0 | 0 | 0 | 0 | 0 | 0 |
| 8 | 0 | 0 | 0 | 0 | $\mathbb{Z}_2^2$ | 0 | 0 | 0 | 0 |
| 9 | 0 | 0 | $\mathbb{Z}_2^4$ | 0 | 0 | 0 | 0 | 0 | 0 |
| 10 | $\mathbb{Z}_2^2$ | 0 | 0 | 0 | 0 | 0 | 0 | 0 | 0 |
| 11 | 0 | 0 | 0 | 0 | 0 | 0 | 0 | 0 | 0 |
| 12 | 0 | 0 | 0 | 0 | $\mathbb{Z}_2^6$ | 0 | 0 | 0 | 0 |

- Assume $q$ is even and $b = n$. Then $\mu_1 = 0$ and $\mu_2 = 1 + \lfloor \gcd(k,b) - \frac{1}{2} \rfloor$. So, we find that

$$\mathcal{L}_{\mathcal{P}}^T = \mathbb{Z}_2^{2\lfloor \gcd(k,b) - \frac{1}{2} \rfloor}. \tag{325}$$

One can easily check that this matches (319) with the identification $l = k - 2n + 1$ and $b = 2n - 2$. It also matches with the results in table 5.

### 8.6.2 Trinions

Using IHS, we can construct trinion theories of the form discussed in section 6.5.2, where each puncture $\mathcal{P}_i$ is an IHS puncture. Let $k_i, b_i$ describe the data of these punctures. To compute the defect group $\mathcal{L}_{\mathcal{P}_1,\mathcal{P}_2,\mathcal{P}_3}$ of the trinion theory, we need to first compute $Z_{\mathcal{P}_i}$:

- If $q_i$ is odd, then $\mu_1 > 0$ and $\nu_2 = 0$, so we have

$$Z_{\mathcal{P}_i} = 0. \tag{326}$$

- If $q_i$ is even, then $\mu_1 = \nu_1 = \mu_3 = 0$ and $\nu_2 = 1$. If $\mu_s > 0$, then

$$Z_{\mathcal{P}_i} = Z_2. \tag{327}$$

- If $q_i$ is even and $\mu_s = 0$, then

$$Z_{\mathcal{P}_i} = Z_G, \tag{328}$$

where $Z_G = \mathbb{Z}_2^2$ if $n$ is even, and $Z_G = \mathbb{Z}_4$ if $n$ is odd.

Thus, if any of the $q_i$ is odd, then

$$Z_{\mathcal{P}_1,\mathcal{P}_2,\mathcal{P}_3} = 0 \tag{329}$$

and hence using (221), we find that

$$\mathcal{L}_{\mathcal{P}_1,\mathcal{P}_2,\mathcal{P}_3} = \left( \prod_{i=1}^{3} \mathcal{L}_{\mathcal{P}_i}^T \right)^2. \tag{330}$$

Now assume that all $q_i$ are even. If $\mu_s$ for any puncture $i$ is non-zero, then we have

$$Z_{\mathcal{P}_1,\mathcal{P}_2,\mathcal{P}_3} = \mathbb{Z}_2 \tag{331}$$

and hence

$$\mathcal{L}_{\mathcal{P}_1,\mathcal{P}_2,\mathcal{P}_3} = \left( \prod_{i=1}^{3} \mathcal{L}^T_{\mathcal{P}_i} \times \mathbb{Z}_2 \right)^2 . \tag{332}$$

If $\mu_s = 0$ for all punctures, then we have

$$Z_{\mathcal{P}_1,\mathcal{P}_2,\mathcal{P}_3} = Z_G \tag{333}$$

and hence

$$\mathcal{L}_{\mathcal{P}_1,\mathcal{P}_2,\mathcal{P}_3} = \left( \prod_{i=1}^{3} \mathcal{L}^T_{\mathcal{P}_i} \times Z_G \right)^2 . \tag{334}$$

This can be matched with the defect group results obtained from the isolated hypersurface singularities, appearing in Tables 15, 16 and 20 of [29].

## 9 Twisted D

In this section, we describe our proposal for computing the defect groups associated to twisted punctures for $6d$ D-type $(2,0)$ theories. The analysis will largely be similar to that for untwisted D-type punctures, with only minor cosmetic differences. The key information that we use about these twisted punctures is discussed in subsection 9.1. Again, we not only consider conformal punctures of IHS type, but also numerous other classes of conformal and non-conformal punctures.

In subsection 9.2, we make our proposal. For each puncture, we assign a $4d$ $\mathcal{N} = 2$ generalized quiver gauge theory involving both perturbative and strongly coupled matter sectors, and propose that the defect groups associated to the generalized quiver gauge theory capture the defect groups of the associated punctures.

In subsection 9.2.1, we expand on our motivation for making the proposal discussed in subsection 9.2. We describe a large class of punctures $\mathcal{P}$ for which the $4d$ $\mathcal{N} = 2$ theories $\mathfrak{T}^{\mathcal{P}^o_0}_{\mathcal{P}}$ coincide with the proposed generalized quivers. A subclass of this class of punctures, whose associated generalized quivers involve only perturbative matter content, can be realized in terms of Hanany-Witten brane setups in IIA involving O4 planes.

Using the properties of the strongly coupled matter sectors derived in section 8.3, we derive in subsection 9.3 the defect groups associated to twisted D-type punctures. Like untwisted D-type punctures, we find that the twisted D-type punctures can also carry trapped 1-form symmetries.

Non-trivial checks of this proposal are provided using the IIB ALE fibrations in subsection 9.4. In subsection 9.5, we use the IHS realization to compute the trapped defect groups of IHS punctures, and find a perfect match with the trapped defect groups computed using our proposal.

### 9.1 Punctures

Let us consider twisted punctures for $6d$ $D_n$ $\mathcal{N} = (2,0)$ theory. These are described in terms of a Hitchin field in vector representation of $D_n$. The $i$-th block of the Hitchin field takes the form

$$B_i = \frac{1}{t^{1+r_i}} A_i + \cdots . \tag{335}$$

Again we order the blocks such that $r_i \geq r_{i+1}$ and let $s$ be biggest value of $i$ for which $r_i > 0$. For each $i \leq s$, we write

$$r_i = \frac{p_i}{q_i}, \tag{336}$$

where $1 \leq q_i \leq 2n - 2$ and $\gcd(p_i, q_i) = 1$.

Let us again define

$$n_i := 2n - \sum_{j=1}^{i} q_j. \tag{337}$$

We impose extra conditions on the punctures that can be described in terms of $n_i$ as follows. If $n_i$ is even (including $n_0 := 2n$) and $q_{i+1}$ is odd, then we require

$$q_{i+2} = q_{i+1}. \tag{338}$$

This in particular ensures that $n_{i-1}$ and $n_{i+1}$ are even if $n_i$ is odd. Furthermore, we require that $n_s$ is even. If $s$ is even then $n_s \geq 2$, and if $s$ is odd then $n_s \geq 0$.

As for untwisted A and D type punctures, the twisted D type punctures are also conformal only if $r_i = r_j$ for all $i, j \in \{1, 2, \cdots, s\}$. Out of these, the punctures of type IHS discussed in [89] are those for which $n_s = 0, 2$.

## 9.2 Class $\mathcal{GQ}$ of Generalized Quivers

Consider a puncture $\mathcal{P}$ of the above type. Then we conjecture that $\mathcal{P} \in \mathcal{F}_\tau$ where $\tau$ is a generalized quiver that can be constructed using the data of $n_i$ as follows. The quiver takes the form

$$\mathfrak{usp}(2n-2) \overline{\qquad} N_1 \overline{\qquad} N_2 \overline{\qquad} \cdots \overline{\qquad} N_{s-1} \overline{\qquad} [\mathfrak{g}_s]. \tag{339}$$

The node $N_i$ is either a gauge algebra or a matter SCFT, depending on whether $n_i$ is even or odd respectively. If $n_i$ is even, we write

$$N_i = \mathfrak{g}_i. \tag{340}$$

If $i$ is odd, we have

$$\mathfrak{g}_i = \mathfrak{so}(n_i) \tag{341}$$

and if $i$ is even, we have

$$\mathfrak{g}_i = \mathfrak{usp}(n_i - 2). \tag{342}$$

The same applies to the flavor algebra $\mathfrak{g}_s$. If $s$ is odd, we have

$$\mathfrak{g}_i = \mathfrak{so}(n_s) \tag{343}$$

and if $s$ is even, we have

$$\mathfrak{g}_i = \mathfrak{usp}(n_s - 2). \tag{344}$$

If $n_i$ is odd and $i$ is even, then

$$N_i = C(n_i), \tag{345}$$

where $C(n_i)$ is a matter SCFT discussed in detail in the previous section. If $n_i$ is odd and $i$ is odd, then

$$N_i = D(n_i), \tag{346}$$

where $D(n_i)$ is another matter SCFT discussed in the previous section.

The edges have the same meaning as discussed in section 8.2.

### 9.2.1 Class $\mathcal{GQ}'$ and Type IIA Punctures

The Class $\mathcal{GQ}'$ of punctures is given by punctures having $p_i = 1$ for all $1 \leq i \leq s$. The Type IIA punctures form a subclass of $\mathcal{GQ}'$ for which all $q_i$ and hence all $n_i$ are even. In such a situation, every node $N_i$ in (5.2) is a gauge algebra, which is $\mathfrak{so}(n_i)$ for $i$ odd and $\mathfrak{usp}(n_i - 2)$ for $i$ even. The Type IIA brane construction corresponding to a Type IIA puncture $\mathcal{P}'$ is

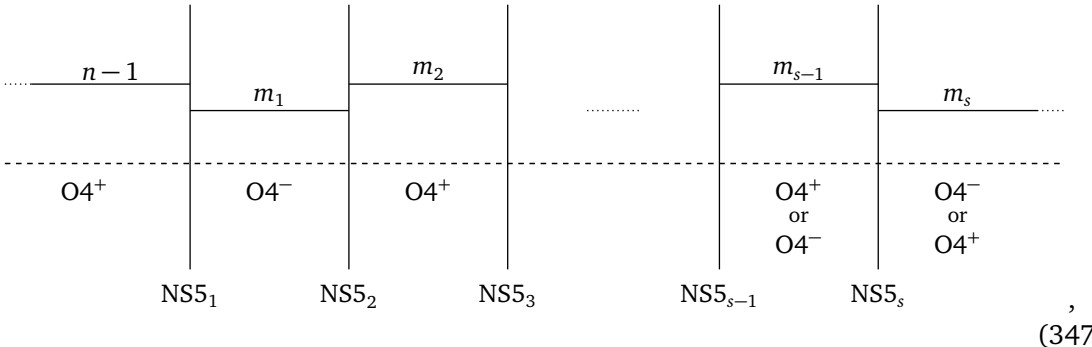

$$\tag{347}$$

where $m_i$ number of physical D4 branes have been placed in the $i^{\text{th}}$ interval, where $m_i = \frac{n_i}{2}$ for $i$ odd and $m_i = \frac{n_i}{2} - 1$ for $i$ even. Moreover, the $i^{\text{th}}$ interval carries an O4$^-$ plane if $i$ is odd and an O4$^+$ plane if $i$ is even.

### 9.3 1-Form Symmetries

As proposed above, we can compute the defect group associated to a twisted puncture $\mathcal{P}$ by computing the defect group associated to a generalized quiver of the form (339). We describe the result in terms of the quantities $\mu_1, \nu_1, \mu_2, \mu_3$ introduced in section 8.4. We also define a new quantity $\nu_s$ defined as $\nu_s := n_s$ if $s$ is odd, and $\nu_s := n_s - 2$ if $s$ is even. Then, we can divide the result into the following cases:

- Consider the case when $\mu_1 + \mu_3 + \nu_s > 0$. Then we have

$$
\begin{aligned}
\mathcal{H}_{\mathcal{P}}^T &= \mathbb{Z}_2^{\nu_1 + \mu_2}, \\
Z_{\mathcal{P}} &= 0, \\
\mathcal{H}_{\mathcal{P}} &= \mathcal{H}_{\mathcal{P}}^T \times Z_{\mathcal{P}},
\end{aligned}
\tag{348}
$$

with $i_{\mathcal{P}}$ mapping $\mathcal{H}_{\mathcal{P}}^T$ identically to the $\mathcal{H}_{\mathcal{P}}^T$ subfactor of $\mathcal{H}_{\mathcal{P}}$.

- Consider the case when $\mu_1 = \mu_3 = \nu_s = 0$, and $n_{2i+1} = 4m_{2i+1}$ for all integers $0 < i < \frac{s-1}{2}$. Then we have

$$
\begin{aligned}
\mathcal{H}_{\mathcal{P}}^T &= \mathbb{Z}_2^{\mu_2}, \\
Z_{\mathcal{P}} &= \mathbb{Z}_2, \\
\mathcal{H}_{\mathcal{P}} &= \mathcal{H}_{\mathcal{P}}^T \times Z_{\mathcal{P}},
\end{aligned}
\tag{349}
$$

with $i_{\mathcal{P}}$ mapping $\mathcal{H}_{\mathcal{P}}^T$ identically to the $\mathcal{H}_{\mathcal{P}}^T$ subfactor of $\mathcal{H}_{\mathcal{P}}$.

- Consider the case when $\mu_1 = \mu_3 = \nu_s = 0$, and $n_{2i+1} = 4m_{2i+1} + 2$ for at least one integer $i$ lying in the range $0 < i < \frac{s-1}{2}$. Then we have

$$
\begin{aligned}
\mathcal{H}_{\mathcal{P}}^T &= \mathbb{Z}_2^{\mu_2 - 1} \times \mathbb{Z}_2, \\
Z_{\mathcal{P}} &= \mathbb{Z}_2, \\
\mathcal{H}_{\mathcal{P}} &= \mathbb{Z}_2^{\mu_2 - 1} \times \mathbb{Z}_4.
\end{aligned}
\tag{350}
$$

Unlike the above cases, in this case, the short exact sequence

$$0 \rightarrow \mathcal{H}_{\mathcal{P}}^{T} \rightarrow \mathcal{H}_{\mathcal{P}} \rightarrow Z_{\mathcal{P}} \rightarrow 0 \tag{351}$$

does not split. The map $i_{\mathcal{P}}$ maps $\mathbb{Z}_2^{\mu_2-1}$ subfactor of $\mathcal{H}_{\mathcal{P}}^{T}$ identically to the $\mathbb{Z}_2^{\mu_2-1}$ subfactor of $\mathcal{H}_{\mathcal{P}}$, and the $\mathbb{Z}_2$ subfactor of $\mathcal{H}_{\mathcal{P}}^{T}$ to the $\mathbb{Z}_2$ subgroup of the $\mathbb{Z}_4$ subfactor of $\mathcal{H}_{\mathcal{P}}$. Correspondingly, the map $\pi_{\mathcal{P}}$ maps the $\mathbb{Z}_2^{\mu_2-1}$ subfactor of $\mathcal{H}_{\mathcal{P}}$ to zero in $Z_{\mathcal{P}}$, and the generator of the $\mathbb{Z}_4$ subfactor of $\mathcal{H}_{\mathcal{P}}$ to the generator of $Z_{\mathcal{P}} = \mathbb{Z}_2$.

## 9.4 Spectral Cover Monodromies

The behavior of the Higgs field near the puncture takes the form

$$F(v, t) = v^{2n} + \sum_{i \in \mathcal{I}} \frac{v^{n_i}}{t^{P_i}} + \frac{1}{t} = 0, \tag{352}$$

where $\mathcal{I}$ is the set of $1 \leq i \leq s$ for which $n_i$ is even and

$$P_j := \sum_{i=1}^{j} p_i. \tag{353}$$

The extra constant term $v^0 t^{-1}$ is added to resolve the distances between the sheets, which allows us to read the full monodromy.

Fix a point $t_0 \neq 0$ in the vicinity of the puncture. As discussed earlier, there are only $n$ independent values of $v$ solving $F(v, t_0) = 0$. We again identify a set of $n$ independent values of $v$ solving $F(v, t_0) = 0$ as $n$ of the weights $e_a$ for $1 \leq a \leq n$. The other $n$ values of $v$ solving $F(v, t_0) = 0$ are identified with the weights $-e_a$.

As one goes around the puncture $t_0 \rightarrow e^{2\pi i} t_0$, the $2n$ weights are permuted into each other. This induces an action on the roots, which can be identified as in (299) and (300). Let the action on the roots associated to a puncture $\mathcal{P}$ be denoted by an $n \times n$ matrix $\mathcal{M}_{\mathcal{P}}$. Then, as before, we have the identification

$$\mathcal{L}_{\mathcal{P}}^{0} = \text{Tor} \frac{\mathbb{Z}^n}{\mathcal{T}_{\mathcal{P}} \cdot \mathbb{Z}^n}, \tag{354}$$

where

$$\mathcal{T}_{\mathcal{P}} := \mathcal{M}_{\mathcal{P}} - 1. \tag{355}$$

**Example: $\mathcal{P}_0$ puncture.** Let us consider the example of $\mathcal{P}_0$ puncture, which has $s = 1, p_1 = 1, q_1 = 2n$. The curve is

$$v^{2n+2} + \frac{1}{t} = 0, \tag{356}$$

These are the same class of curves as for untwisted A-type gauge algebras in section 7.5. There is a $\mathbb{Z}_{2n+2}$ monodromy around the $\mathcal{P}_0$ puncture which permutes sheets cyclically. We pick a set of $n + 1$ pairwise linearly independent sheets and label these by weights $e_a$ of the $2n + 2$-dimensional vector representation such that these are permuted as

$$e_a \rightarrow e_{a+1}, \qquad e_{n+1} \rightarrow -e_1, \tag{357}$$

where $a = 1, \ldots, n$. With respect to this labelling the monodromy action on sheets leads to the following action on the roots represented by the monodromy matrix

$$M_{\mathcal{P}_0} = \begin{pmatrix} 0 & 0 & \cdots & 0 & 0 & 0 & 1 & -1 \\ 1 & 0 & \cdots & 0 & 0 & 0 & 1 & -1 \\ 0 & 1 & \ddots & 0 & 0 & 0 & 1 & -1 \\ \vdots & \ddots & \ddots & \ddots & \vdots & \vdots & \vdots & \vdots \\ 0 & 0 & \ddots & 1 & 0 & 0 & 1 & -1 \\ 0 & 0 & \cdots & 0 & 1 & 0 & 1 & -1 \\ 0 & 0 & \cdots & 0 & 0 & 1 & 0 & -1 \\ 0 & 0 & \cdots & 0 & 0 & 0 & 1 & 0 \end{pmatrix}. \tag{358}$$

From here we define

$$T_{\mathcal{P}_0} = M_{\mathcal{P}_0} - 1, \tag{359}$$

and the reader can check that

$$\mathrm{SNF}\left(T_{\mathcal{P}_0}\right) = \mathrm{diag}\left(2, 1, \ldots, 1\right), \tag{360}$$

implying that

$$\mathcal{L}_{\mathcal{P}_0}^0 = \mathbb{Z}_2. \tag{361}$$

**Example: Puncture with trapped defects and non-trivial extension.** Let us consider the puncture $\mathcal{P}'$ engineering the $\mathcal{G}\mathcal{Q}'$ quiver

$$[\mathfrak{usp}(6)] \relbar\joinrel\relbar \mathfrak{so}(6), \tag{362}$$

which has $s = 2, p_i = 1, q_1 = 2, q_2 = 4$ and the curve

$$v^8 + \frac{v^6 + 1}{t} + \frac{v^2}{t^2} = 0. \tag{363}$$

Here the term $1/t$ resolves a degeneracy at $v = 0$, it is the leading order contribution for generic Coulomb branch parameters. There is a $\mathbb{Z}_4 \times \mathbb{Z}_2^2$ monodromy around the $\mathcal{P}'$ puncture. We pick a set of 4 pairwise linearly independent sheets and label these by weights $e_a$ of the 8-dimensional vector representation such that these are permuted as

$$e_1 \to e_2 \to -e_1 \to -e_2, \qquad e_3 \to -e_3, \qquad e_4 \to -e_4. \tag{364}$$

With respect to this labelling the monodromy action on sheets leads to the action on the roots summarized by the monodromy matrix

$$M_{\mathcal{P}'} = \begin{pmatrix} 1 & -1 & 0 & 0 \\ 2 & -1 & 0 & 0 \\ 1 & 0 & -1 & 0 \\ 1 & 0 & 0 & -1 \end{pmatrix}. \tag{365}$$

From here we define

$$T_{\mathcal{P}'} = M_{\mathcal{P}'} - 1, \tag{366}$$

Table 6: Defect groups for the $x_1^2 + x_2^{l+1} + x_2 x_3^{n-1} + x_3 x_4^2 = 0$. $k$ increase left to right, $n$ increases top to bottom.

| $\mathrm{Tor}\, H_2(\partial X):\ n\backslash l$ | 1 | 2 | 3 | 4 | 5 | 6 | 7 | 8 | 9 | 10 | 11 | 12 | 13 | 14 |
|---|---|---|---|---|---|---|---|---|---|---|---|---|---|---|
| 3 | 0 | $\mathbb{Z}_2^2$ | 0 | 0 | 0 | $\mathbb{Z}_2^2$ | 0 | 0 | 0 | $\mathbb{Z}_2^2$ | 0 | 0 | 0 | $\mathbb{Z}_2^2$ |
| 4 | 0 | 0 | $\mathbb{Z}_2^2$ | 0 | 0 | 0 | 0 | 0 | $\mathbb{Z}_2^2$ | 0 | 0 | 0 | 0 | 0 |
| 5 | 0 | $\mathbb{Z}_2^2$ | 0 | $\mathbb{Z}_2^4$ | 0 | $\mathbb{Z}_2^2$ | 0 | 0 | 0 | $\mathbb{Z}_2^2$ | 0 | $\mathbb{Z}_2^4$ | 0 | $\mathbb{Z}_2^2$ |
| 6 | 0 | 0 | 0 | 0 | $\mathbb{Z}_2^4$ | 0 | 0 | 0 | 0 | 0 | 0 | 0 | 0 | 0 |
| 7 | 0 | $\mathbb{Z}_2^2$ | $\mathbb{Z}_2^2$ | 0 | 0 | $\mathbb{Z}_2^6$ | 0 | 0 | $\mathbb{Z}_2^2$ | $\mathbb{Z}_2^2$ | 0 | 0 | 0 | $\mathbb{Z}_2^2$ |
| 8 | 0 | 0 | 0 | 0 | 0 | 0 | $\mathbb{Z}_2^6$ | 0 | 0 | 0 | 0 | 0 | 0 | 0 |
| 9 | 0 | $\mathbb{Z}_2^2$ | 0 | $\mathbb{Z}_2^4$ | 0 | $\mathbb{Z}_2^2$ | 0 | $\mathbb{Z}_2^8$ | 0 | $\mathbb{Z}_2^2$ | 0 | $\mathbb{Z}_2^4$ | 0 | $\mathbb{Z}_2^2$ |
| 10 | 0 | 0 | $\mathbb{Z}_2^2$ | 0 | 0 | 0 | 0 | 0 | $\mathbb{Z}_2^8$ | 0 | 0 | 0 | 0 | 0 |

and the reader can check that

$$\mathrm{SNF}\,(T_{\mathcal{P}'}) = \mathrm{diag}\,(4,2,1,1)\,, \tag{367}$$

implying that

$$\mathcal{L}_{\mathcal{P}'}^0 = \mathbb{Z}_2 \times \mathbb{Z}_4\,. \tag{368}$$

## 9.5 Trapped Defect Group from Type IIB on CY3

For a puncture $\mathcal{P}$ of IHS type, we can compute its trapped defect group using an isolated hypersurface singularity $X_{\mathcal{P}}$ providing a check for our above proposal.

There are again two types of IHS that are relevant for twisted D type theories: the first is

$$X_{\{7,1\}(2,l+1,n-1,2)}:\qquad x_1^2 + x_2^{l+1} + x_2 x_3^{n-1} + x_3 x_4^2 = 0\,. \tag{369}$$

This singularity has associated to it $b = 2n - 2$ and $k = l + 2n - 2$. And the second is

$$X_{\{10,1\}(2,2,n-1,l+2)}:\qquad x_1^2 + x_2^2 x_3 + x_3^{n-1} x_4 + x_4^{l+2} x_2 = 0\,. \tag{370}$$

This singularity has associated to it $b = 2n$ and $k = l + n + 1$. The defect groups computed using IHS techniques are presented in tables 6 and 7.

Let us now find the trapped defect group for an IHS puncture according to our proposal outlined in the above subsections. An IHS puncture is characterized by two numbers $b, k$, where the possible values of $b$ are $2n, 2n - 2$. These values of $b$ correspond respectively to $n_s = 0, 2$. The number $k$ is any positive integer. Since the IHS punctures are conformal, they have the property that all $p_i$ are equal and all $q_i$ are equal. Let $r = r_i, p = p_i, q = q_i$. Then, we can write $r$ as

$$r = \frac{k}{b}\,, \tag{371}$$

for $b = 2n - 2$ and as

$$r = \frac{2k+1}{b}\,, \tag{372}$$

Table 7: Defect groups for the $x_1^2 + x_2^2 x_3 + x_3^{n-1} x_4 + x_4^{l+2} x_2 = 0$. $k$ increase left to right, $n$ increases top to bottom.

| $\operatorname{Tor} H_2(\partial X):\ n\backslash l$ | 1 | 2 | 3 | 4 | 5 | 6 | 7 | 8 | 9 | 10 | 11 | 12 | 13 |
|---|---|---|---|---|---|---|---|---|---|---|---|---|---|
| 3 | 0 | 0 | $\mathbb{Z}_2^2$ | 0 | 0 | $\mathbb{Z}_2^2$ | 0 | 0 | $\mathbb{Z}_2^2$ | 0 | 0 | $\mathbb{Z}_2^2$ | 0 |
| 4 | 0 | 0 | 0 | 0 | 0 | 0 | 0 | 0 | 0 | 0 | 0 | 0 | 0 |
| 5 | $\mathbb{Z}_2^4$ | 0 | 0 | 0 | 0 | $\mathbb{Z}_2^4$ | 0 | 0 | 0 | 0 | $\mathbb{Z}_2^4$ | 0 | 0 |
| 6 | 0 | 0 | $\mathbb{Z}_2^2$ | 0 | 0 | $\mathbb{Z}_2^2$ | 0 | 0 | $\mathbb{Z}_2^2$ | 0 | 0 | $\mathbb{Z}_2^2$ | 0 |
| 7 | 0 | $\mathbb{Z}_2^6$ | 0 | 0 | 0 | 0 | 0 | 0 | $\mathbb{Z}_2^6$ | 0 | 0 | 0 | 0 |
| 8 | 0 | 0 | 0 | 0 | 0 | 0 | 0 | 0 | 0 | 0 | 0 | 0 | 0 |
| 9 | 0 | 0 | $\mathbb{Z}_2^8$ | 0 | 0 | $\mathbb{Z}_2^2$ | 0 | 0 | $\mathbb{Z}_2^2$ | 0 | 0 | $\mathbb{Z}_2^8$ | 0 |
| 10 | $\mathbb{Z}_2^4$ | 0 | 0 | 0 | 0 | $\mathbb{Z}_2^4$ | 0 | 0 | 0 | 0 | $\mathbb{Z}_2^4$ | 0 | 0 |

for $b = 2n$. From this we find

$$
\begin{aligned}
p &= \frac{k}{\gcd(k, b)} & \text{if } b = 2n - 2\,, \\
p &= \frac{2k + 1}{\gcd(2k + 1, b)} & \text{if } b = 2n
\end{aligned}
\tag{373}
$$

and

$$
\begin{aligned}
q &= \frac{b}{\gcd(k, b)} & \text{if } b = 2n - 2\,, \\
q &= \frac{b}{\gcd(2k + 1, b)} & \text{if } b = 2n\,.
\end{aligned}
\tag{374}
$$

We also compute

$$
\begin{aligned}
s &= \gcd(k, b) & \text{if } b = 2n - 2\,, \\
s &= \gcd(2k + 1, b) & \text{if } b = 2n\,.
\end{aligned}
\tag{375}
$$

We divide the further analysis into the following three cases:

- Assume $q$ is odd. Then $\mu_1 = \mu_2 = 0$, so we find that

$$
\mathcal{L}_{\mathcal{P}}^T = 0\,.
\tag{376}
$$

- Assume $q$ is even. Then $\mu_1 = 0$ and $\mu_2 = \left\lfloor \frac{s+1}{2} \right\rfloor$. So, we find that

$$
\mathcal{L}_{\mathcal{P}}^T = \mathbb{Z}_2^{2\left\lfloor \frac{s}{2} \right\rfloor}\,.
\tag{377}
$$

This can be verified with the results appearing in tables 6 and 7.

## 10  Untwisted E

In this section, we derive the full defect groups associated to untwisted IHS punctures of E-type 6d $\mathcal{N} = (2, 0)$ theories. This is achieved by combining the two different string-theoretic

Table 8: Various contributions to the defect group for the untwisted punctures $\mathcal{P}_6^{(8)}[k]$ of type $E_6$ $\mathcal{N} = (2,0)$ theory. The columns parametrize different values of $k$. The results can be extended beyond this table as they only depend on $k$ (mod 8).

| $E_6^{(8)}[k]$ | 1 | 2 | 3 | 4 | 5 | 6 | 7 | 8 |
|---|---|---|---|---|---|---|---|---|
| $\mathcal{L}_{\mathcal{P}}^T$ | 0 | 0 | 0 | $\mathbb{Z}_2^2$ | 0 | 0 | 0 | 0 |
| $\mathcal{L}_{\mathcal{P}}^0$ | 0 | 0 | 0 | $\mathbb{Z}_2^2$ | 0 | 0 | 0 | 0 |
| $\widehat{Z}_{\mathcal{P}}$ | 0 | 0 | 0 | 0 | 0 | 0 | 0 | 0 |
| $\mathcal{W}_{\mathcal{P}}$ | 0 | 0 | 0 | $\mathbb{Z}_2$ | 0 | 0 | 0 | 0 |
| $\mathcal{W}_{\mathcal{P}}^T$ | 0 | 0 | 0 | $\mathbb{Z}_2$ | 0 | 0 | 0 | 0 |

constructions we have been using so far: namely the IHS construction, and the construction as ALE fibration with monodromy. Using the IHS description, we read the trapped defect group $\mathcal{L}_{\mathcal{P}}^T$ from which we deduce

$$\mathcal{H}_{\mathcal{P}}^T = \sqrt{\mathcal{L}_{\mathcal{P}}^T}. \tag{378}$$

On the other hand, using the ALE fibration description, we read the genuine part $\mathcal{L}_{\mathcal{P}}^0$ of the defect group $\mathcal{L}_{\mathcal{P}}$. Recall that $\mathcal{L}_{\mathcal{P}}^0$ must admit a decomposition as

$$\mathcal{L}_{\mathcal{P}}^0 = \mathcal{H}_{\mathcal{P}}^T \times \mathcal{W}_{\mathcal{P}}. \tag{379}$$

Combining it with the knowledge of $\mathcal{H}_{\mathcal{P}}^T$ from (378), we can read the group $\mathcal{W}_{\mathcal{P}}$. Thus, combining the two computations, we can deduce $\mathcal{W}_{\mathcal{P}}$ and $\mathcal{W}_{\mathcal{P}}^T = \widehat{\mathcal{H}}_{\mathcal{P}}^T$. The final data needed is the surjective map

$$\mathcal{W}_{\mathcal{P}} \to \mathcal{W}_{\mathcal{P}}^T, \tag{380}$$

which is apriori not determined even after combining the above two string-theoretic computations. However, luckily, for untwisted E-type IHS punctures, we find that $\mathcal{W}_{\mathcal{P}}$ and $\mathcal{W}_{\mathcal{P}}^T$ are such that there is a unique surjection (380). Moreover, this surjection is such that we can decompose

$$\mathcal{W}_{\mathcal{P}} = \widehat{Z}_{\mathcal{P}} \times \mathcal{W}_{\mathcal{P}}^T. \tag{381}$$

In other words, the short exact sequence

$$0 \to \widehat{Z}_{\mathcal{P}} \to \mathcal{W}_{\mathcal{P}} \to \mathcal{W}_{\mathcal{P}}^T \to 0 \tag{382}$$

splits, and hence the trapped and non-trapped parts do not mix with each other.

Note that for $E_8$, we do not need to perform the spectral cover computation because for any such puncture $\mathcal{P}$, we must have $\widehat{Z}_{\mathcal{P}} = 0$ as $Z_{E_8} = 0$. This implies that

$$\mathcal{L}_{\mathcal{P}}^0 = \mathcal{L}_{\mathcal{P}}^T, \tag{383}$$

which can be determined solely from the IHS computation.

Below, in subsection 10.1, we discuss how the weights and roots are encoded for $E_{6,7}$ spectral covers. Using this, we perform the computation of $\mathcal{L}_{\mathcal{P}}^0$ in subsection 10.2 for $E_6$, $E_7$ punctures. The data of $\mathcal{L}_{\mathcal{P}}^T$, $\mathcal{L}_{\mathcal{P}}^0$, $\widehat{Z}_{\mathcal{P}}$, $\mathcal{W}_{\mathcal{P}}$ and $\mathcal{W}_{\mathcal{P}}^T$ is given in tables 8, 9 and 10 for $E_6$ punctures; and in tables 11 and 12 for $E_7$ punctures. The data of $\mathcal{L}_{\mathcal{P}}^T$ is given in table13 for $E_8$ punctures. The short exact sequence (382) is then the trivial split sequence derived using this data.

Table 9: Various contributions to the defect group for the untwisted punctures $\mathcal{P}_6^{(9)}[k]$ of type $E_6$ $\mathcal{N} = (2,0)$ theory. The columns parametrize different values of $k$. The results can be extended beyond this table as they only depend on $k \pmod 9$.

| $E_6^{(9)}[k]$ | 1 | 2 | 3 | 4 | 5 | 6 | 7 | 8 | 9 |
|---|---|---|---|---|---|---|---|---|---|
| $\mathcal{L}_{\mathcal{P}}^T$ | 0 | 0 | $\mathbb{Z}_3^2$ | 0 | 0 | $\mathbb{Z}_3^2$ | 0 | 0 | 0 |
| $\mathcal{L}_{\mathcal{P}}^0$ | $\mathbb{Z}_3$ | $\mathbb{Z}_3$ | $\mathbb{Z}_3^3$ | $\mathbb{Z}_3$ | $\mathbb{Z}_3$ | $\mathbb{Z}_3^3$ | $\mathbb{Z}_3$ | $\mathbb{Z}_3$ | 0 |
| $\widehat{Z}_{\mathcal{P}}$ | $\mathbb{Z}_3$ | $\mathbb{Z}_3$ | $\mathbb{Z}_3$ | $\mathbb{Z}_3$ | $\mathbb{Z}_3$ | $\mathbb{Z}_3$ | $\mathbb{Z}_3$ | $\mathbb{Z}_3$ | 0 |
| $\mathcal{W}_{\mathcal{P}}$ | $\mathbb{Z}_3$ | $\mathbb{Z}_3$ | $\mathbb{Z}_3^2$ | $\mathbb{Z}_3$ | $\mathbb{Z}_3$ | $\mathbb{Z}_3^2$ | $\mathbb{Z}_3$ | $\mathbb{Z}_3$ | 0 |
| $\mathcal{W}_{\mathcal{P}}^T$ | 0 | 0 | $\mathbb{Z}_3$ | 0 | 0 | $\mathbb{Z}_3$ | 0 | 0 | 0 |

Table 10: Various contributions to the defect group for the untwisted punctures $\mathcal{P}_6^{(12)}[k]$ of type $E_6$ $\mathcal{N} = (2,0)$ theory. The columns parametrize different values of $k$. The results can be extended beyond this table as they only depend on $k \pmod{12}$.

| $E_6^{(12)}[k]$ | 1 | 2 | 3 | 4 | 5 | 6 | 7 | 8 | 9 | 10 | 11 | 12 |
|---|---|---|---|---|---|---|---|---|---|---|---|---|
| $\mathcal{L}_{\mathcal{P}}^T$ | 0 | 0 | 0 | $\mathbb{Z}_3^2$ | 0 | $\mathbb{Z}_2^2$ | 0 | $\mathbb{Z}_3^2$ | 0 | 0 | 0 | 0 |
| $\mathcal{L}_{\mathcal{P}}^0$ | $\mathbb{Z}_3$ | $\mathbb{Z}_3$ | 0 | $\mathbb{Z}_3^3$ | $\mathbb{Z}_3$ | $\mathbb{Z}_2^2$ | $\mathbb{Z}_3$ | $\mathbb{Z}_3^3$ | 0 | $\mathbb{Z}_3$ | $\mathbb{Z}_3$ | 0 |
| $\widehat{Z}_{\mathcal{P}}$ | $\mathbb{Z}_3$ | $\mathbb{Z}_3$ | 0 | $\mathbb{Z}_3$ | $\mathbb{Z}_3$ | 0 | $\mathbb{Z}_3$ | $\mathbb{Z}_3$ | 0 | $\mathbb{Z}_3$ | $\mathbb{Z}_3$ | 0 |
| $\mathcal{W}_{\mathcal{P}}$ | $\mathbb{Z}_3$ | $\mathbb{Z}_3$ | 0 | $\mathbb{Z}_3^2$ | $\mathbb{Z}_3$ | $\mathbb{Z}_2$ | $\mathbb{Z}_3$ | $\mathbb{Z}_3^2$ | 0 | $\mathbb{Z}_3$ | $\mathbb{Z}_3$ | 0 |
| $\mathcal{W}_{\mathcal{P}}^T$ | 0 | 0 | 0 | $\mathbb{Z}_3$ | 0 | $\mathbb{Z}_2$ | 0 | $\mathbb{Z}_3$ | 0 | 0 | 0 | 0 |

Table 11: Various contributions to the defect group for the untwisted punctures $\mathcal{P}_7^{(14)}[k]$ of type $E_7$ $\mathcal{N} = (2,0)$ theory. The columns parametrize different values of $k$. The results can be extended beyond this table as they only depend on $k \pmod{14}$.

| $E_7^{(14)}[k]$ | 1 | 2 | 3 | 4 | 5 | 6 | 7 | 8 | 9 | 10 | 11 | 12 | 13 | 14 |
|---|---|---|---|---|---|---|---|---|---|---|---|---|---|---|
| $\mathcal{L}_{\mathcal{P}}^T$ | 0 | 0 | 0 | 0 | 0 | 0 | $\mathbb{Z}_2^6$ | 0 | 0 | 0 | 0 | 0 | 0 | 0 |
| $\mathcal{L}_{\mathcal{P}}^0$ | $\mathbb{Z}_2$ | 0 | $\mathbb{Z}_2$ | 0 | $\mathbb{Z}_2$ | 0 | $\mathbb{Z}_2^7$ | 0 | $\mathbb{Z}_2$ | 0 | $\mathbb{Z}_2$ | 0 | $\mathbb{Z}_2$ | 0 |
| $\widehat{Z}_{\mathcal{P}}$ | $\mathbb{Z}_2$ | 0 | $\mathbb{Z}_2$ | 0 | $\mathbb{Z}_2$ | 0 | $\mathbb{Z}_2$ | 0 | $\mathbb{Z}_2$ | 0 | $\mathbb{Z}_2$ | 0 | $\mathbb{Z}_2$ | 0 |
| $\mathcal{W}_{\mathcal{P}}$ | $\mathbb{Z}_2$ | 0 | $\mathbb{Z}_2$ | 0 | $\mathbb{Z}_2$ | 0 | $\mathbb{Z}_2^4$ | 0 | $\mathbb{Z}_2$ | 0 | $\mathbb{Z}_2$ | 0 | $\mathbb{Z}_2$ | 0 |
| $\mathcal{W}_{\mathcal{P}}^T$ | 0 | 0 | 0 | 0 | 0 | 0 | $\mathbb{Z}_2^3$ | 0 | 0 | 0 | 0 | 0 | 0 | 0 |

Table 12: Various contributions to the defect group for the untwisted punctures $\mathcal{P}_7^{(18)}[k]$ of type $E_7$ $\mathcal{N}=(2,0)$ theory. The columns parametrize different values of $k$. The results can be extended beyond this table as they only depend on $k$ (mod 18).

| $E_7^{(18)}[k]$ | 1 | 2 | 3 | 4 | 5 | 6 | 7 | 8 | 9 | 10 | 11 | 12 | 13 | 14 | 15 | 16 | 17 | 18 |
|---|---|---|---|---|---|---|---|---|---|---|---|---|---|---|---|---|---|---|
| $\mathcal{L}_\mathcal{P}^T$ | 0 | 0 | 0 | 0 | 0 | $\mathbb{Z}_3^2$ | 0 | 0 | $\mathbb{Z}_2^6$ | 0 | 0 | $\mathbb{Z}_3^2$ | 0 | 0 | 0 | 0 | 0 | 0 |
| $\mathcal{L}_\mathcal{P}^0$ | $\mathbb{Z}_2$ | 0 | $\mathbb{Z}_2$ | 0 | $\mathbb{Z}_2$ | $\mathbb{Z}_3^2$ | $\mathbb{Z}_2$ | 0 | $\mathbb{Z}_2^7$ | 0 | $\mathbb{Z}_2$ | $\mathbb{Z}_3^2$ | $\mathbb{Z}_2$ | 0 | $\mathbb{Z}_2$ | 0 | $\mathbb{Z}_2$ | 0 |
| $\widehat{Z}_\mathcal{P}$ | $\mathbb{Z}_2$ | 0 | $\mathbb{Z}_2$ | 0 | $\mathbb{Z}_2$ | 0 | $\mathbb{Z}_2$ | 0 | $\mathbb{Z}_2$ | 0 | $\mathbb{Z}_2$ | 0 | $\mathbb{Z}_2$ | 0 | $\mathbb{Z}_2$ | 0 | $\mathbb{Z}_2$ | 0 |
| $\mathcal{W}_\mathcal{P}$ | $\mathbb{Z}_2$ | 0 | $\mathbb{Z}_2$ | 0 | $\mathbb{Z}_2$ | $\mathbb{Z}_3$ | $\mathbb{Z}_2$ | 0 | $\mathbb{Z}_2^4$ | 0 | $\mathbb{Z}_2$ | $\mathbb{Z}_3$ | $\mathbb{Z}_2$ | 0 | $\mathbb{Z}_2$ | 0 | $\mathbb{Z}_2$ | 0 |
| $\mathcal{W}_\mathcal{P}^T$ | 0 | 0 | 0 | 0 | 0 | $\mathbb{Z}_3$ | 0 | 0 | $\mathbb{Z}_2^3$ | 0 | 0 | $\mathbb{Z}_3$ | 0 | 0 | 0 | 0 | 0 | 0 |

Table 13: Trapped defect groups for untwisted punctures $\mathcal{P}_8^{(b)}[k]$ of type $E_8$ $\mathcal{N}=(2,0)$ theory. The columns parametrize different values of $k$. The results can be extended beyond this table as they only depend on $k$ (mod $b$).

| $\mathcal{L}_{\mathcal{P}_8^{(b)}[k]}^T$ | 1 | 2 | 3 | 4 | 5 | 6 | 7 | 8 | 9 | 10 | 11 | 12 | 13 | 14 | 15 |
|---|---|---|---|---|---|---|---|---|---|---|---|---|---|---|---|
| $b=20$ | 0 | 0 | 0 | $\mathbb{Z}_5^2$ | $\mathbb{Z}_2^4$ | 0 | 0 | $\mathbb{Z}_5^2$ | 0 | $\mathbb{Z}_2^8$ | 0 | $\mathbb{Z}_5^2$ | 0 | 0 | $\mathbb{Z}_2^4$ |
| $b=24$ | 0 | 0 | $\mathbb{Z}_2^2$ | 0 | 0 | $\mathbb{Z}_2^4$ | 0 | $\mathbb{Z}_3^4$ | $\mathbb{Z}_2^2$ | 0 | 0 | $\mathbb{Z}_2^8$ | 0 | 0 | $\mathbb{Z}_2^2$ |
| $b=30$ | 0 | 0 | 0 | 0 | 0 | $\mathbb{Z}_5^2$ | 0 | 0 | 0 | $\mathbb{Z}_3^4$ | 0 | $\mathbb{Z}_5^2$ | 0 | 0 | $\mathbb{Z}_2^8$ |
| Continued | 16 | 17 | 18 | 19 | 20 | 21 | 22 | 23 | 24 | 25 | 26 | 27 | 28 | 29 | 30 |
| $b=20$ | $\mathbb{Z}_5^2$ | 0 | 0 | 0 | 0 | 0 | 0 | 0 | $\mathbb{Z}_5^2$ | $\mathbb{Z}_2^4$ | 0 | 0 | $\mathbb{Z}_5^2$ | 0 | $\mathbb{Z}_2^8$ |
| $b=24$ | $\mathbb{Z}_3^4$ | 0 | $\mathbb{Z}_2^4$ | 0 | 0 | $\mathbb{Z}_2^2$ | 0 | 0 | 0 | 0 | 0 | $\mathbb{Z}_2^2$ | 0 | 0 | $\mathbb{Z}_2^4$ |
| $b=30$ | 0 | 0 | $\mathbb{Z}_5^2$ | 0 | $\mathbb{Z}_3^4$ | 0 | 0 | 0 | $\mathbb{Z}_5^2$ | 0 | 0 | 0 | 0 | 0 | 0 |

## 10.1 Spectral Covers and Weights Systems

$E$-type spectral curves are discussed in [112–114]. The $E_6$ spectral curve with respect to the representation **27** has six Casimirs

$$u_2, u_5, u_6, u_8, u_9, u_{12} \tag{384}$$

and the equation

$$F_{E_6}(v, t) = \frac{1}{2} v^3 u_{12}^2 - Q(v) u_{12} + \frac{1}{2v^3} \left[ Q(v)^2 - P_1(v)^2 P_2(v) \right] = 0. \tag{385}$$

Here the polynomials $P_1, P_2, Q$ are

$$
\begin{aligned}
P_1(v) &= 78v^{10} + 60v^8 u_2 + 14v^6 u_2^2 - 33v^5 u_5 + 2v^4 u_{12} - 5v^3 u_2 u_5 - v^2 u_8 - v u_9 - u_5^2, \\
P_2(v) &= 12v^{10} + 12v^8 u_2 + 4v^6 u_2^2 - 12v^5 u_5 + v^4 u_6 - 4v^3 u_2 u_5 - 2v^2 u_8 + 4v u_9 + u_5^2, \\
Q(v) &= 270v^{15} + 342v^{13} u_2 + 162v^{11} u_2^2 - 252v^{10} u_5 + v^9 \left( 26u_2^3 + 18u_6 \right) - 162v^8 u_2 u_5 \\
&\quad + v^7 (6u_2 u_6 - 27u_8) - v^6 \left( 30u_2^2 u_5 - 36u_9 \right) + v^5 \left( 27u_5^2 - 9u_2 u_8 \right) \\
&\quad - v^4 (3u_5 u_6 - 6u_2 u_9) - 3v^3 u2 u_5^2 - 3v u_5 u_9 - u_5^3.
\end{aligned} \tag{386}
$$

The spectral curve has $\deg F_{E_6} = 27$ sheets $v_i$ and for any value of the Casimirs the sheets group into 45 triplets $[i, j, k]$ such that each sheet appears in 5 triplets with each triplet satisfying

$$v_i + v_j + v_k = 0. \tag{387}$$

The weights $w_i$ of the representation **27** of $E_6$ also group into 45 triplets $[i, j, k]$ with each weight appearing in 5 triplets and each triplet satisfying

$$w_i + w_j + w_k = 0. \tag{388}$$

Sheets of the spectral curve are labelled by weights such that these structures match. The labelling is unique up to Weyl transformations.

The $E_7$ spectral curve with respect to the representation **56** has seven Casimirs

$$u_2, u_6, u_8, u_{10}, u_{12}, u_{14}, u_{18} \tag{389}$$

and the equation

$$F_{E_7}(v, t) = -\frac{1}{729} v^2 \left[ u_{18}^3 + A_2(v) u_{18}^2 + A_1(v) u_{18} + A_0(v) \right]. \tag{390}$$

The polynomials $A_0, A_1, A_2$ are

$$
\begin{aligned}
A_0(v) &= -\left( \frac{9}{16v^2} \right)^3 \Big[ 4R(v)^2 P_1(v)^3 + 6Q(v) R(v) P_1(v)^2 P_2(v) + 9Q(v)^2 P_1(v)^2 P_3(v) \\
&\quad - 6R(v) P_1(v) P_2(v) P_3(v) - 6Q P_1(v) P_3(v)^2 + 2R(v) P_2(v)^3 \\
&\quad + 3Q(v) P_2(v)^2 P_3 + P_3(v)^3 \Big], \\
A_1(v) &= \left( \frac{9}{16v^2} \right)^2 \Big[ 9Q(v)^2 P_1(v)^2 - 6R(v) P_1(v) P_2(v) - 12Q(v) P_1(v) P_3(v) \\
&\quad + 3Q(v) P_2(v)^2 + 3P_3(v)^2 \Big], \\
A_2(v) &= \frac{9}{16v^2} \Big[ 6Q(v) P_1(v) - 3P_3(v) \Big].
\end{aligned} \tag{391}
$$

Here the polynomials $P_1, P_2, P_3, Q, R$ are

$$P_1(v) = -u_{10} - 2u_8 v^2 + 7u_6 v^4 + 88u_2^2 v^6 + 660u_2 v^8 + 1596v^{10},$$

$$P_2(v) = \left(\frac{218}{8613}u_2 u_6^2 - \frac{4}{3}u_{14}\right)v + \left(\frac{2}{9}u_6^2 - \frac{4}{3}u_{12}\right)v^3 + \left(68u_{10} - \frac{68}{3}u_2 u_8\right)v^5$$

$$\left(\frac{100}{3}u_2 u_6 - 100u_8\right)v^7 + \left(264u_2^3 + 176u_6\right)v^9$$

$$+ 2952u_2^2 v^{11} + 11368u_2 v^{13} + 16872v^{15},$$

$$P_3(v) = u_{10}^2 + \frac{4}{3}u_8 u_{10} v^2 + \left(-\frac{2}{3}u_6 u_{10} - \frac{64}{3}u_{14}u_2 + \frac{3488}{8613}u_2^2 u_6^2 + \frac{4}{9}u_8^2\right)v^4$$

$$+ \left(-16u_2^2 u_{10} - \frac{32}{9}u_2 u_{12} - \frac{416}{3}u_{14} + \frac{27776}{8613}u_2 u_6^2 - \frac{4}{9}u_6 u_8\right)v^6$$

$$+ \left(-40u_2 u_{10} - \frac{64}{3}u_{12} + 32u_2^2 u_6 + \frac{11}{3}u_6^2\right)v^8$$

$$+ \left(192u_2^4 + 312u_2 u_6 - \frac{1552}{3}u_8\right)v^{12} + \left(2880u_2^3 + \frac{2216}{3}u_6\right)v^{14}$$

$$+ 16080u_2^2 v^{16} + 41568u_2 v^{18} + 44560v^{20},$$

$$Q(v) = -\frac{1}{3}u_{10} + \frac{2}{9}u_8 v^2 - \frac{1}{3}u_6 v^4 - \frac{8}{3}u_2^2 v^6 - \frac{44}{3}u_2 v^8 - 28v^{10},$$

$$R(v) = \left(\frac{109}{8613}u_2 u_6^2 - \frac{2}{3}u_{14}\right)v + \left(2u_{10} - \frac{2}{3}u_2 u_8\right)v^5 + \left(4u_2^3 + \frac{8}{3}u_6\right)v^9$$

$$+ \left(\frac{2}{3}u_2 u_6 - 2u_8\right)v^7 + 36u_2^2 v^{11} + \left(\frac{1}{81}u_6^2 - \frac{2}{27}u_{12} + 116u_2\right)v^{13} + 148v^{15}.$$

$$(392)$$

The spectral curve has $\deg F_{E_7} = 56$ sheets $v_i$ and for any value of the Casimirs the sheets group into 630 quadruplets $[i, j, k, l]$ such that each sheet appears in 45 quadruplets with each quadruplets satisfying

$$v_i + v_j + v_k + v_l = 0, \tag{393}$$

and no pair of sheets in the same quadruplet summing to zero. Further there are 28 pairs $[i, j]$ such that

$$v_i + v_j = 0. \tag{394}$$

Similarly the weights $w_i$ of the representation **56** of $E_7$ group into 630 quadruplets and 28 pairs and permitted labelings of sheets by weights are such that these two structures match. Such labelings are unique up to Weyl transformations. Further details are discussed in the appendices of [112].

## 10.2  Spectral Cover Monodromies

In the spectral cover descriptions punctures are specified by boundary conditions on the Casimirs. We consider punctures localized at $t = 0$ and labelled by two integers $(b, k)$ such that

$$u_b = \frac{1}{t^k} \tag{395}$$

constitutes the leading order behaviour in the limit $t \to 0$ among the Casimirs $u_i$ with all other Casimirs subleading. We denote these punctures by $\mathcal{P}_n^{(b)}[k]$ with $n = 6, 7, 8$ for $E_6, E_7, E_8$ respectively.

The monodromy on weights informs the mondromy on simple roots, see section 5.3, and we denote the associated monodromy matrix by $M_{\mathcal{P}_n^{(b)}[k]}$. From (395) it now follows that the monodromy matrices of the different punctures are related as

$$M_{\mathcal{P}_n^{(b)}[k]} = M_{\mathcal{P}_n^{(b)}[1]}^k \tag{396}$$

and with this we compute the group of genuine lines to be

$$\mathcal{L}^0_{\mathcal{P}_n^{(b)}[k]} = \text{Tor coker}\left(M_{\mathcal{P}_n^{(b)}[1]}^k - 1\right) \tag{397}$$

and collect the results in tables 8–12.

**Example:** $\mathcal{P}_6^{(12)}[1]$ **puncture.** Compactification of 6d $\mathcal{N} = (2,0)$ theory with $\mathfrak{g} = \mathfrak{e}_6$ on a sphere with two $\mathcal{P}_6^{(12)}[1]$ punctures engineers 4d $\mathcal{N} = 2$ SYM with $\mathfrak{g} = \mathfrak{e}_6$. This puncture therefore plays the same role as $\mathcal{P}_0$ punctures for classical bulk algebras of type A and D. To study the puncture $\mathcal{P}_6^{(12)}[1]$ we set $u_{12} = 1/t$ and $u_{2,5,6,8} = 0$ and $u_9 = 1$ in the $E_6$ spectral curve (385) which then becomes

$$0 = v^{27} + 28v^{18} + \frac{5v^{15}}{t} - \frac{53v^9}{3} + \frac{2v^6}{3t} - \frac{v^3}{108t^2} + \frac{1}{27}, \tag{398}$$

where we have moved onto the Coulomb branch with $u_9 \neq 0$ to resolve a degree three degeneracy with $v = 0$. Next we label the sheets $v_i$ by weights $w_i$. For this we compute the roots at $t = 1$, we depict these as crosses in

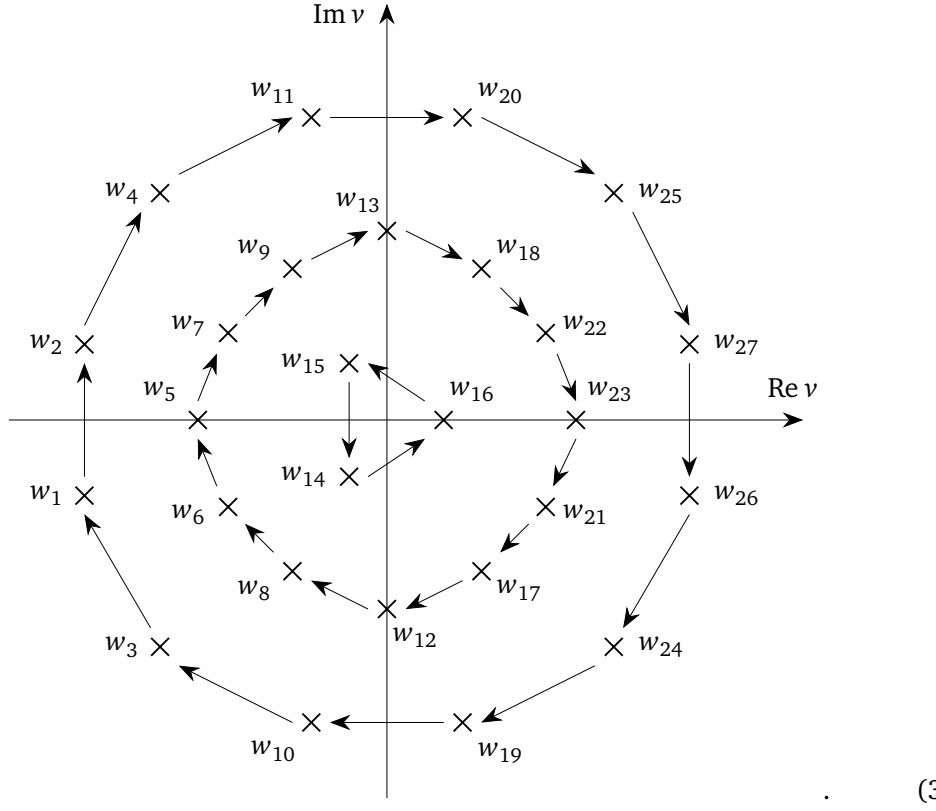

$$\tag{399}$$

Here we have further displayed the monodromy action on sheets upon encircling the puncture positively and given a labelling consistent with (387) and (388). See appendix B for `Mathematica` code used to study the monodromies of (398). The weights and their vanishing triples are given explicitly in table 14. The monodromy on roots is computed with this labelling to

$$M_{\mathcal{P}_6^{(12)}[1]} = \begin{pmatrix} 0 & 0 & -1 & 1 & 0 & 0 \\ 0 & -1 & 0 & 1 & 0 & 0 \\ 1 & 0 & -1 & 1 & 0 & 0 \\ 1 & -1 & -1 & 2 & -1 & 1 \\ 0 & 0 & 0 & 1 & -1 & 1 \\ 0 & 0 & 0 & 1 & -1 & 0 \end{pmatrix}, \tag{400}$$

from which we compute

$$\mathcal{L}^0_{\mathcal{P}_6^{(12)}[1]} = \mathrm{Tor}\,\mathrm{coker}\left(M_{\mathcal{P}_6^{(12)}[1]} - 1\right) \cong \mathbb{Z}_3\,, \tag{401}$$

which is the center of $E_6$ matching the field theory expectation.

## 10.3 Trapped Defect Group from Type IIB on CY3

In [81] the superconformal theories $\mathfrak{T}_{\mathcal{P}_n^{(b)}[k]}$ engineered from the punctures $\mathcal{P}_n^{(b)}[k]$ were denoted by $E_n^{(b)}[k]$. These theories admit an IHS realization in IIB (see also table 2)

$$\begin{aligned}
E_6^{(8)}[k] &= X_{\{2,1\}(2,4,3,\kappa)} : & 0 &= u^2 + x^3 + y^4 + xz^\kappa\,, \\
E_6^{(9)}[k] &= X_{\{2,1\}(2,3,4,\kappa)} : & 0 &= u^2 + x^3 + y^4 + yz^\kappa\,, \\
E_6^{(12)}[k] &= X_{\{1,1\}(2,3,4,\kappa)} : & 0 &= u^2 + x^3 + y^4 + z^\kappa\,, \\
E_7^{(14)}[k] &= X_{\{7,1\}(2,3,3,\kappa)} : & 0 &= u^2 + x^3 + xy^3 + yz^\kappa\,, \\
E_7^{(18)}[k] &= X_{\{2,1\}(2,\kappa,3,3)} : & 0 &= u^2 + x^3 + xy^3 + z^\kappa\,, \\
E_8^{(20)}[k] &= X_{\{2,1\}(2,5,3,\kappa)} : & 0 &= u^2 + x^3 + y^5 + xz^\kappa\,, \\
E_8^{(24)}[k] &= X_{\{2,1\}(2,3,5,\kappa)} : & 0 &= u^2 + x^3 + y^5 + yz^\kappa\,, \\
E_8^{(30)}[k] &= X_{\{1,1\}(2,3,5,\kappa)} : & 0 &= u^2 + x^3 + y^5 + z^\kappa\,.
\end{aligned} \tag{402}$$

In these equations

$$\kappa = k - b\,. \tag{403}$$

Again the alternate nomenclature is the one used in [55]. From the hypersurface we compute the trapped defect group via the boundary topology of the associated Calabi-Yau threefold $X_n^{(b)}[k]$ using the methods in [55]. These are in agreement (for low values of $k$) with the trapped contributions in tables 8–13. Note, that the rows of the tables are periodic with $k \sim k + b$.

# 11 Conclusions and Outlook

The main goal of this work is to study the 1-form symmetry groups and line defect groups of arbitrary 4d $\mathcal{N} = 2$ Class S theories. This problem was recently tackled extensively for the

Table 14: Weights of the $E_6$ representation **27** as given by `SAGE`. Weights are represented with respect to their embedding into the weight lattice of $E_8$ (Ambient Space). We also give the presentations of simple roots $\alpha_i$ and their decomposition into weights.

| **27** | Weights (Ambient Space) | Vanishing Triples |
|---|---|---|
| $w_1$ | $\left(0,0,0,0,0,-\frac{2}{3},-\frac{2}{3},+\frac{2}{3}\right)$ | $\{[1,13,26],[1,14,27],[1,17,25],[1,20,24],[1,22,23]\}$ |
| $w_2$ | $\left(-\frac{1}{2},+\frac{1}{2},+\frac{1}{2},+\frac{1}{2},+\frac{1}{2},-\frac{1}{6},-\frac{1}{6},+\frac{1}{6}\right)$ | $\{[2,12,27],[2,15,26],[2,18,24],[2,19,25],[2,21,23]\}$ |
| $w_3$ | $\left(+\frac{1}{2},-\frac{1}{2},+\frac{1}{2},+\frac{1}{2},+\frac{1}{2},-\frac{1}{6},-\frac{1}{6},+\frac{1}{6}\right)$ | $\{[3,9,27],[3,11,26],[3,16,25],[3,18,22],[3,20,21]\}$ |
| $w_4$ | $\left(+\frac{1}{2},+\frac{1}{2},-\frac{1}{2},+\frac{1}{2},+\frac{1}{2},-\frac{1}{6},-\frac{1}{6},+\frac{1}{6}\right)$ | $\{[4,8,26],[4,10,27],[4,16,24],[4,17,21],[4,19,22]\}$ |
| $w_5$ | $\left(-\frac{1}{2},-\frac{1}{2},-\frac{1}{2},+\frac{1}{2},+\frac{1}{2},-\frac{1}{6},-\frac{1}{6},+\frac{1}{6}\right)$ | $\{[5,6,27],[5,7,26],[5,16,23],[5,17,18],[5,19,20]\}$ |
| $w_6$ | $\left(+\frac{1}{2},+\frac{1}{2},+\frac{1}{2},-\frac{1}{2},+\frac{1}{2},-\frac{1}{6},-\frac{1}{6},+\frac{1}{6}\right)$ | $\{[5,6,27],[6,8,25],[6,11,24],[6,13,21],[6,15,22]\}$ |
| $w_7$ | $\left(+\frac{1}{2},+\frac{1}{2},+\frac{1}{2},+\frac{1}{2},-\frac{1}{2},-\frac{1}{6},-\frac{1}{6},+\frac{1}{6}\right)$ | $\{[5,7,26],[7,9,24],[7,10,25],[7,12,22],[7,14,21]\}$ |
| $w_8$ | $\left(-\frac{1}{2},-\frac{1}{2},+\frac{1}{2},+\frac{1}{2},-\frac{1}{2},-\frac{1}{6},-\frac{1}{6},+\frac{1}{6}\right)$ | $\{[4,8,26],[6,8,25],[8,9,23],[8,12,20],[8,14,18]\}$ |
| $w_9$ | $\left(-\frac{1}{2},+\frac{1}{2},-\frac{1}{2},-\frac{1}{2},+\frac{1}{2},-\frac{1}{6},-\frac{1}{6},+\frac{1}{6}\right)$ | $\{[3,9,27],[7,9,24],[8,9,23],[9,13,19],[9,15,17]\}$ |
| $w_{10}$ | $\left(-\frac{1}{2},-\frac{1}{2},+\frac{1}{2},-\frac{1}{2},+\frac{1}{2},-\frac{1}{6},-\frac{1}{6},+\frac{1}{6}\right)$ | $\{[4,10,27],[7,10,25],[10,11,23],[10,13,18],[10,15,20]\}$ |
| $w_{11}$ | $\left(-\frac{1}{2},+\frac{1}{2},-\frac{1}{2},+\frac{1}{2},-\frac{1}{2},-\frac{1}{6},-\frac{1}{6},+\frac{1}{6}\right)$ | $\{[3,11,26],[6,11,24],[10,11,23],[11,12,17],[11,14,19]\}$ |
| $w_{12}$ | $\left(+\frac{1}{2},-\frac{1}{2},-\frac{1}{2},-\frac{1}{2},+\frac{1}{2},-\frac{1}{6},-\frac{1}{6},+\frac{1}{6}\right)$ | $\{[2,12,27],[7,12,22],[8,12,20],[11,12,17],[12,13,16]\}$ |
| $w_{13}$ | $\left(0,0,0,+1,0,+\frac{1}{3},+\frac{1}{3},-\frac{1}{3}\right)$ | $\{[1,13,26],[6,13,21],[9,13,19],[10,13,18],[12,13,16]\}$ |
| $w_{14}$ | $\left(0,0,0,0,+1,+\frac{1}{3},+\frac{1}{3},-\frac{1}{3}\right)$ | $\{[1,14,27],[7,14,21],[8,14,18],[11,14,19],[14,15,16]\}$ |
| $w_{15}$ | $\left(+\frac{1}{2},-\frac{1}{2},-\frac{1}{2},+\frac{1}{2},-\frac{1}{2},-\frac{1}{6},-\frac{1}{6},+\frac{1}{6}\right)$ | $\{[2,15,26],[6,15,22],[9,15,17],[10,15,20],[14,15,16]\}$ |
| $w_{16}$ | $\left(-\frac{1}{2},+\frac{1}{2},+\frac{1}{2},-\frac{1}{2},-\frac{1}{2},-\frac{1}{6},-\frac{1}{6},+\frac{1}{6}\right)$ | $\{[3,16,25],[4,16,24],[5,16,23],[12,13,16],[14,15,16]\}$ |
| $w_{17}$ | $\left(0,0,+1,0,0,+\frac{1}{3},+\frac{1}{3},-\frac{1}{3}\right)$ | $\{[1,17,25],[4,17,21],[5,17,18],[9,15,17],[11,12,17]\}$ |
| $w_{18}$ | $\left(+\frac{1}{2},+\frac{1}{2},-\frac{1}{2},-\frac{1}{2},-\frac{1}{2},-\frac{1}{6},-\frac{1}{6},+\frac{1}{6}\right)$ | $\{[2,18,24],[3,18,22],[5,17,18],[8,14,18],[10,13,18]\}$ |
| $w_{19}$ | $\left(+\frac{1}{2},-\frac{1}{2},+\frac{1}{2},-\frac{1}{2},-\frac{1}{2},-\frac{1}{6},-\frac{1}{6},+\frac{1}{6}\right)$ | $\{[2,19,25],[4,19,22],[5,19,20],[9,13,19],[11,14,19]\}$ |
| $w_{20}$ | $\left(0,+1,0,0,0,+\frac{1}{3},+\frac{1}{3},-\frac{1}{3}\right)$ | $\{[1,20,24],[3,20,21],[5,19,20],[8,12,20],[10,15,20]\}$ |
| $w_{21}$ | $\left(-\frac{1}{2},-\frac{1}{2},-\frac{1}{2},-\frac{1}{2},-\frac{1}{2},-\frac{1}{6},-\frac{1}{6},+\frac{1}{6}\right)$ | $\{[2,21,23],[3,20,21],[4,17,21],[6,13,21],[7,14,21]\}$ |
| $w_{22}$ | $\left(-1,0,0,0,0,+\frac{1}{3},+\frac{1}{3},-\frac{1}{3}\right)$ | $\{[1,22,23],[3,18,22],[4,19,22],[6,15,22],[7,12,22]\}$ |
| $w_{23}$ | $\left(+1,0,0,0,0,+\frac{1}{3},+\frac{1}{3},-\frac{1}{3}\right)$ | $\{[1,22,23],[2,21,23],[5,16,23],[8,9,23],[10,11,23]\}$ |
| $w_{24}$ | $\left(0,-1,0,0,0,+\frac{1}{3},+\frac{1}{3},-\frac{1}{3}\right)$ | $\{[1,20,24],[2,18,24],[4,16,24],[6,11,24],[7,9,24]\}$ |
| $w_{25}$ | $\left(0,0,-1,0,0,+\frac{1}{3},+\frac{1}{3},-\frac{1}{3}\right)$ | $\{[1,17,25],[2,19,25],[3,16,25],[6,8,25],[7,10,25]\}$ |
| $w_{26}$ | $\left(0,0,0,-1,0,+\frac{1}{3},+\frac{1}{3},-\frac{1}{3}\right)$ | $\{[1,13,26],[2,15,26],[3,11,26],[4,8,26],[5,7,26]\}$ |
| $w_{27}$ | $\left(0,0,0,0,-1,+\frac{1}{3},+\frac{1}{3},-\frac{1}{3}\right)$ | $\{[1,14,27],[3,9,27],[4,10,27],[5,6,27],[9,15,27]\}$ |
| | Simple Roots | Simple Roots from Weights |
| $\alpha_1$ | $\left(+\frac{1}{2},-\frac{1}{2},-\frac{1}{2},-\frac{1}{2},-\frac{1}{2},-\frac{1}{2},-\frac{1}{2},+\frac{1}{2}\right)$ | $-2w_{22}-\frac{3}{2}w_{23}+\frac{1}{2}(w_{24}+w_{25}+w_{26}+w_{27})$ |
| $\alpha_2$ | $(+1,+1,0,0,0,0,0,0)$ | $w_{23}-w_{24}$ |
| $\alpha_3$ | $(-1,+1,0,0,0,0,0,0)$ | $w_{22}-w_{24}$ |
| $\alpha_4$ | $(0,-1,+1,0,0,0,0,0)$ | $w_{24}-w_{25}$ |
| $\alpha_5$ | $(0,0,-1,+1,0,0,0,0)$ | $w_{25}-w_{26}$ |
| $\alpha_6$ | $(0,0,0,-1,+1,0,0,0)$ | $w_{26}-w_{27}$ |

case of regular punctures in [31], and it was shown that these groups can be encoded in the topological properties of 1-cycles on the Riemann surface used for Class S construction.

In the present work, we include irregular punctures into the analysis. Once irregular punctures are included, one quickly realizes, using alternative descriptions of the 4d theories as discussed in section 3, that there can be extra contributions to the 1-form symmetry and line defect groups of the Class S theory that are not accounted by 1-cycles on the Riemann surface. These additional contributions have to be somehow trapped at the locations of the irregular punctures. Indeed, as we show in this paper, there is a non-trivial line defect group associated to irregular punctures that contributes to the line defect group of a Class S theory.

This insight from Class S implies a more general, conceptual point: the existence of relative defects in relative theories.

It is well-known, and we review in sections 2.1 and 2.2, that a non-trivial defect group indicates that the theory under study is a relative theory attached to a TQFT in one higher dimension. In the same way, as explained in detail in section 2, the fact that a puncture of a 6d $\mathcal{N} = (2, 0)$ theory carries a defect group means that it is a relative codimension-two defect attached to a topological system in one higher dimension. Since 6d $\mathcal{N} = (2, 0)$ theory is a relative theory, we learn that a general codimension-two defect of a 6d $\mathcal{N} = (2, 0)$ theory is a relative defect inside a relative theory. This has the interesting consequence that the topological system attached to the relative codimension-two defect is itself a topological defect of the TQFT attached to the 6d $\mathcal{N} = (2, 0)$ theory.

We discuss the general formalism of relative defects in relative theories in section 2. The results of this paper provide interesting and concrete examples of such relative defects in the form of codimension-two defects of 6d $\mathcal{N} = (2, 0)$ theories.

The full line defect group of a puncture can be divided into trapped and non-trapped parts. The non-trapped part is related to the surface defect group of the parent 6d $\mathcal{N} = (2, 0)$ theory, while the trapped part is not related to the bulk surface defect group, but arises as a contribution from the puncture (i.e. codimension two defect). It is important to know how the trapped and non-trapped parts combine to form the full line defect group associated to a puncture. The combination is captured by a short exact sequence, which is non-split whenever the trapped and non-trapped parts combine non-trivially. The details about the structure of line defect groups of punctures, and its usage in the computation of the line defect group of an arbitrary Class S theory, are discussed in section 4.

We obtain explicit expressions for the line defect groups of many classes of conformal and non-conformal untwisted A- and D-types, and twisted D-type punctures. This is done by proposing that the line defect groups of punctures are the same as line defect groups of a class of 4d $\mathcal{N} = 2$ generalized quiver gauge theories.

Many of the classes of punctures that we discuss (even the conformal ones) have not appeared prior in the literature. A sub-class of these new punctures can be constructed using Hanany-Witten brane setups in Type IIA string theory, and generalizations thereof.

We use two geometric constructions of punctures in Type IIB string theory to test the proposed defect groups. Using such geometric constructions, we can compute important pieces of the full defect groups of the punctures, which can then be matched with the proposals. One of these constructions employs the use of isolated hypersurface Calabi-Yau threefold singularities in IIB, using which we can compute trapped parts of the defect groups associated to punctures. The other construction involves ALE fibrations over a punctured complex plane. The monodromies of these ALE fibrations can be used to compute the genuine part of the defect group associated to the punctures.

Combining these two geometric constructions, we are able to also derive defect groups associated to a class of well-known conformal untwisted E-type punctures.

Conceptually, we expect the ALE-fibration to contain the same information as the Class S

construction. What we have shown in this paper, is how to geometrically extract the genuine part of the defect group associated to the punctures. However, as we explained in section 5.3, we do not understand how to combine this and describe the full defect group of the puncture, which is locally modelled by the ALE-fibration. It would be very interesting to develop this line of reasoning further.

In this paper we focused on codimension 2 defects in Class S constructions. Naturally, this should have extensions to $\mathcal{N} = 1$ versions of Class S theories, and potentially interesting implications for studies of confinement. More broadly, we expect relative defects in relative theories to also play a role in other contexts of compactifications from higher-dimensional relative theories, e.g. 6d to 3d and 2d.

## Acknowledgements

We thank Cyril Closset, Iñaki García Etxebarria, Dave Morrison and Yinan Wang for discussions. This work is supported in part by the European Union's Horizon 2020 Framework through the ERC grants 682608 (LB, SG and SSN) and 787185 (LB). The work of SG is also supported by the INFN grant "Per attività di formazione per sostenere progetti di ricerca" (GRANT 73/STRONGQFT). MH is funded and SSN is supported in part by the "Simons Collaboration on Special Holonomy in Geometry, Analysis and Physics".

## A Glossary

### A.1 Glossary for Sections 2.1 and 2.2

- $\mathfrak{D}_P$: A $p$-dimensional and generically non-topological defect.

- $\mathfrak{T}_d$: A $d$-dimensional relative theory.

- $\mathfrak{S}_{d+1}$: A $(d+1)$-dimensional non-invertible TQFT associated to a $d$-dimensional relative theory.

- $\mathcal{L}_p$: The group formed by invertible $p$-dimensional topological defects in a TQFT $\mathfrak{S}_{d+1}$. Also the group of $(d-p)$-form symmetries of $\mathfrak{S}_{d+1}$.

- $\mathcal{L}$: The defect group associated to $\mathfrak{S}_{d+1}$.

- $\mathfrak{T}_d^\Lambda$: An absolute $d$-dimensional theory associated to polarization $\Lambda$ obtained from a relative theory $\mathfrak{T}_d$.

- $\mathfrak{B}_d^\Lambda$: A topological boundary condition of a TQFT $\mathfrak{S}_{d+1}$ associated to polarization $\Lambda$.

- $\Lambda_p$: The subgroup of $\mathcal{L}_p$ characterizing the topological defects that can end on the boundary $\mathfrak{B}_d^\Lambda$.

- $\widehat{\Lambda}_{p+1}$: The $p$-form symmetry group of the absolute theory $\mathfrak{T}_d^\Lambda$.

### A.2 Glossary for Section 2.4

- $\mathfrak{T}_D$: A $D$-dimensional relative theory.

- $\mathfrak{T}_d$: A $d$-dimensional relative defect in $\mathfrak{T}_D$.

- $\mathfrak{S}_{D+1}$: The $(D+1)$-dimensional non-invertible TQFT associated to $\mathfrak{T}_D$.

- $\mathfrak{S}_{d+1}$: The $(d+1)$-dimensional non-invertible topological defect of $\mathfrak{S}_{D+1}$ associated to $\mathfrak{T}_d$.

- $\mathcal{L}_{D,p}$: The group formed by invertible $p$-dimensional topological defects of $\mathfrak{S}_{D+1}$. Also the group of $(D-p)$-form symmetries of $\mathfrak{S}_{D+1}$.

- $\mathcal{L}_{d,p}$: The group formed by invertible $p$-dimensional topological sub-defects of $\mathfrak{S}_{d+1}$. Also the group of $(d-p)$-form symmetries localized along $\mathfrak{S}_{d+1}$.

- $\mathcal{L}_{d,p}^0$: The group formed by invertible $p$-dimensional topological sub-defects of $\mathfrak{S}_{d+1}$ that are genuine i.e. unattached to other topological defects of $\mathfrak{S}_{D+1}$.

- $\mathcal{L}_D$: The defect group associated to $\mathfrak{S}_{D+1}$.

- $\mathcal{L}_d$: The defect group associated to $\mathfrak{S}_{d+1}$.

- $\pi_p$: The map from $\mathcal{L}_{d,p}$ to $\mathcal{L}_{D,p+1}$ describing the $(p+1)$-dimensional topological defect of $\mathfrak{S}_{D+1}$ attached to a $p$-dimensional topological sub-defect of $\mathfrak{S}_{d+1}$.

- $s_{d-p}$: The map from $\mathcal{L}_{D,D-p-1}$ to $\mathcal{L}_{d,d-p}^0$ describing the genuine topological sub-defect of $\mathfrak{S}_{d+1}$ obtained by squeezing a topological defect of $\mathfrak{S}_{D+1}$ onto $\mathfrak{S}_{d+1}$.

- $\mathcal{L}_{d,p}^T$: The group formed by invertible $p$-dimensional topological sub-defects of $\mathfrak{S}_{d+1}$ that are trapped i.e. cannot be related to topological defects of $\mathfrak{S}_{D+1}$.

- $\mathfrak{T}_D^{\Lambda_D}$: An absolute $D$-dimensional theory associated to polarization $\Lambda_D$ obtained from relative theory $\mathfrak{T}_D$.

- $\mathfrak{T}_d^{\Lambda_d}$: An absolute $d$-dimensional defect associated to polarization $\Lambda_d$ obtained from relative defect $\mathfrak{T}_d$.

- $\mathfrak{B}_D^{\Lambda_D}$: A topological boundary condition of $\mathfrak{S}_{D+1}$ associated to polarization $\Lambda_D$.

- $\mathfrak{B}_d^{\Lambda_d}$: A topological sub-defect of $\mathfrak{B}_D^{\Lambda_D}$ associated to polarization $\Lambda_d$.

- $\Lambda_{D,p}$: The subgroup of $\mathcal{L}_{D,p}$ characterizing the topological defects that can end on the boundary $\mathfrak{B}_D^{\Lambda_D}$.

- $\Lambda_{d,p}$: The subgroup of $\mathcal{L}_{d,p}$ characterizing the topological defects that can end on the boundary $\mathfrak{B}_d^{\Lambda_d}$.

- $\widehat{\Lambda}_{D,p+1}$: The $p$-form symmetry group of the absolute theory $\mathfrak{T}_D^{\Lambda_D}$.

- $\widehat{\Lambda}_{d,p+1}$: The $p$-form symmetry group of the absolute defect $\mathfrak{T}_d^{\Lambda_d}$.

## A.3 Glossary for the Rest of the Paper

- $\widehat{Z}_G$: The Pontryagin dual of the center $Z_G$ of a Lie group $G$.

- $\mathcal{O}_{\mathfrak{g}}$: The group of outer-automorphisms modulo inner automorphisms of a Lie algebra $\mathfrak{g}$. Also, the group of 0-form symmetries of a $6d$ $\mathcal{N}=(2,0)$ theory of type $\mathfrak{g}$.

- $\mathcal{L}_X$: The line defect group of the $4d$ $\mathcal{N}=2$ SCFT constructed by compactifying IIB on the Calabi-Yau threefold $X$.

- $\mathcal{L}_{\mathfrak{T}}$: The line defect group of a 4d $\mathcal{N} = 2$ relative theory $\mathfrak{T}$.

- $\mathcal{O}_{\mathfrak{T}_\Lambda}$: The 1-form symmetry group of a 4d $\mathcal{N} = 2$ absolute theory $\mathfrak{T}_\Lambda$.

- $\mathcal{S}_{6d}$: The surface defect group of a 6d $\mathcal{N} = (2,0)$ theory.

- $\mathcal{S}_{6d}^o$: The surface defect group of a 6d $\mathcal{N} = (2,0)$ theory modded out by the action of an outer-automorphism 0-form symmetry $o$.

- $\mathcal{S}_{6d,o}$: The surface defect group of a 6d $\mathcal{N} = (2,0)$ theory left invariant by the action of an outer-automorphism 0-form symmetry $o$.

- $\mathcal{P}$: A puncture of a 6d $\mathcal{N} = (2,0)$ theory.

- $\mathcal{P}_0$: A special untwisted puncture of a 6d $\mathcal{N} = (2,0)$ theory, that is obtained by gauging the flavor symmetry carried by an untwisted maximal regular puncture.

- $\mathcal{P}_0^o$: A special $o$-twisted puncture of a 6d $\mathcal{N} = (2,0)$ theory, that is obtained by gauging the flavor symmetry carried by an $o$-twisted maximal regular puncture.

- $\mathcal{L}_{\mathcal{P}}$: The full line defect group of a puncture $\mathcal{P}$.

- $\mathcal{L}_{\mathcal{P}}^T$: The trapped part of the line defect group of a puncture $\mathcal{P}$.

- $\mathcal{L}_{\mathcal{P}}^0$: The genuine part of the line defect group of a puncture $\mathcal{P}$.

- $\mathcal{H}_{\mathcal{P}}$: The magnetic part of the line defect group of a puncture $\mathcal{P}$.

- $\mathcal{H}_{\mathcal{P}}^T$: The trapped magnetic part of the line defect group of a puncture $\mathcal{P}$.

- $i_{\mathcal{P}}$: The injective map from $\mathcal{H}_{\mathcal{P}}^T$ to $\mathcal{H}_{\mathcal{P}}$ describing the trapped part as a subgroup of the full magnetic line defect group.

- $\mathcal{W}_{\mathcal{P}}$: The electric part of the line defect group of a puncture $\mathcal{P}$.

- $\mathcal{W}_{\mathcal{P}}^T$: The trapped electric part of the line defect group of a puncture $\mathcal{P}$.

- $Z_{\mathcal{P}}$: The subgroup of surface defects of a 6d $\mathcal{N} = (2,0)$ theory that can end at a puncture $\mathcal{P}$.

- $\pi_{\mathcal{P}}$: The projection map from $\mathcal{H}_{\mathcal{P}}$ to $Z_{\mathcal{P}}$ describing the bulk surface defect that a line defect living on puncture $\mathcal{P}$ is attached to.

- $\widehat{Z}_{\mathcal{P}}$: The subgroup of genuine line defects in $\mathcal{W}_{\mathcal{P}}$ that can be lifted to bulk surface defects.

- $\mathfrak{T}_{\mathcal{P}}$: A special 4d $\mathcal{N} = 2$ theory obtained by compactifying a 6d $\mathcal{N} = (2,0)$ theory on a sphere with a puncture $\mathcal{P}$. If $\mathcal{P}$ is an untwisted puncture, no other punctures are included. On the other hand, if $\mathcal{P}$ is a twisted puncture, then a minimal twisted puncture is also included.

- $\mathfrak{T}_{\mathcal{P}}^*$: A special 4d $\mathcal{N} = 2$ theory obtained by compactifying a 6d $\mathcal{N} = (2,0)$ theory on a sphere with a puncture $\mathcal{P}$. If $\mathcal{P}$ is an untwisted puncture, a maximal untwisted regular puncture is also included. On the other hand, if $\mathcal{P}$ is a twisted puncture, then a maximal twisted regular puncture is included.

- $\mathfrak{T}_{\mathcal{P}}^{\mathcal{P}_0}$: A special 4d $\mathcal{N} = 2$ theory obtained by compactifying a 6d $\mathcal{N} = (2,0)$ theory on a sphere with an untwisted puncture $\mathcal{P}$ and another puncture $\mathcal{P}_0$.

- $\mathfrak{T}_{\mathcal{P}}^{\mathcal{P}_0^o}$: A special 4d $\mathcal{N} = 2$ theory obtained by compactifying a 6d $\mathcal{N} = (2,0)$ theory on a sphere with an $o^{-1}$-twisted puncture $\mathcal{P}$ and another puncture $\mathcal{P}_0^o$.

- $\phi$ or $\Phi$: Higgs (Hitchin) field associated to Class S compactification.

- SNF($M$): Smith Normal Form of a matrix $M$.

# B   Spectral Cover Monodromies

Consider a puncture $\mathcal{P}$ in a theory of class S with bulk Lie algebra $\mathfrak{g}$. Let $t$ be a local coordinate for a patch of the UV curve centered on the puncture and $v$ a coordinate on the cotangent fibers projecting to this patch. The differential is $\lambda = v\,dt/t$ and the spectral curve is locally a polynomial

$$F(v,t) = 0. \tag{B.1}$$

For further details on the set-up see the discussion in section 5.3. The curves for untwisted Lie algebras $\mathfrak{g} = A_n, D_n, E_6$ were discussed in sections 7.5, 8.5, 10 respectively.

The vanishing locus of the discriminant of $F$ with respect to $v$ determines the $t$ coordinate of the branch points, these solve

$$\Delta(F,v)(t) = 0. \tag{B.2}$$

The monodromy action on the sheets $v_i$ of the spectral curve $F$ associated to the puncture $\mathcal{P}$ follows from considering a small loop linking $\mathcal{P}$ which does not enclose any of the branch points solving (B.2).

We present a convenient numerical method for determining the monodromy action on sheets. This method employs `Mathematica` and is independent of the bulk Lie algebra $\mathfrak{g}$ which determines the map from this monodromy action to the monodromy on roots after choosing a labelling of sheets by weights. As an example let us consider the curve

$$F(v,t) = P_{n_1}(v)t^2 + P_{n_2}(v)t + P_{n_3}(v) = 0, \tag{B.3}$$

with $n_2 - n_3 > n_1 - n_2 > 0$. Here $P_{n_i}$ are polynomials of degree $n_i$. We set all physical constants to one as we are solely interested in ramification structure of the cover at $t = 0$. The leading order terms of the curve in the limit $t \to 0$ are

$$F_0(v,t) = v^{n_1}t^2 + v^{n_2}t + v^{n_3} = 0. \tag{B.4}$$

The roots of this equation are $n_3$-fold degenerate for $v = 0$ and all $t$. This degeneracy needs to be lifted to study the monodromy at generic points of the extended Coulomb branch. We therefore tune to a more generic part of the extended Coulomb branch by turning on the constants $c_i$ as

$$P_0(v,t) = (v^{n_1} + c_1)t^2 + (v^{n_2} + c_2)t + v^{n_3} + c_3 = 0. \tag{B.5}$$

The physics of the set-up constrains which constants $c_i$ are permitted to be non-vanishing. The constant $c_i$ with the smallest index then determines the generic monodromy behaviour of the previously degenerate roots.

The roots of $P_0$ organize into three monodromy orbits of order $n_1 - n_2, n_2 - n_3, n_3$. The orbits group roots by their scaling behavior in the limit $t \to 0$. Roots of the first two orbits diverge upon approaching the puncture. Encircling the puncture counter-clockwise the orbits with $n_1 - n_2$ and $n_2 - n_3$ roots experience a cyclic permutation clockwise. The orbit with $n_3$ roots is permuted $3 - k$ times counter-clockwise where $k$ is the largest index for which $c_i \neq 0$.

An explicit example can be viewed using the `Mathematica` code:

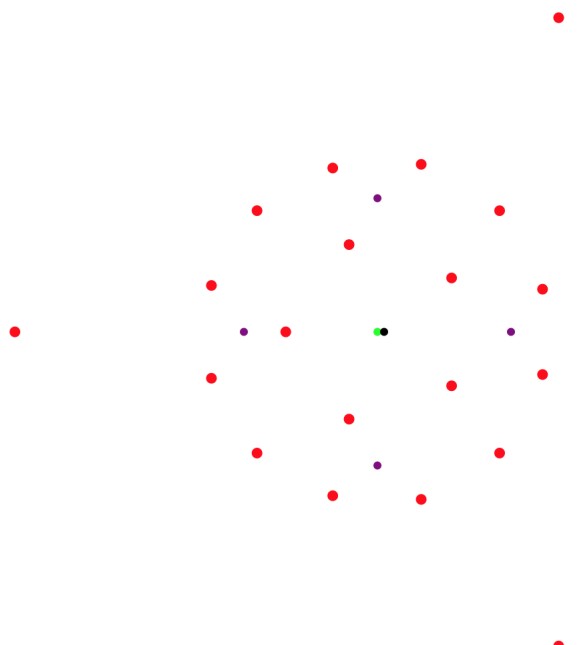

Figure 26: Example of the `Mathematica` out-put for $(n_1, n_2, n_3) = (20, 17, 5)$ and $(c_1, c_2, c_3) = (1, 1, 0)$ and $k = 2$. The $t$-plane and $v$-plane are plotted in the same plane. The black dot sets the value of $t$ and can be dragged. The green dot marks $t = 0$ while the purple dots mark a fixed reference frame in the t-plane. The roots $v_i(t)$ are displayed by red dots. Drag the black dot around the green dot to watch them permute. We clearly see three monodromy orbits of cardinality $3, 12, 5$ permuted cyclically.

```
Manipulate[n=20;index=Table[i,{i,n}];
Poly=v^5+(v^(17)+1)(t[[1]]+I t[[2]])+(v^(20)+1)(t[[1]]+I t[[2]])^2;
scale=0.5;roots=NSolve[Poly==0,v];rts=v/.roots[[index]];
Graphics[{Green,Disk[{0, 0},.03], Black,Disk[t,.03],Red,
Table[Disk[{Re[rts[[i]]],Im[rts[[i]]]},.04],{i,n}],Purple,
{Disk[{1,0},.03],Disk[{-1,0},.03],Disk[{0,1},.03],Disk[{0,-1},.03]}},
PlotRange->5, ImageSize->1000],{{t,{1/20,0}}, {-5,-5},{5,5},
Locator,Appearance->None},TrackedSymbols->True]
```

which produces the out-put shown in figure 26. More general polynomials $F(v, t)$ of arbitrary order in $t$ are analyzed similarly. Once the monodromy on sheets is known, a labelling of sheets by weights is picked and from it the monodromy action on roots is inferred.

As a final example we include the `Mathematica` code used in the analysis of the $E_6$ example of section 10.2.

```
Manipulate[n = 27; index = Table[i, {i, n}];
Poly = 1/27 - (53 v^9)/3 + 28 v^18 + v^27
- v^3/( 108 (t[[1]] + I t[[2]])^2) + (2 v^6)/(3 (t[[1]] + I t[[2]]))
+ (5 v^15)/(t[[1]] + I t[[2]]);
scale = 0.5; roots = NSolve[Poly == 0, v]; rts = v /.  roots[[index]];
```

```
Graphics[{Green, Disk[{0., 0}, .03], Black, Disk[t, .04], Red,
Table[Disk[{Re[rts[[i]]],
Im[rts[[i]]]}, .02], {i, 27}], Purple, {Disk[{1, 0}, .03],
Disk[{-1, 0}, .03], Disk[{0, 1}, .03], Disk[{0, -1}, .03]}},
PlotRange -> 5, ImageSize -> 1000], {{t, {1/2, 0}}, {-5, -5}, {5, 5},
Locator, Appearance -> None}, TrackedSymbols -> True]
```

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
