# Peer review of "Relative Defects in Relative Theories: Trapped Higher-Form Symmetries and Irregular Punctures in Class S"

_SciPost Physics, doi:SciPost Phys. 13, 101 (2022)_

## Round 1 · Referee Report · Anonymous · 2022-6-10

Report

This paper studied the line defect groups of large classes of irregular punctures in 6d $\mathcal{N}=(2,0)$ theories as well as the one-form symmetry and the line defect group of the Class S theories constructed using these punctures . These irregular punctures are interesting and important because they provide extra contributions to the one-form symmetry and the line defect group of the Class S theories. Conceptually, since the 6d $\mathcal{N}=(2,0)$ theories is a relative theory, these irregular punctures with nontrivial line defect group should be viewed as relative defects inside relative theories where the line defect group on the irregular punctures and the surface defect group of the 6d theory can have interesting interplay. The authors developed a general formalism for these relative defects in relative theories. Combining various techniques, they computed the line defects groups of large classes of punctures. Along the way, they also studied various new punctures that have not discussed in the literature.

The paper contained many interesting, useful and concrete results. I thus recommend the publication of this paper.

---

## Round 1 · Referee Report · Anonymous · 2022-8-22

Strengths

This paper provides a very clear and thorough discussion for relative defects in various settings. In particular it explains well how they differ from absolute defects and the various consequences associated to that. It is clear enough that it can serve as an example entrances to the subject.

The paper does a very nice job at exhibiting the phenomena of relative defects in class S theories. Here they provide a more fundamental exposition on what differentiates irregular punctures with regular punctures. In particular they discuss how the additional data needed to specify irregular punctures can be understood in terms of presences of relative defects stuck within the puncture.

Weaknesses

The paper is fairly long and will require significant investment of time to understand and appreciate what is in it. The authors could in the future consider splitting into two papers.

Report

This paper should be accepted. See above discussion.

---

## Editorial Decision

published